# Low-dose radiotherapy combined with dual PD-L1 and VEGFA blockade elicits antitumor response in hepatocellular carcinoma mediated by activated intratumoral CD8+ exhausted-like T cells

Siqi Li[1,2,7], Kun Li[1,3,7], Kang Wang[4,7], Haoyuan Yu[1,2], Xiangyang Wang[5], Mengchen Shi[6], Zhixing Liang[1,2], Zhou Yang[1,2], Yongwei Hu[1,2], Yang Li[1], Wei Liu[2], Hua Li[1]✉, Shuqun Cheng[4]✉, Linsen Ye[1,2]✉ & Yang Yang[1]✉

Atezolizumab (anti-PD-L1) combined with bevacizumab (anti-VEGFA) is the first-line immunotherapy for advanced hepatocellular carcinoma (HCC), but the number of patients who benefit from this regimen remains limited. Here, we combine dual PD-L1 and VEGFA blockade (DPVB) with low-dose radiotherapy (LDRT), which rapidly inflames tumors, rendering them vulnerable to immunotherapy. The combinatorial therapy exhibits superior antitumor efficacy mediated by CD8+ T cells in various preclinical HCC models. Treatment efficacy relies upon mobilizing exhausted-like CD8+ T cells (CD8+ Tex) with effector function and cytolytic capacity. Mechanistically, LDRT sensitizes tumors to DPVB by recruiting stem-like CD8+ Tpex, the progenitor exhausted CD8+ T cells, from draining lymph nodes (dLNs) into the tumor via the CXCL10/CXCR3 axis. Together, these results further support the rationale for combining LDRT with atezolizumab and bevacizumab, and its clinical translation.

Hepatocellular carcinoma (HCC) incidence has increased annually, and it is the third-leading cause of cancer-related death worldwide[1]. Despite efforts to advocate for early screening of at-risk individuals, more than 80% of patients are diagnosed at a late stage with unresectable disease, and current treatment strategies are inadequate in preventing the high metastasis and recurrence rates of HCC patients[2,3]. Therefore, multidisciplinary therapy (MDT) for HCC has received attention. Recently, atezolizumab (anti-PD-L1) combined with bevacizumab (anti-VEGFA)

(T + A) showed promising efficacy in the treatment of unresectable HCC. Furthermore, phase III clinical trial IMbrave150 showed that the T + A therapy group had longer overall and progression-free survival than the sorafenib treatment group, making it the first successful first-line immunotherapy for HCC around the world[4–6]. Although T + A therapy achieved more favorable survival outcomes than conventional sorafenib therapy, only 27.3% of patients with HCC could benefit from it, and the therapy lacked effective predictors[4]. Meanwhile, the clinical

[1]Department of Hepatic Surgery and Liver Transplantation Center, The Third Affiliated Hospital of Sun Yat-sen University, Guangzhou 510630, China. [2]Guangdong Provincial Key Laboratory of Liver Disease Research, The Third Affiliated Hospital of Sun Yat-sen University, Guangzhou 510630, China. [3]Department of Biotherapy Center, The Third Affiliated Hospital of Sun Yat-sen University, Guangzhou 510630, China. [4]Department of Hepatic Surgery VI, Eastern Hepatobiliary Surgery Hospital, Naval Medical University, Shanghai 200433, China. [5]Scientific Research Center, The Seventh Affiliated Hospital of Sun Yat-sen University, Shenzhen, Guangdong 517108, China. [6]Department of Clinical Laboratory, The Sixth Affiliated Hospital, Sun Yat-sen University, Guangzhou, Guangdong 510655, China. [7]These authors contributed equally: Siqi Li, Kun Li, Kang Wang. ✉e-mail: lihua100@yeah.net; chengshuqun@aliyun.com; ye_linsen@163.com; yysysu@163.com

study also highlighted the therapeutic potential value of combining immunotherapy with a tumor microenvironment (TME) modulator[4]. HCC is a solid tumor with complex pathophysiological barriers, poor infiltration of immune cells and a strong tumor immunosuppressive microenvironment, greatly limiting the effectiveness of immunotherapy. Thus, further optimization of HCC tumor immune microenvironment characteristics will be of great significance for improving the efficacy of immunotherapy in HCC patients.

Radiotherapy (RT) is one of the mainstream treatments for HCC. In addition to directly killing tumor cells, RT also mediates the immunomodulatory effects of the TME[7]. Based on the effect of RT on the tumor immune microenvironment, an increasing number of preclinical and clinical studies have begun to investigate the antitumor efficacy of combined RT and immunotherapy[8–13]. Although most studies have shown superior effects of RT and immunotherapy combination therapy, there are no guidance-recommended irradiation dose and fractionated regimes or RT combined immunotherapy regimes, as combination therapy is still in the preliminary stage. Furthermore, studies have noted that in patients with Child–Pugh (CP) scores >7, the risk of radiation-induced liver disease (RILD) and CP score deterioration after conventional-dose RT is significantly higher[14,15]. Many HCC patients have a history of liver fibrosis, liver function damage, and portal hypertension, which are the factors leading to RILD; thus, the radiotherapy dose of the liver needs to be strictly controlled in the above HCC patients[16]. Therefore, it is critical to further explore the synergistic effect of RT and immunotherapy combined with appropriate doses and regimes to improve the prognosis of patients with HCC.

Recently, an increasing number of studies have focused on low-dose radiation therapy (LDRT, i.e., 0.5–2 Gy) in cancer immunotherapy. Preclinical studies have shown that LDRT can reshape the tumor immune microenvironment by transiently inflaming tumors, rendering them more suitable for immunotherapy in various tumors[17–19]. One of the research groups translated the preclinical findings into a phase I clinical trial investigating the combination of LDRT, cyclophosphamide and immune checkpoint blocking (ICB) in tumor patients with immune desertification, which supported the rationale of combining LDRT and immunotherapy to effectively treat tumors with low immune cell infiltration[17]. Furthermore, clinical evidence has shown that preoperative LDRT can achieve portal vein tumor thrombus (PVTT) downstaging for HCC patients, reducing the postoperative recurrence rate without increasing the risk of surgery and the incidence of postoperative liver failure. However, whether LDRT sensitizes T + A therapy and how LDRT mechanistically enhances the antitumor effect of T + A therapy remain unclear.

During tumor development, CD8+ T cells exhaustion (CD8+ Tex) is a gradual process[20–22]. Terminal exhausted CD8+ T cells are characterized by loss of proliferative potential and effector function (i.e., the ability to produce TNF-α and IFN-γ), as well as increased expression of several inhibitory receptors (e.g., PD-1 and Tim-3) and exhaustion-associated transcription factors (e.g., Blimp-1, Tox and Eomes)[23,24], which largely limit the antitumor immune response of CD8+ T cells. CD8+ Tex are mainly derived from the undifferentiated progenitor exhausted CD8+ T cells (CD8+ Tpex), which are also called stem-like CD8+ Tpex, due to their capacity for expansion, regeneration, and differentiation[25]. In contrast to terminally exhausted CD8+ T cells, CD8+ Tpex have recently been described to generate effector T-cell progeny[26]. After ICB treatment, these stem-like CD8+ Tpex can rapidly differentiate into transitory effector CD8+ T cells with strong effector function and cytolytic ability, which are maintained for a short time, and finally differentiate into terminally exhausted CD8+ T cells[27]. Previous studies have shown that stem-like CD8+ Tpex cells have at least two functions in the chronic immune response: maintaining the ongoing T-cell response and mediating the response to ICB therapy[28]. The expression of TCF1 is required for both functions and provides an important molecular marker for the recognition of stem-like CD8+

Tpex[23]. Increasingly, studies have shown that a small number of TCF1+ PD-1+ CD8+ T cells are present in tumors, and their presence is associated with better outcomes after immunotherapy[23,24,28]. Therefore, further exploration of the mechanism of expanding and maintaining the stem-like CD8+ Tpex pool is of great importance for improving the response to immunotherapy combinations.

Here, we investigate whether LDRT could enhance the antitumor effect of dual VEGFA and PD-L1 blockade (DPVB) in various preclinical HCC model. We make the discovery that the combinatorial therapy exhibits superior CD8+ T cells-mediated antitumor efficacy. High-dimensional single-cell RNA sequencing (scRNA-seq) reveals that LDRT combined with DPVB enhances tumor-rejecting CD8+ Tex with effector function and cytolytic capacity. Moreover, using multiple HCC preclinical models and an ex vivo tumor fragment platform, we dissect the mechanisms by which LDRT sensitizes DPVB by enlarging intratumoral stem-like CD8+ Tpex, which are recruited from the draining lymph nodes (dLNs) into the tumor through the CXCL10/CXCR3 axis. This study provides preclinical evidence for the clinical translation of this combination treatment strategy.

## Results
### Combined LDRT and DPVB (LR-DPVB) induces tumor regression
To assess the antitumor effects of LDRT in combination with T + A treatment, we utilized three distinct mouse models of liver neoplasia induced by carcinogens: Hepa1-6 orthotopic tumor-bearing mouse model (Hepa1-6 HCC model), diethylnitrosamine and repeated carbon tetrachloride exposure-induced mouse HCC model (DEN+CCl4 HCC model) as well as hydrodynamic tail vein injection $Trp53^{KO}/MYC^{OE}$ somatic genome editing-mediated mouse HCC model ($Trp53^{KO}/MYC^{OE}$ HCC model) (Fig. 1a, e, k). The mouse-reactive antibodies, anti-PD-L1 and anti-VEGFA, were utilized for subsequent experiments involving dual PD-L1 and VEGFA blockade (DPVB). Compared with DPVB alone, LR-DPVB showed tumor regression and survival benefit in Hepa1-6 HCC model (Fig. 1b–d, Supplementary Fig. 1a, b). Most mice that received LR-DPVB exhibited a tumor regression response (57.14% (8/14) of complete response (CR) and 28.6% (4/14) reached partial response (PR)) on day 14 (Fig. 1b, c, Supplementary Fig. 1a). Subsequently, we employed the DEN+CCl4 HCC model, which was established through a single injection of DEN (25 mg/kg) followed by repeated administration of CCl4 (20%, 1 μl/g, twice a week), to faithfully replicate genotoxic injury and advanced fibrosis-associated HCC in humans (Fig. 1e). Immunohistochemical staining (IHC) revealed a scarcity of CD8+ T cell infiltrates in the liver tumors of the CON group from the DEN + CCl4 HCC model (Fig. 1i), indicating that this model accurately represents an immune desert environment resembling "cold" human HCC with limited responsiveness to immunotherapy. In the DEN+CCl4 HCC model, treatment with LR-DPVB demonstrated efficacy in reducing advanced liver lesions, as evidenced by gross appearance, liver weight, tumor numbers, and H&E staining (Fig. 1f–j). Compared with the DPVB group, mice in the LR-DPVB showed robust tumor control with a lower liver weight and fewer tumor nodules (Fig. 1f–h). To validate this observation of LR-DPVB efficacy and establish the clinical relevance between efficacy and key driver genes of human HCC, we exploited hydrodynamic tail-vein injections to generate a $Trp53^{KO}/MYC^{OE}$ HCC model in which oncogenic MYC can be genomically integrated and Trp53 is deficient to recapitulate the features of HCC, as previously described[29–31]. We observed more therapeutic effects and survival benefits from LR-DPVB compared to other groups in $Trp53^{KO}/MYC^{OE}$ HCC model which is known to be an immune desert (cold) model with few CD8+ T cell infiltrates and ICB resistance reported in previous studies[29] (Fig. 1l–p). Notably, LDRT alone did not exert a significant impact on tumor progression across all three HCC models (Fig. 1b–d, f–j, l–p, Supplementary Fig. 1b, d), which was consistent with previous findings of LDRT in an orthotopic intraperitoneal murine model[17].

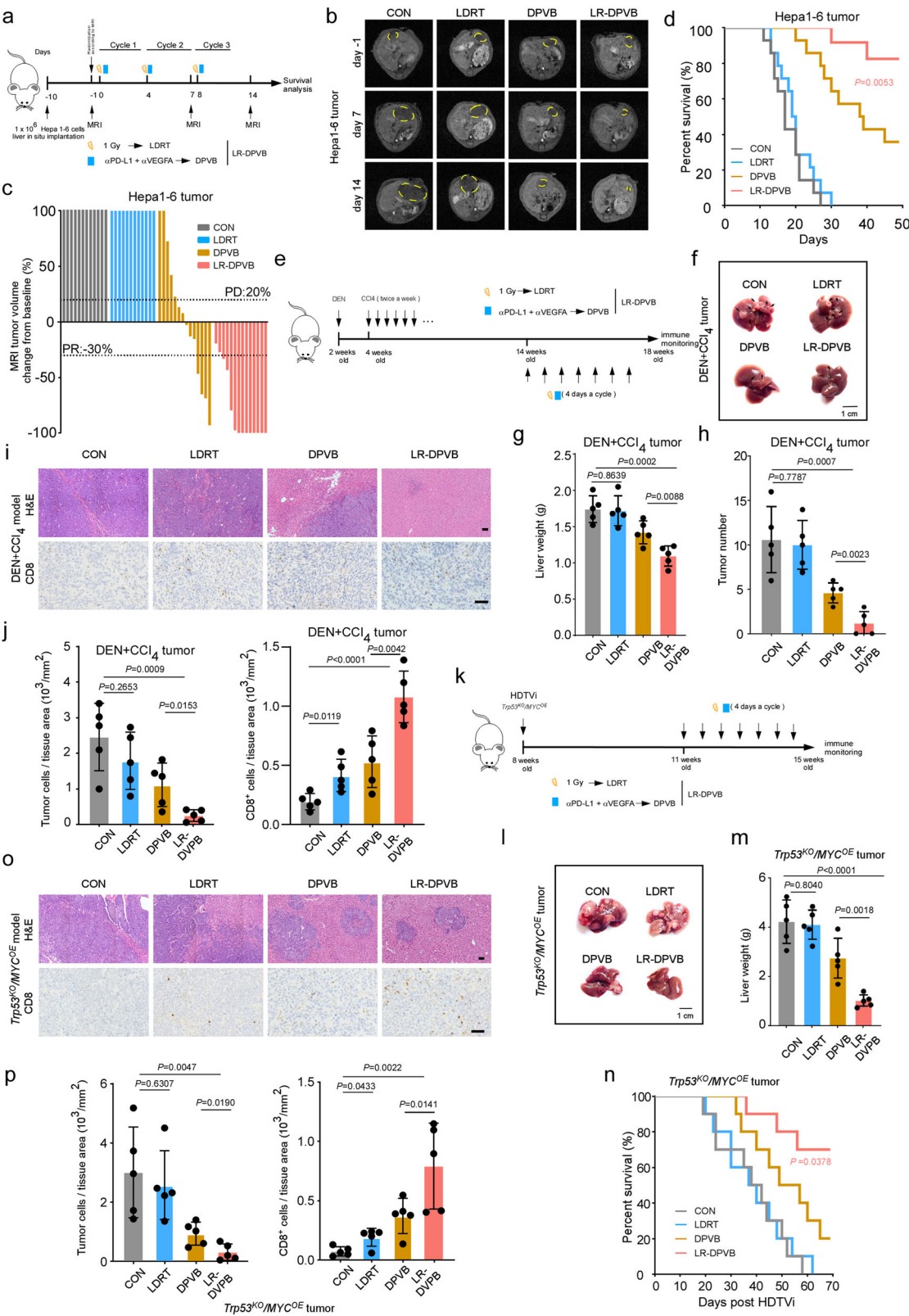

Additionally, no significant differences in mouse body weight were observed between the LR-DPVB group and other groups in the Hepa1-6 HCC model (Supplementary Fig. 1c). Liver hematoxylin-eosin (H&E) staining revealed no histopathological lesions in Hepa1-6 HCC mice treated with LR-DPVB (Supplementary Fig. 1d). Moreover, there were no significant differences in circulating liver enzymes alanine transaminase (ALT) and aspartate aminotransferase (AST) between the LR-DPVB group and other groups in the same HCC mouse model (Supplementary Fig. 1e–j). These findings suggest minimal hepatotoxicity associated with long-term administration of LR-DPVB.

Collectively, these results confirmed that LR-DPVB led to a marked therapeutic response with no obvious toxicity in multiple preclinical HCC models.

**Fig. 1 | Combined LDRT and DPVB (LR-DPVB) confers tumor responsiveness.** **a** Schematic illustration of the in vivo study evaluating different regimes in Hepa1-6 HCC model. **b** Representative images of Hepa1-6 HCC model liver MRI scanning. **c** Waterfall plot of the percentage change of the tumor volume for Hepa1-6 HCC model with different treatments. $n = 14$ mice/group. **d** Kaplan−Meier curve of Hepa1-6 HCC model mice treated with 4 different regimes ($n = 14$ mice/group). **e** Schematic illustration of the in vivo study evaluating different regimes in DEN + CCl$_4$ HCC model. **f** Representative image of DEN + CCl$_4$ HCC model. **g** The liver weight of DEN + CCl$_4$ HCC model. **h** The tumor number of DEN + CCl$_4$ HCC model. **i** H&E staining for livers from DEN+CCl$_4$ HCC model (top), representative images (bottom) of CD8 IHC staining in sections of tumor tissue in indicated groups. Scale bars: 50 μm. **j** Quantification of tumor cells (X10$^3$) / tissue area (mm$^2$) and CD8$^+$ T cells (left) and CD8$^+$ T cells (X10$^3$) / tissue area (mm$^2$) (right) of sections in indicated groups of DEN+CCl$_4$ tumors. **k** Schematic illustration of the in vivo study evaluating different regimes in $Trp53^{KO}/MYC^{OE}$ HCC model. **l** Representative image of $Trp53^{KO}/MYC^{OE}$ HCC model. **m** The liver weight of $Trp53^{KO}/MYC^{OE}$ HCC model. **n** Kaplan−Meier curve of $Trp53^{KO}/MYC^{OE}$ HCC model ($n = 10$ mice/group). **o** H&E staining for livers from $Trp53^{KO}/MYC^{OE}$ HCC model (top), representative images (bottom) of CD8 IHC staining in sections of tumor tissue in indicated groups. Scale bars: 50 μm. **p** Quantification of tumor cells (X10$^3$) / tissue area (mm$^2$) (left) and CD8$^+$ T cells (X10$^3$) / tissue area (mm$^2$) (right) of sections in indicated groups of $Trp53^{KO}/MYC^{OE}$ tumors. Data shown as means ± SD. **g**−**h**, **j**, **l**−**m** and **p**, $n = 5$ mice/group. $P$ values of **d** and **n** were determined by log-rank test (Mantel-Cox). $P$ values of **g**, **h**, **j** and **p** were calculated using a two-sided unpaired Student's $t$ test. Source data are provided as a Source Data file.

## LR-DPVB exhibits superior CD8$^+$ TIIC-mediated antitumor efficacy

To uncover the immunological cellular mechanisms underlying the enhanced therapeutic efficacy of combination therapy, we analyzed the immune subset profiles in the above three models. We employed single-cell RNA sequencing (scRNA-seq) to analyze tumor-infiltrating immune cells (TIICs) in Hepa1-6 HCC tumors from four groups (CON, LDRT, DPVB, LR-DPVB) (Supplementary Fig. 2a, Fig. 2a: Reference map). The scRNA-seq identified eight distinct cell states (Fig. 2a, Supplementary Fig. 2b, c). The most striking changes in TIICs from tumors treated with LR-DPVB were the highest enrichment in the CD8$^+$ T cell population, directly linking the CD8$^+$ T cell compartment to tumor control (Fig. 2b, c, Supplementary Fig. 2d). Correspondingly, flow cytometry and immunohistochemistry (IHC) serial sections analysis of resected tumors showed obvious LR-DPVB-induced accumulation of CD8$^+$ T cells, whereas CD4$^+$ T cells were not significantly changed (Fig. 2d−g, Supplementary Fig. 2e, f). Next, we employed the DEN + CCl$_4$ HCC model to investigate the correlation between the LR-DPVB efficacy and immune subset profiles. To better elucidate the subsets of immune cells that play the greatest role in the efficacy of combination therapy, we sorted CD45$^+$ TIICs from four groups of the DEN + CCl$_4$ HCC model for scRNA-seq analysis. The scRNA-seq analysis revealed the presence of nine distinct cellular states (Fig. 2h, Supplementary Fig. 3a). Comparative analysis of immune cell proportions demonstrated that CD8$^+$ T cells exhibited the most pronounced changes following LR-DPVB treatment in the DEN + CCl$_4$ model (Fig. 2h, i). Furthermore, flow cytometry and IHC analysis confirmed a noticeable increase in CD8$^+$ T cell accumulation induced by LR-DPVB treatment in both DEN + CCl$_4$ model and $Trp53^{KO}/MYC^{OE}$ HCC model (Figs. 1i, j, o, p, Fig. 2j−m).

To assess whether CD8$^+$ T cells are essential for the therapeutic efficacy of LR-DPVB, we next conducted T-cell depletion experiment in above three mice HCC model. The results clearly indicated that depletion of CD8$^+$ T cells compromised the anti-tumor effect of LR-DPVB, while depletion of CD4$^+$ T cells did not exert a significant impact on tumor progression (Supplementary Fig. 2g−h, Supplementary Fig. 3b−e).

Collectively, these data showed that the intratumoral accumulation of CD8$^+$ T cells largely governed the tumor size decrease and the therapeutic benefit of LR-DPVB.

## LR-DPVB reinforces the effector function and cytolytic capacity of tumor-rejecting CD8$^+$ Tex

To gain insight into the CD8$^+$ T cell phenotypes that benefit from LR-DPVB versus DPVB, we first analyzed the differentially expressed genes (DEGs) in CD8$^+$ T cells between LR-DPVB and DPVB in Hepa1-6 tumors (Supplementary Fig. 4a, b). Volcano plot and heatmap analyses showed that the CD8$^+$ T cell in LR-DPVB group exhibited significantly elevated levels of exhaustion markers (*Lag3, Pdcd1, Havcr2*), exhaustion-associated transcription factors (*Nr4a2, Tox, Runx3*) and activation/effector/cytolysis molecules (*Tnfrsf9, Fasl, Tbx21, Nfatc1*) (|log$_2$FC|>

0.25 and $p < 0.05$) compared to those in the DPVB group. By further analyzing of these scRNA-seq data, we also identified that CD8$^+$ T cells populations featured with a higher exhaustion and effector functions were inclined to accumulate after LR-DPVB than after DPVB (Supplementary Fig. 4c). Notably, DEGs enrichment of CD8$^+$ T cells from LR-DPVB-treated tumors were significantly positively associated with T cell differentiation and the T cell receptor signaling pathway (Supplementary Fig. 4d). In the DEN + CCl$_4$ HCC model, the DEGs enrichment analysis of CD8$^+$ T cell from LR-DPVB-treated tumors also revealed a significant association with T cell differentiation and activation, effector molecular response, as well as inflammatory and immune responses (Supplementary Fig. 5a). The volcano map and heatmap showed significantly increased levels of activation/effector markers (*Tnfsf8, Tnfaip8, Gzma, Gzmg, Tnfsf11, Cd28*) (|log2FC|> 0.25 and $p < 0.05$) in the CD8$^+$ T cell of the LR-DPVB group compared with those of the DPVB group (Supplementary Fig. 5b, c). Meanwhile, flow cytometry confirmed the significant enrichment of CD8$^+$ T cell with exhaustion, effector function and cytolytic capacity in LR-DPVB-treated tumors of the above three models (Supplementary Fig. 4e−h, 5d−g). Collectively, these results suggested that the efficacy of LR-DPVB may be related to the exhausted and effected function of CD8$^+$ T cells.

To further elucidate the in-depth mechanisms behind the CD8$^+$ T cell-mediated curative effects after LR-DPVB, we explored the subpopulations of tumor-infiltrating T cells in detail. By utilizing ProjecTILs to map the aforementioned scRNA-seq data onto a reference TIL spectrum, we recognized nine different T-cell clusters that displayed a relative expression pattern of essential markers defining the lineage of T cells across tumor types (Fig. 3a, b, Supplementary Fig. 6a). As expected, T cells infiltration from LR-DPVB-treated tumors was prevalent in exhausted and effector CD8$^+$ TIL populations, which mainly included exhausted CD8$^+$ T cells (CD8_Tex), progenitor-exhausted CD8$^+$ T cells (CD8_Tpex) and effector-memory CD8$^+$ T cells (CD8_EM) (Fig. 3a, c, Supplementary Fig. 6b). Compared with that in DPVB-treated tumors, CD8_Tex was the most differentially enriched T cell subset after LR-DPVB treatment (Fig. 3a, c, Supplementary Fig. 6b). Importantly, the CD8_Tex from the LR-DPVB-treated tumors expressed higher levels of molecules with effector function and cytolytic capacity related to DPVB, such as *Ifng, Tnf*, and *Gzmb* (Fig. 3d). These data collectively revealed that the enhancement of the tumor-rejecting effect after LR-DPVB treatment was largely dependent on the magnitude of intratumoral CD8$^+$ Tex, which featured with potent effector function and cytolytic capacity. Pseudotime curve analysis revealed an intermediate evolution state among CD8_Tpex differentiating into CD8_Tex (Supplementary Fig. 6c, Fig. 3e). We analyzed the characteristic genes of three cell subsets along this trajectory (Supplementary Fig. 8b). As the heatmap showed, the left cell subset of both heatmap and pseudotime curve was characterized by significant upregulation of progenitor-associated genes (*Tcf7* and *Slamf6*) and activation-related genes (*Tnfrsf4, Tnfrsf18, Nr4a3*), medium expression of exhaustion genes (*Havcr2, Pdcd1*), indicating an

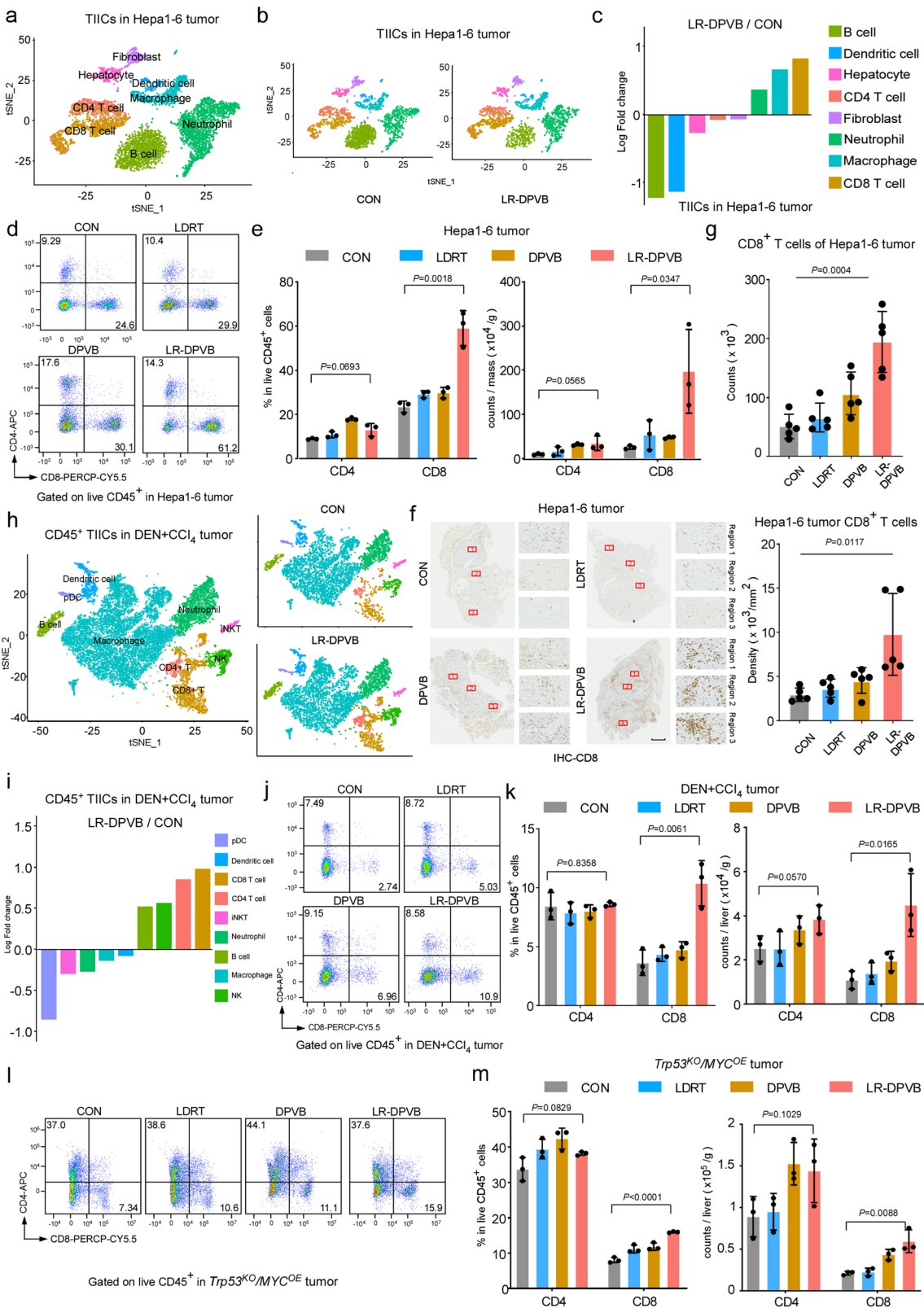

exhausted precursor CD8_Tpex state (CD8_Tpex'); the right cell subset was distinctly reduced in progenitor-associated genes and activation-related genes, as well as at the end state of evolution, which suggested a terminal exhausted state (CD8_Tex_term); the middle cell subset had obviously higher expression of cytolysis-associated genes (*Gzma, Gzmb, Prf1, Nlrc1, Nkg1*) during the evolution path, which was highly consistent with the previously defined transitory effected T cells[27]

among the exhausted process of CD8⁺ T cell (CD8_Tef) (Supplementary Fig. 8b, Fig. 3f). We used the pseudotime algorithm of monocle to calculate the genes whose expression variation trend was most relevant to the time trajectory, and the gene variation trend was relatively smooth (Supplementary Fig. 8c). Furthermore, related to DPVB, the pseudotime curve of exhausted CD8⁺ T cells (including CD8_Tpex', CD8_Tef, CD8_Tex_term) from LR-DPVB-treated tumors showed

**Fig. 2 | The antitumor effect of LR-DPVB is mediated by CD8$^+$ T cells. a**, **b** tSNE maps of scRNAseq data from TIICs of Hepa1-6 HCC tumors ($n = 3$ mice/ group). **c** Fold-change of TIICs populations from LR-DPVB and CON in Hepa1-6 HCC tumors. $n = 3$ mice/group. **d** Representative flow cytometry plots of % CD4$^+$ and % CD8$^+$ in CD45$^+$ cells of Hepa1-6 HCC tumor TIICs. **e** Percentage quantification (left panel) of % CD4$^+$ and % CD8$^+$ in CD45$^+$ cells and immune cell count/tumor weight (right panel) in Hepa1-6 HCC tumor TIICs. $n = 3$ mice/group. **f**, **g** Representative images **f** and quantification **g** of CD8 IHC staining in serial sections of Hepa1-6 HCC tumor tissue in indicated groups. Scale bars: 500 μm. $n = 5$ mice/group. **h** tSNE maps of scRNAseq data from CD45$^+$ TIICs of DEN + CCl$_4$ HCC tumors ($n = 3$ per group). **i** Fold-change of CD45$^+$ TIICs populations from LR-DPVB and CON groups in DEN + CCl$_4$ HCC tumors. $n = 3$ mice/group. **j** Representative flow cytometry plots of

% CD4$^+$ and % CD8$^+$ in CD45$^+$ cells in DEN + CCl$_4$ HCC tumor TIICs. **k** Percentage quantification (left panel) of % CD4$^+$ and % CD8$^+$ in CD45$^+$ cells and immune cell count/liver weight (right panel) in DEN + CCl$_4$ HCC tumor TIICs. $n = 3$ mice/group. **l** Representative flow cytometry plots of % CD4$^+$ and % CD8$^+$ in CD45$^+$ cells in $Trp53^{KO}/MYC^{OE}$ HCC TIICs at the end of the therapeutic cycle. **m** Percentage quantification (left panel) of % CD4$^+$ and % CD8$^+$ in CD45$^+$ cells and immune cell count/ liver weight (right panel) in $Trp53^{KO}/MYC^{OE}$ HCC TIICs. $n = 3$ mice/group. Data of **e**, **k** and **m** shown as means ± SD derived from tumor mouse models ($n = 3$ mice/group). Data of **g** shown as means ± SD derived from tumor mouse models ($n = 5$ mice/group). $P$ values of **e**, **g**, **k** and **m** were calculated using a two-sided unpaired Student's $t$ test. Source data are provided as a Source Data file.

significant enrichment of CD8_Tef (Fig. 3f, g). Last but not least, the effector and cytolysis-related genes (*Tnf*, *Gzmb*, *Ifnγ*) were higher in CD8_Tef from tumors with LR-DPVB than in those with DPVB, which strongly supported that the tumor-rejecting CD8$^+$ Tex with effector function and cytolytic capacity was mainly contributed by the CD8_Tef subset (Fig. 3h). We recurred to DEN + CCl$_4$ HCC models to analyze the alterations in T cell subsets following the LR-DPVB. Single-cell RNA-seq analysis of T cell from DEN + CCl$_4$ HCC model tumors with 4 different treatments revealed seven cell transcriptomic states (Supplementary Fig. 5h-i). The pseudotime analysis provided support for a model of state evolution, wherein the progenitor CD8$^+$ T cells (CD8_Tpex) undergo differentiation into terminally differentiated exhausted CD8$^+$ T cells (CD8_Tex_term) via an intermediate transitory effector-like CD8$^+$ T state (CD8_Tef) (Supplementary Fig. 6e). The CD8_Tef subset exhibited the most significant enrichment in the DEN + CCl$_4$ HCC model with LR-DPVB treatment, as compared to the DPVB-treated tumors (Supplementary Fig. 6f). Comprehensive analysis of exhausted CD8_T cell (including CD8_Tpex, CD8_Tef, and CD8_Tex_term) from LR-DPVB-treated DEN + CCl$_4$ HCC tumors revealed a significant enrichment of CD8_Tef compared to tumors treated with DPVB (Supplementary Fig. 6g). Moreover, the volcano map and heatmap showed significantly increased levels of effector and activated association markers (*Tnfrsf9*, *Il2ra*, *Tnfrsf23*, *Tnfsf11*, *Gzme*, *Gzmc*, *Tnfaip3*, *Gzmb*, *Il2rb*) (|log2FC|> 0.25 and $p < 0.05$) in the CD8_Tef of the LR-DPVB group compared with the DPVB group (Supplementary Fig. 6h, i). The violin diagram also showed a higher effector function and cytolytic molecules were inclined to accumulate after LR-DPVB (Supplementary Fig. 6j). The enrichment analysis of DEGs of LR-DPVB/DPVB in CD8_Tef revealed a significant association with the cytokine-mediated signaling pathway in LR-DPVB treated DEN-CCl$_4$ HCC tumors (Supplementary Fig. 6k). Flow cytometry analysis in the aforementioned models confirmed a significant enrichment of effector and cytolytic markers in TCF1$^-$ PD1$^+$ CD8$^+$ T cells following LR-DPVB treatment (Fig. 3i, Supplementary Fig. 7a–c).

Notably, the Treg/T-cell subset ratio was significantly decreased by LR-DPVB compared with DPVB, especially Treg/CD8_Tef, in both the Hepa1-6 HCC model and DEN + CCl$_4$ HCC model (Fig. 3j, k, Supplementary Fig. 6l). These results suggested that Treg/CD8_Tef may become the curative effect index after LR-DPVB treatment.

As illustrated of scRNA-seq analysis in Supplementary Fig. 2d and Fig. 2c, except for the CD8 T cell population, macrophages exhibited the highly enrichment among tumor-infiltrating immune cells (TIICs) in Hepa1-6 HCC tumors treated with LR-DPVB compared to the CON group. In addition, we also analyzed the alterations of immune cell populations between LR-DPVB and DPVB in Hepa1-6 HCC tumors by scRNA-seq. Notably, compared to DPVB alone, the proportion of macrophages was significantly elevated in TIICs of Hepa1-6 HCC tumors following treatment with LR-DPVB (Supplementary Fig. 9a). These findings suggest that macrophages may also play an important role in the therapeutic efficacy of LR-DPVB. The pivotal role of antigen-presenting cells (APCs) in the immune response to tumors following radiotherapy has been widely acknowledged[17,32], thus prompting our

further investigation into alterations in APC populations subsequent to LR-DPVB treatment. As depicted in Supplementary Fig. 9b, LR-DPVB significantly upregulated the proportion of macrophages. Flow cytometry analysis confirmed a significant increase in macrophage infiltration following LR-DPVB treatment compared to DPVB alone (Supplementary Fig. 9c). Additionally, scRNA-seq data analysis revealed that LR-DPVB treatment resulted in an upregulation of the inflammation marker *Tnfrsf1a* within the macrophage compartment of Hepa1-6 tumors (Supplementary Fig. 9d). Macrophages are commonly classified as pro-inflammatory M1-like and anti-inflammatory M2-like. By assessing the gene expression levels of *Nos2* (CD86) and *Mrc1* (CD206), we briefly identified distinct populations of M1-type and M2-type macrophages of DEN + CCl$_4$ tumors (Supplementary Fig. 9e, f). ScRNA-seq analysis revealed that LR-DPVB exhibited an upregulation of genes associated with enhanced antigen presentation ability, such as *H2k1* and *H2d1* (Supplementary Fig. 9g). We utilized flow cytometry to assess the phenotype alteration of macrophages. The LR-DPVB group exhibited a significant upregulation of the M1-like macrophage phenotype in Hepa1-6 tumors, while there was a marked decrease in the accumulation of M2-like macrophages, indicating recruitment and M1-repolarization of macrophages with LR-DPVB-induced immunoactivity (Supplementary Fig. 9h). Meanwhile, we found different status of dendritic cells (DCs) of scRNA-seq data from DEN + CCl$_4$ tumor, which could be further annotated as Type 1 conventional dendritic cells (cDC1), cDC2, mature DC[33,34] (Supplementary Fig. 9i, j). Notably, the infiltration of cDC1 and mature DCs was significantly enhanced in tumors treated with LR-DPVB and LDRT, compared to the DPVB-treated tumors and CON group tumor, respectively (Supplementary Fig. 9k). The cDC1 has been reported to play a crucial role in the therapeutic responses to immune checkpoint blockade (ICB)[35–37]. Volcano plot revealed that LR-DPVB exhibited an upregulation of genes associated with major histocompatibility complex-I (MHC-I) presentation, such as *Psma5* and *H2d1*, both in DCs and mature DC (Supplementary Fig. 9l, m). Flow cytometry analysis confirmed the upregulation of antigen-presenting markers CD86 and MHC-I molecules in tumor-infiltrating DCs of the LR-DPVB-treated group compared to the DPVB-treated group (Supplementary Fig. 9n), indicating enhanced maturation and antigen-presenting function. Collectively, these findings provide compelling evidence supporting the significant enhancement of pro-inflammatory innate immunocyte infiltration into tumors by combining LDRT with DPVB treatment, promoting activation of the tumor's innate immune microenvironment.

## LDRT enlarges intratumoral stem-like CD8$^+$ Tpex, the precursor of CD8$^+$ Tex, which is essential for DVPB sensitization

To further investigate the mechanism of LDRT-sensitized DPVB therapy effectiveness, we evaluated the alteration of cell subsets in T-cell infiltration of LDRT-treated tumors in HCC mouse models (Hepa1-6 HCC model, DEN+CCl$_4$ HCC model and $Trp53^{KO}/MYC^{OE}$ HCC model). The scRNA-seq analysis showed that T cells infiltration from LDRT-treated Hepa1-6 tumors and DEN+CCl$_4$ tumor was both highly abundant in CD8_Tpex populations relative to the untreated group (Fig. 4a,

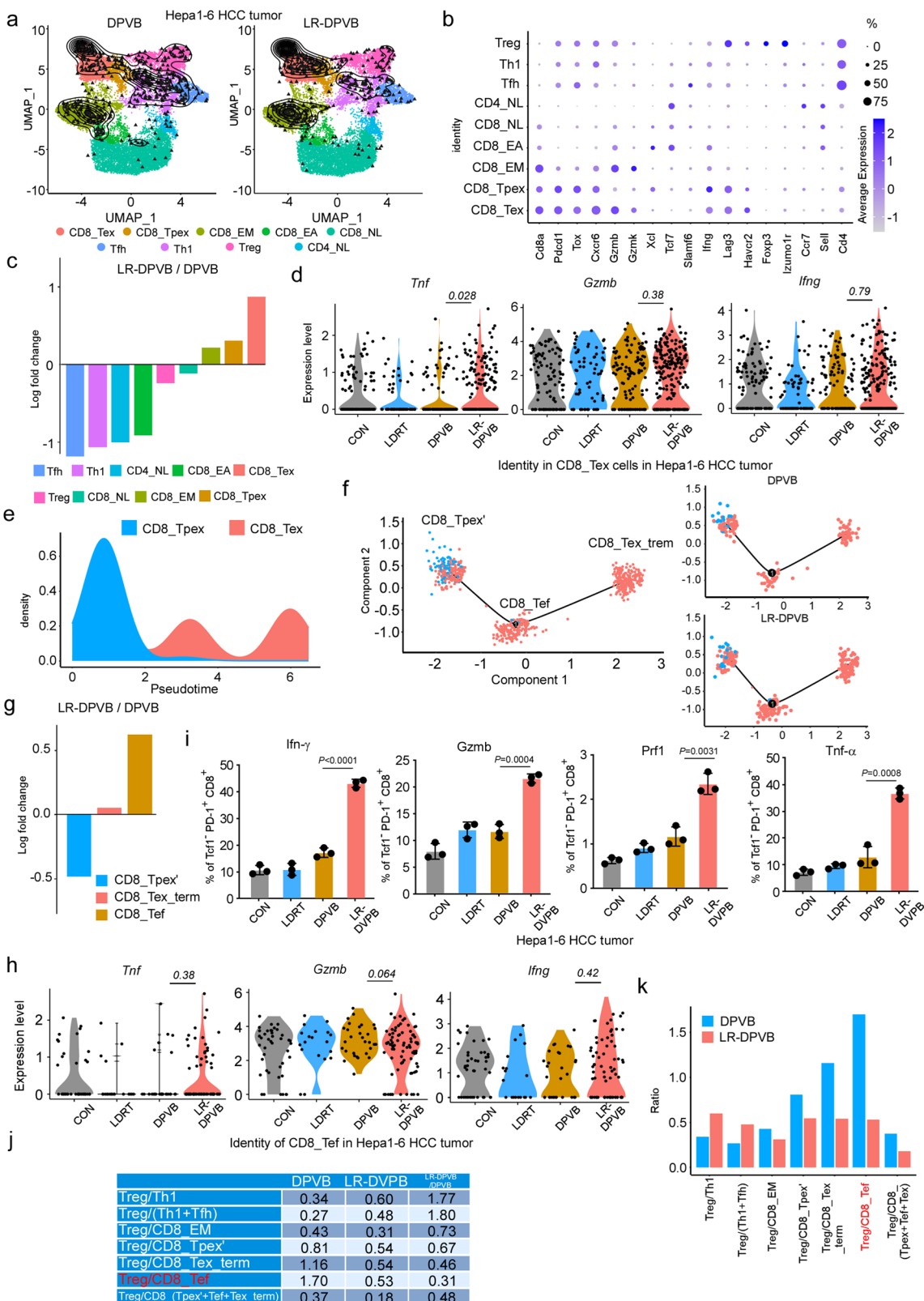

Supplementary Fig. 10a). Accordingly, flow cytometry analysis revealed a significant enrichment of TCF1[+] PD-1[+] CD8[+] T cells, which was reported to be the definition of progenitor exhausted CD8[+] T cells[25,38], in LDRT-treated HCC tumors (Fig. 4b, Supplementary Fig. 10b, d).

Given that the previous pseudotime analysis suggested that the tumor-rejecting CD8_Tex with effector function and cytolytic capacity

from the LR-DPVB-treated tumor evolved from CD8_Tpex (Fig. 3e, f, Supplementary Fig. 6c), we further explored whether intratumoral stem-like CD8[+] Tpex could enhance the antitumor effect of DPVB using the stem-like CD8[+] Tpex transfer mouse model in vivo (Fig. 4d). Since previous studies showed that TCF-1 and SLAMF6 were highly coexpressed in stem-like CD8[+] Tpex, while few terminal exhausted CD8[+] T expressed SLAMF6[39], we chose SLAMF6 as a cell-surface marker to

**Fig. 3 | LR-DPVB expands antitumor CD8[+] Tex exhibiting states of effector function and cytolytic capacity in Hepa1-6 tumors. a** UMAP plots of T cells scRNA-seq data (n = 3 per treatment, collected at the end of therapeutic cycle 2). Left and right: contour plots reveal cell density of indicated groups. Using the ProjecTILs method, T cells were subdivided into CD8_Tex, CD8_Tpex, CD8_effector memory (CD8_EM), CD8_ early activated (CD8_EA), CD8_ naïve like (CD8_NL), CD4_ naïve like (CD4_NL), Treg, T helper 1 (Th1) and T follicular helper (Tfh). **b** Hepa1-6 HCC tumors T cells expressing indicated genes across major cell subsets and their corresponding average expression. **c** Fold-change of T cell subsets of Hepa1-6 HCC tumors following LR-DPVB vs DPVB. **d** Violin plots representing the expression of various effected function and cytotoxicity markers in CD8_Tex cell subsets of Hepa1-6 HCC tumors with indicated treatment. **e** Density profiles show the distribution of CD8_Tex and CD8_Tpex of Hepa1-6 HCC tumors along the pseudotime trajectory. **f** Pseudotime trajectory analysis of CD8_Tpex and CD8_Tex clusters was performed by the Monocle tool. Left: Reference trajectory map of Hepa1-6 HCC tumor in all groups; right: pseudotime trajectory maps of indicated groups. **g** Fold-change of exhausted CD8_T cell subsets in Hepa1-6 HCC tumors following LR-DPVB vs DPVB. **h** Violin plots representing the expression of various effected function and cytotoxicity markers in CD8_Tef cell subsets of Hepa1-6 HCC tumors with indicated treatment. **i** Quantification of flow cytometry plots of % Prf1[+], % Gzmb[+], % Tnf-α[+] and % Ifn-γ[+] in TCF1[+] PD-1[+] CD8[+] T cells in Hepa1-6 HCC tumors of the indicated groups. n = 3 per group. P values were calculated using a two-sided unpaired Student's t test. **j, k** Table **j** and histogram **k** describes Treg: CD8 or CD4 ratios in Hepa1-6 HCC tumors of LR-DPVB and DPVB groups. Data shown as means ± SD derived from tumor mouse models (n = 3 mice/group). Source data are provided as a Source Data file.

isolate live stem-like CD8[+] Tpex for further investigation (Fig. 4c, d). Our results showed that the adoptive transfer of SLAMF6[+] PD-1[+] CD8[+] T cells, but not SLAMF6[-] PD-1[+] CD8[+] T cells, could promote DPVB-mediated tumor regression by enhancing the cytotoxic capacity of CD8[+] T cells (Fig. 4e–h). These data together indicated that stem-like CD8[+] Tpex supports DPVB-mediated antitumor effects.

Encouraged by these promising findings, we further verified the correlations between stem-like CD8[+] Tpex and the T + A therapeutic effect using patient-derived tumor fragments (PDTFs). Patient-derived tumor fragments (PDTFs), which preserve tumor microenvironment and architecture, are serve as a platform to dissect the early immunological response of human tumor tissue to ex vivo drug treatment[40]. The early immunological response of PDTFs is correlated with the clinical response of the corresponding patients. This method has been applied to test the early immunological response to several treatments, such as immune checkpoint blockade, therapeutic nanoparticles and anti-inflammatory drugs[40–42]. We selected 6 human primary HCC specimens for independent experiments. The resected HCC specimens were then divided into two pieces, one was for the flow cytometry detection of the infiltration degree of intratumoral stem-like CD8[+] Tpex, the other was for the PDTFs observation after T + A treatment (Fig. 4i). The infiltration degree of intratumoral stem-like CD8[+] Tpex was determined by dividing it with the median value of the ratio between TCF1[+] PD1[+] CD8[+] T cells and total CD8[+] T cells, as evaluated by flow cytometry in 6 HCC tissues. After 48 h ex vivo T + A treatment, we observed more intense tissue destruction in PDTFs derived from HCC tissues with higher stem-like CD8[+] Tpex infiltration than in those derived lower CD8[+] Tpex infiltrated HCC tissues (Fig. 4j). Moreover, there was a significant increase in apoptotic tumor cells in the PDTFs derived from stem-like CD8[+] Tpex-high HCC tissues after T + A treatment (Fig. 4j, k). These results suggested that the high accumulation of stem-like CD8[+] Tpex in tumors was correlated with the expanded efficacy of T + A therapy. In summary, by using PDTF platform, we demonstrated that higher infiltration of stem-like CD8[+] Tpex was associated with better early T + A treatment efficacy in HCC, which has great clinical value for guiding T + A treatment application and improving the efficacy of T + A treatment in HCC patients in the future.

**Intratumoral stem-like CD8[+] Tpex expanded by LDRT are recruited from the dLNs through the CXCL10/CXCR3 axis**
Next, we analyzed the CD8[+] T cell characteristics of the LDRT-treated tumors. Volcano Plot and heatmap of scRNA-seq data from Hepa1-6 tumors showed that both CD8[+] T cells and CD8_Tpex expressed elevated levels of activation molecules (e.g., CD8[+] T cells: *Cd28, Tnfrsf9, Tnfrs4, Icos, Il7r*; CD8_Tpex: *Id2, Icos*) after LDRT (Supplementary Fig. 11a, b). The enrichment analysis of DEGs between LDRT and CON groups in scRNA-seq data from DEN + CCl[4] tumors revealed a significant positive correlation in the pathway associated with T cell-mediated immune response activation, observed in both CD8_T cells

and CD8_Tpex (Supplementary Fig. 10e, f). Previous study indicated that intratumoral stem-like CD8[+] Tpex mainly accumulated via two ways, 1) tumor in-site differentiation and expansion; 2) migration from local dLNs[43]. To determine the origin of stem-like CD8[+] Tpex enhanced by LDRT, we first explored the proliferation capacity of CD8_Tpex after LDRT treatment. The cell cycle analysis of the scRNA-seq from Hepa1-6 tumors indicated that the sum of the percentage of S phase and G2/M phase, reflecting the cell proliferation capacity[44], was not enhanced by LDRT (Supplementary Fig. 11c). Flow cytometry analysis also showed that the percentage of Ki-67[+] cells among TCF-1[+] PD-1[+] CD8[+] T cells was not increased after LDRT treatment in the three HCC models (Supplementary Fig. 10c-d, 11d). Thus, we speculated that stem-like CD8[+] Tpex cells originated from immune niches beyond the tumor. The DEGs enrichment indicated that the TILs from the LDRT tumors expressed stronger leukocyte migratory and chemotaxis signatures (Fig. 5a). Further analysis of DEGs enrichment of CD8_Tpex showed that the LDRT group was significantly related to Rho protein signal transduction-associated pathways, which have been reported to closely related with cell migration[45] (Supplementary Fig. 11e). We next analyzed stem-like CD8[+] Tpex cells in hepatic draining lymph nodes (dLNs) of Hepa1-6 and DEN+CCl[4] model mice by flow cytometry. The results showed that TCF-1[+] PD-1[+] CD8[+] T cells were increased after LDRT treatment and the percentage of Ki-67[+] cells among TCF-1[+] PD-1[+] CD8[+] T cells was significantly increased in dLNs after LDRT treatment (Fig. 5b, c, Supplementary Fig. 9g). Furthermore, when we used FTY720 to block the peripheral drainage of lymphoid tissues in LDRT-treated mice, intratumoral TCF-1[+] PD-1[+] CD8[+] T cells were remarkably diminished, implying that immune cell migration from the dLNs played a critical role in the treatment of LDRT (Fig. 5d, e, Supplementary Fig. 10h, i).

Then, we explored the mechanisms underlying the drainage of stem-like CD8[+] Tpex cells from dLNs to the tumor. KEGG enrichment pathway analysis of RNA-seq data showed that the LDRT-treated Hepa1-6 tumors were significantly related to *the cytokine and cytokine receptor interaction pathway* (Fig. 5f). Notably, although numerous cytokine- and chemokine-encoded genes derived from LDRT-treated tumors were largely upregulated in this pathway, we observed a significant enhancement in the expression of the same family genes, *Cxcl9* and *Cxcl10* (Fig. 5g). Previous studies have demonstrated that the *Trp53[KO]/MYC[OE]* model exhibits reduced levels of chemokine Cxcl10 and a diminished enrichment of the progenitor exhausted CD8[+] T cell phenotype[29], making it an ideal model for validating the efficacy of LDRT treatment. The qPCR experiments showed the upregulation of *Cxcl10* in both LDRT-treated Hepa1-6 tumors and *Trp53[KO]/MYC[OE]* tumors (Supplementary Fig. 10j, 11f). We further investigated which subset of cells was primarily responsible for the upregulation of CXCL10 following LDRT. As previous study has demonstrated, CXCL10 is primarily produced by antigen-presenting cells (APCs), including dendritic cells and macrophages, as well as tumor cells[46]. RT-qPCR and ELISA assays showed that the expression level of CXCL10 in LDRT-

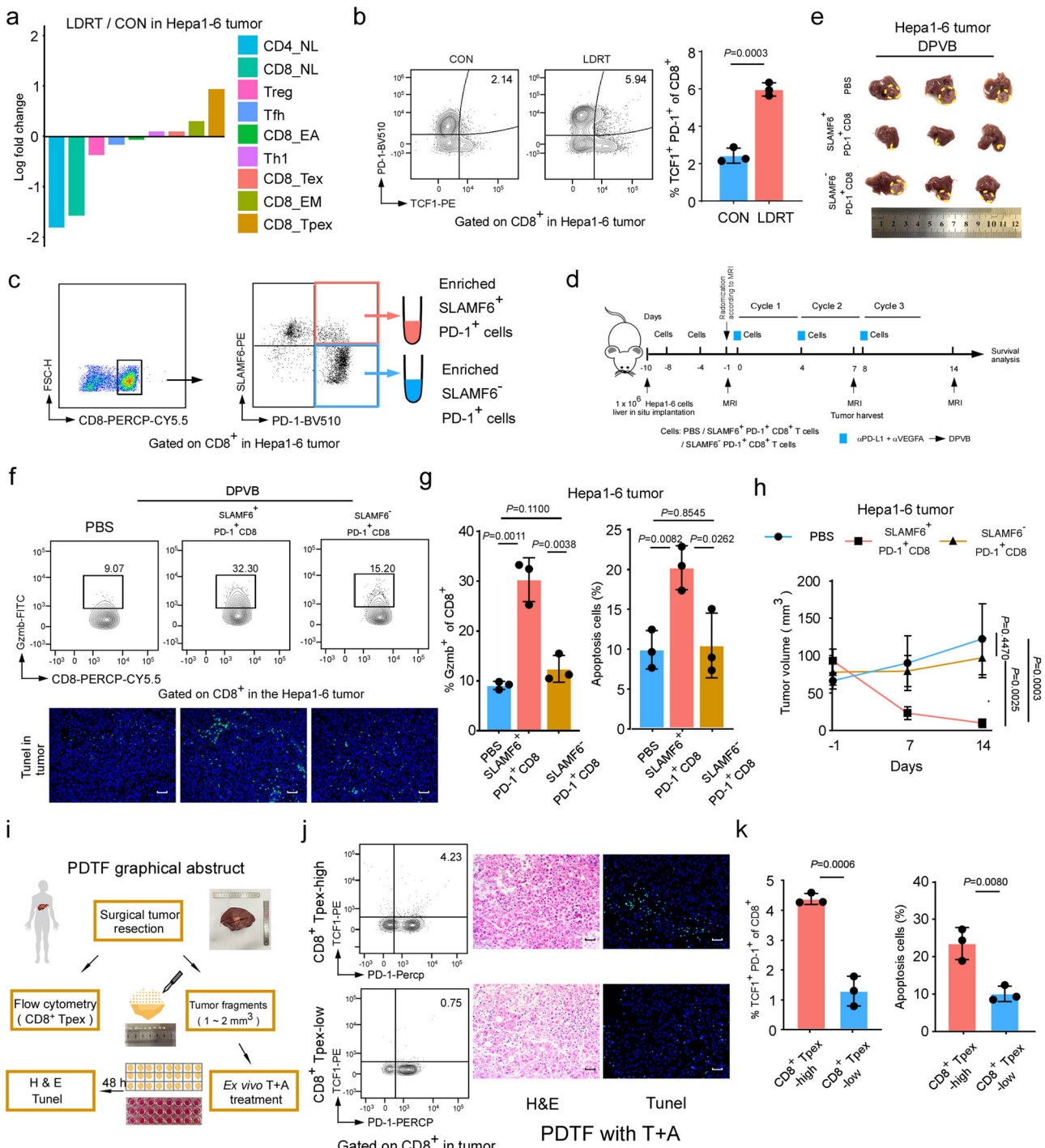

**Fig. 4 | The stem-like CD8⁺ Tpex enriched by LDRT expanded the antitumor effect of DPVB. a** Fold-change of T cell subsets following LDRT vs CON in Hepa1-6 tumors. *n* = 3 mice/group. **b** Representative flow cytometry plots (left panel) and quantification (right panel) of % TCF1⁺ PD-1⁺ in CD8⁺ T cells of the indicated groups. *n* = 3 samples/group. **c, d** Illustration of the in vivo adoptive transfer experiment, as described above in methods. *n* = 3 mice/group. **e** Representative image of Hepa1-6 liver orthotopic tumors at the end of the second therapeutic cycles (circled by yellow lines). *n* = 3 mice/group. **f** Representative flow cytometry plots of % Gzmb⁺ in CD8⁺ T cells (upper panel) and TUNEL assay (lower panel) of the indicated groups. *n* = 3 samples/group. Scale bars: 100 μm. **g** Quantification of % Gzmb⁺ in CD8⁺ T cells (left panel) and (%) apoptosis cells (right panel) of the indicated groups.

*n* = 3 samples/group. **h** Tumor growth curves of Hepa1-6 bearing mice of the indicated days. *n* = 5 mice/group. *P* values were calculated using One-way repeated-measures ANOVA test. Data shown as means ± SD. **i** Graphical abstract of PDTF collection, culture and treatment strategy. **j** Flow cytometry assays (left panel) and histopathological analysis (H&E staining and TUNEL assay, right panel) of the PDTFs after 48 h of ex vivo T + A therapy. *n* = 3 samples/group. Scale bars: 100 μm. **k** Quantification of % TCF1⁺ PD-1⁺ CD8⁺ in CD8⁺ T cells (left panel) and (%) apoptosis cells (right panel) of the indicated groups (*n* = 3 samples/group). *P* values of **b**, **g** and **k** were calculated using a two-sided unpaired Student's *t* test. Data of **b** and **g** shown as means ± SD derived from tumor mouse models (*n* = 3 mice/group). Source data are provided as a Source Data file.

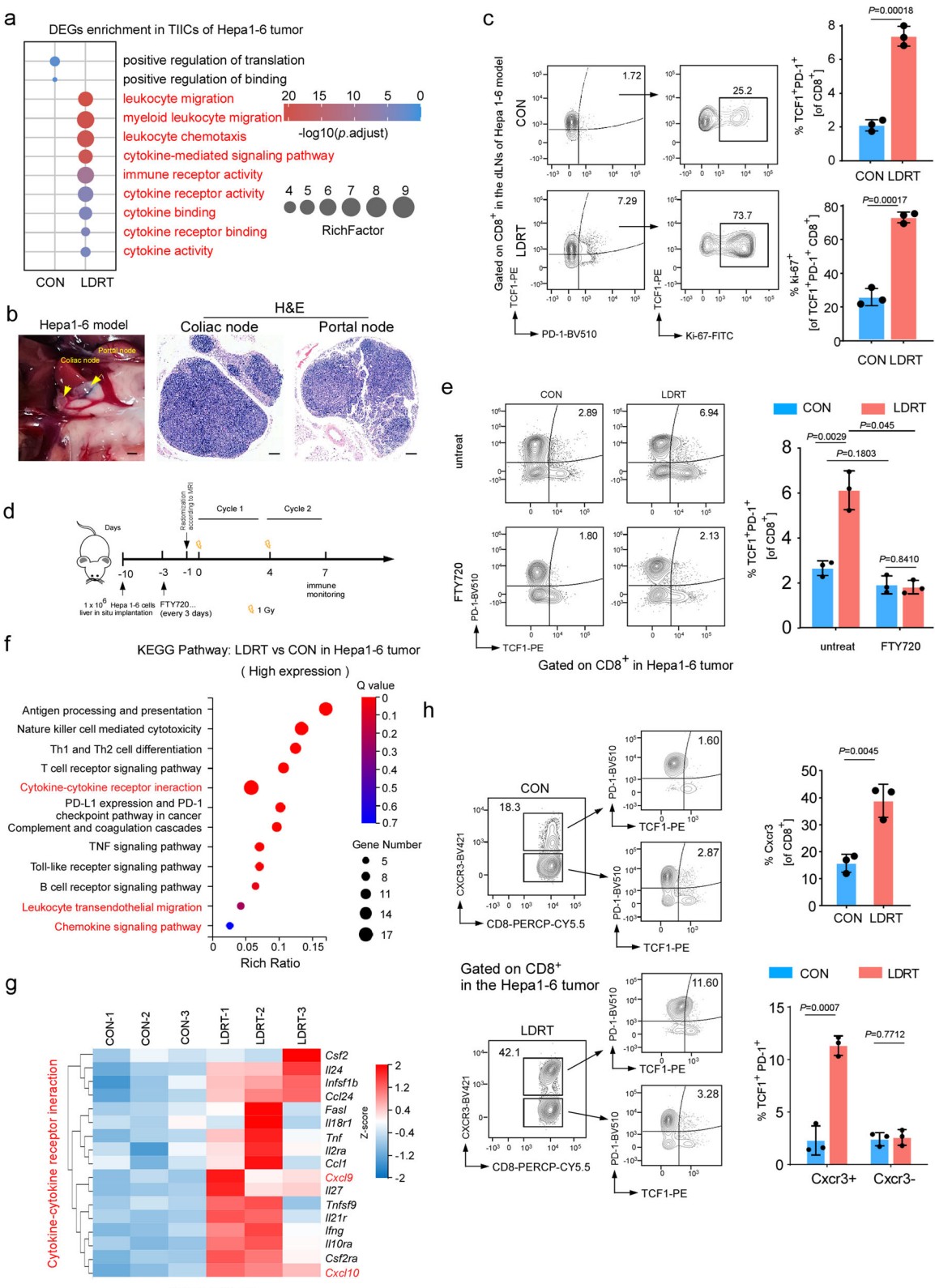

treated Hepa1-6 cells and the supernatant of the cells was not significantly upregulated (Supplementary Fig. 11g). While, the in vivo experiments showed the tumor-associated macrophages (TAMs, F4/80⁺ CD11b⁺ CD45⁺ L/D⁻) and DCs (CD11c⁺ CD45⁺ L/D⁻) cells sorted from TIICs had significantly higher *CXCL10* expression in the LDRT group than those in the CON group (Supplementary Fig. 10k, 11h). Furthermore, the in vitro co-culture assays revealed a significant upregulation

of CXCL10 in bone marrow-dendritic cells (BMDCs) and bone marrow-derived macrophages (BMDMs) when co-cultured with LDRT-treated tumor cells, compared to those co-cultured with untreated tumor cells (Supplementary Fig. 11i). These results suggested that myeloid APCs are the primary source of upregulated CXCL10 following LDRT. Moreover, CXCR3, which is the corresponding receptor of Cxcl9 and Cxcl10, is expressed by stem-like CD8⁺ Tpex cells and functions in CD8⁺

**Fig. 5 | Intratumoral infiltration of stem-like CD8⁺ Tpex from dLNs after LDRT through the CXCL10/CXCR3 axis. a** Bubble chart of DEG pathway enrichment of TILs in LDRT and CON groups. **b** Anatomical view of mouse liver dLNs (left panel, Scale bars: 1 mm) and H&E staining (right panel, Scale bars: 100 μm) of the dLNs sections (*n* = 3 mice/group). **c** Representative flow cytometry plots (left panel) and quantification (right panel) of % TCF1⁺ PD-1⁺ in CD8⁺ T cells and % Ki-67 in TCF1⁺ PD-1⁺ CD8⁺ T cells in dLNs of the indicated groups. **d** Schematic illustration of peripheral drainage of lymphoid tissues blockade experiment in Hepa1-6 bearing LDRT treated mice. **e** Representative flow cytometry plots (left panel) and quantification

(right panel) of % TCF1⁺ PD-1⁺ in CD8⁺ T cells in the tumors of the indicated groups. **f** Bubble chart of KEGG enrichment pathways of RNA-seq in LDRT vs CON groups. *n* = 3 mice/group. **g** Heatmap of the DEGs in the *cytokine-cytokine receptor interaction* pathway. *n* = 3 mice/group. **h** Representative flow cytometry plots (left panel) and quantification (right panel) of % CXCR3⁺ in CD8⁺ T cells, % TCF1⁺ PD-1⁺ in CXCR3⁺ CD8⁺ T cells and CXCR3⁻ CD8⁺ T cells in the Hepa1-6 tumors of the indicated groups. Data of **c**, **e** and **h** shown as means ± SD derived from tumor mouse models (*n* = 3 mice/group). *P* values of **c**, **e** and **h** were calculated using a two-sided unpaired Student's *t* test. Source data are provided as a Source Data file.

T-cell migration and positioning within tumors, as reported by prior studies[43,47–49], was significantly upregulated in CD8⁺ T cells after LDRT treatment of tumor (Fig. 5h, Supplementary Fig. 10l). Notably, flow cytometry results showed that LDRT only increased the stem-like CD8⁺ Tpex in CXCR3⁺ CD8⁺ T cells but did not significantly change the percentage of stem-like CD8⁺ Tpex in CXCR3⁻ CD8⁺ T cells in either Hepa1-6 tumors or *Trp53^{KO}/MYC^{OE}* tumors (Fig. 5h, Supplementary Fig. 10l). To clarify whether the migration of stem-like CD8⁺ Tpex cells after LDRT is mediated by the chemokine receptor CXCR3, we further treated mice with LDRT along with anti-CXCR3 (Supplementary Fig. 10m, 11j). Consistent with our expectation, anti-CXCR3 substantially weakened LDRT-induced accumulation of stem-like CD8⁺ Tpex cells in the tumors (Supplementary Fig. 10n, 11j). Overall, these data supported that the stem-like CD8⁺ Tpex enhanced by LDRT were mainly recruited from the dLNs into the tumor through the CXCL10/CXCR3 axis.

### Stem-like CD8⁺ Tpex cells and Treg/Tef ratio among T + A-treated HCC patients and all HCC patients

To further explore the clinical predictive significance of stem-like CD8⁺ Tpex in T + A-treated HCC patients, we retrospectively analyzed the relationship between the accumulation of TCF-1⁺ PD-1⁺ CD8⁺ T cells in biopsy tissues and T + A response of HCC patients (the T + A response was evaluated by mRECIST, Supplementary Table 1, Fig. 6a). As expected, HCC patients with a better T + A treatment response (achieved PR/CR) had higher infiltration of TCF-1⁺ PD-1⁺ CD8⁺ T cells in the biopsy HCC tissues, while HCC patients without a response to T + A treatment showed lower accumulation of this cell subset in the biopsy HCC tissues (Fig. 6a, b). Additionally, Kaplan–Meier Plotter analysis (https://kmplot.com/analysis/) showed that higher coexpression of TCF-1, PD-1 and CD8 was significantly related to better survival of HCC patients, while coexpression of TCF-1, PD-1 or expression of TCF-1 alone did not show a correlation with the prognosis of HCC patients (Supplementary Fig. 12a). Together, these data suggested that high levels of stem-like CD8⁺ Tpex cells could be considered a signature to identify HCC patients who might benefit from T + A combination therapy.

As our scRNA-seq analysis previously showed that Treg/CD8_Tef may be a potential curative effect index after LR-DPVB treatment (Fig. 3j-k, Supplementary Fig. 6l), we next investigated the clinical relevance of Treg/CD8_Tef (namely, Treg/Tef) and the outcome of T + A-treated HCC. We examined the Treg/Tef ratio in PBMCs of HCC patients after T + A. Linear regression analysis revealed that the percentage change from baseline in diameters of target lesions positively correlated with the Treg/Tef ratio (Supplementary Table 2, *p* = 0.0061, $R^2$ = 0.68) but not Treg cells (*p* = 0.054, $R^2$ = 0.43) or Tef percentage (*p* = 0.063, $R^2$ = 0.41) (Fig. 6c, Supplementary Fig. 12b), directly linking the Treg/Tef ratio to the tumor control of T + A therapy. Then, we analyzed the Treg/Tef ratio in both surgically removed HCC tissues and post-surgery PBMCs to investigate the correlation between Treg/Tef ratio and the recurrence condition within 6 months post-surgery in 20 HCC patients (Supplementary Table 3, Fig. 6d). Flow cytometry showed that the infiltration of Treg and Tef were not significantly correlated with HCC recurrence at 6 months in either the TILs of

surgically removed HCC tissues or PBMCs (Supplementary Fig. 12c). However, the TILs of removed tissues and post-surgery PBMCs from the 6-month relapse HCC patients showed a notably higher Treg/Tef ratio, while the HCC patients without relapse within 6 months exhibited a lower Treg/Tef ratio (Fig. 6d). These results indicated that the Treg/Tef ratio in HCC tissues and postsurgical blood was positively correlated with the short-term recurrence of HCC after tumor resection. Finally, we explored the clinical long-term prognostic relevance of the Treg/Tef ratio in 120 paraffin-embedded HCC samples by immunofluorescence staining (Supplementary Table 4, Fig. 6e). The Treg/Tef ratio was positively and significantly associated with tumor number (*p* = 0.032), 5-year vital status (*p* < 0.0001), and long-term tumor recurrence within 5 years (*p* = 0.003) (Supplementary Table 5). Importantly, Kaplan–Meier survival curves and log-rank tests revealed that patients with high Treg/Tef level had shorter 5-year overall survival (OS, hazard ratio (HR) = 4.859, *p* < 0.0001) and relapse-free survival (RFS, HR = 2.249, *p* = 0.0001) in HCC (Fig. 6f, g). Furthermore, multivariate cox regression analysis indicated that a high Treg/Tef level (*p* = 0.001) and vascular invasion (*p* = 0.016) were independent prognostic factors for 5-year OS in HCC (Supplementary Fig. 12d left panel, Supplementary Table 6), and a high Treg/Tef level (*p* = 0.003) and low tumor differentiation level (*p* = 0.031) were independent prognostic factors for 5-year RFS in HCC (Supplementary Fig. 12d right panel, Supplementary Table 7). These results suggested that the Treg/Tef ratio was preferentially upregulated in relapsed HCC patients and that a high Treg/Tef level correlated with poor HCC prognosis, which indicated that Treg/Tef could be a prognostic biomarker for primary HCC patients.

## Discussion

Results from the current phase III clinical trial IMbrave150 emphasized the breakthrough of T + A therapy in unresectable HCC[4]. Despite the impressive objective response rate (ORR) and prognostic data, a low response in a majority of HCC patients and the lack of effective biomarkers highlight the urgent need to seek an optimized combination therapy regime. Here, we reported that LR-DPVB, which combined LDRT with DPVB, led to tumor regression in multiple HCC mouse models. At the cellular level, LR-DPVB exhibited superior CD8⁺ T cell-mediated antitumor efficacy, and scRNA-seq data suggested that the treatment efficacy mainly relied upon mobilizing the effector function and cytolytic capacity of CD8⁺ Tex. Mechanistically, LDRT sensitized DPVB by enlarging intratumoral stem-like CD8⁺ Tpex, which were recruited from the draining lymph nodes (dLNs) into the tumor through the CXCL10/CXCR3 axis. Our study also showed that high levels of stem-like CD8⁺ Tpex could be a signature to identify HCC patients who might benefit from T + A combination therapy and the Treg/Tef ratio might be a prognostic biomarker for HCC patients with T + A combination therapy and even for all HCC patients (Fig. 7).

Recently, increasing evidence has shown the role of RT in enhancing antitumor immunity. Preclinical data demonstrated that the combination of RT and ICB showed therapeutic synergy and superior tumor control efficiency[8,10,11,50]. In addition to the encouraging preclinical evidence, there were also reports of some cases with noteworthy clinical data. Jiang CL et al. revealed that the ORR of patients

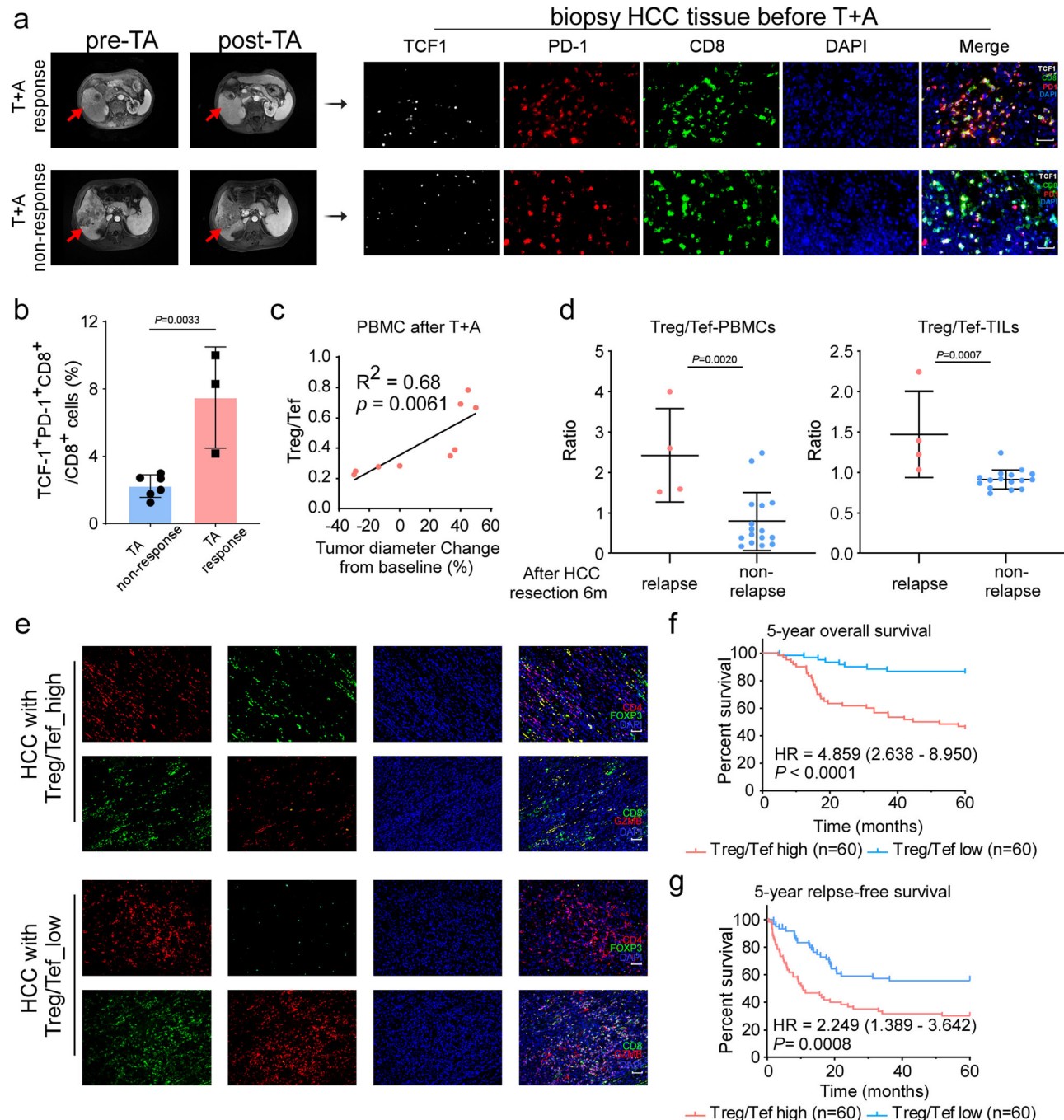

**Fig. 6 | The relevance of stem-like CD8⁺ Tpex infiltration in the response T + A-treated HCC patients and the Treg/Tef ratio in the prognosis of T + A-treated HCC patients and all HCC patients. a** (Left panel) Representative MRI of pre and post T + A in T + A response and non-response HCC. (Right panel) Representative mIHC image of stem-like CD8⁺ Tpex in T + A response and nonresponse HCC. Scale bars: 50 μm. **b** Quantification of mIHC of the TCF1⁺ PD-1⁺ CD8⁺ cells of CD8⁺ cells (%) in the indicated groups. T + A non-response: $n = 6$. T + A response: $n = 3$. **c** Graphs depict correlation between the Treg/Tef ratio and tumor diameter change from baseline (%). $p$ value and $R^2$ value were calculated using linear regression analysis

($n = 9$). **d** Quantification of flow cytometry plots of the Treg/Tef ratio in the PBMCs and TILs of the indicated groups. Relapse: $n = 4$. non-relapse: $n = 16$. **e** Representative IF image of Treg/Tef in HCC. Scale bars: 100 μm. **f, g** Kaplan–Meier curve of 5-year OS (upper panel) and 5-year RFS (lower panel) for HCC patients with low Treg/Tef ratio (Treg/Tef -low; $n = 60$) versus those with high Treg/Tef ratio (Treg/Tef - high; $n = 60$). $P$ values of **b** and **d** were calculated using a two-sided unpaired Student's $t$ test. $P$ values of **f**, **g** were determined by two-sided log-rank test (Mantel–Cox). Data of **b** and **d** shown as means ± SD. Source data are provided as a Source Data file.

who underwent anti-PD-1 treatment following stereotactic body radiation therapy (SBRT) was 100% in 5 patients with unresectable HCC. The PACIFIC trial and KEYNOTE-001 trial demonstrated that ICIs improved the survival of patients with phase III nonsmall cell lung cancer who previously received radiotherapy[51,52]. However, different from other cancers, most patients with HCC have a liver disease

background, manifested as poor liver function. Numerous investigations have found that individuals with impaired liver function have a higher risk of radiation-induced liver toxicity[11,15]. In addition, conventional radiotherapy may also cause gastrointestinal perforation or bleeding, especially in patients with gastrointestinal ulcers and other digestive diseases[53,54]. Thus, exploring the optimal dose and regime of

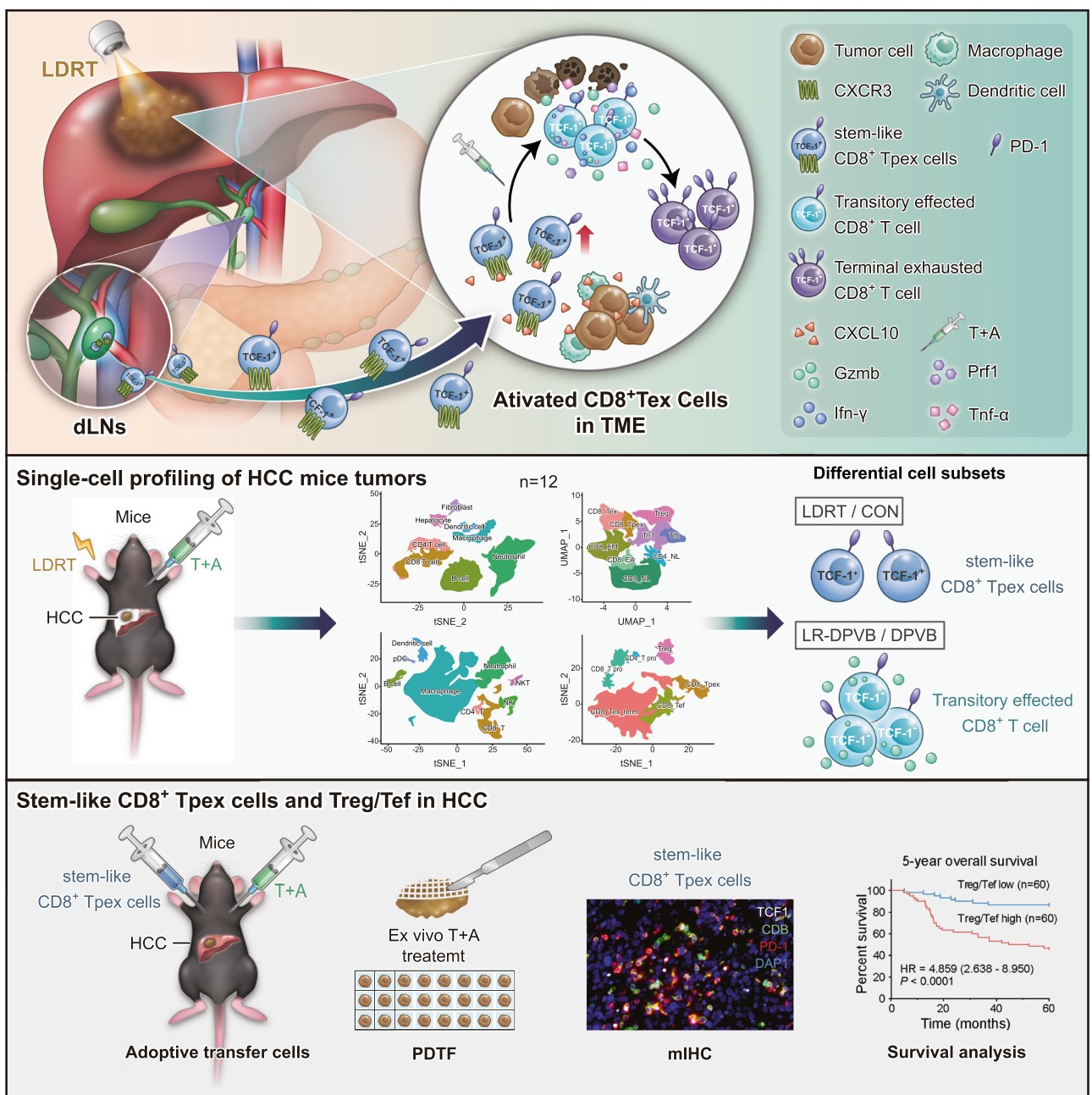

**Fig. 7 | The overview of the mechanism and experimental model investigating the synergistic antitumor effect of combined LDRT with T + A.** The first panel: Schematic diagram showing the mechanism antitumor effect of HCC by LDRT combined with T + A activate the intratumoral CD8[+] Tex. **The second panel:** The scRNA-seq of the HCC mouse model with different treatments. **The third panel:** Relationship of stem-like CD8[+] Tpex and Treg/Tef with the efficacy of T + A treatment and the prognosis of HCC.

radiotherapy to improve the efficacy of immunotherapy in HCC without increasing adverse reactions is crucial. An early study in the NSG mouse model of melanoma demonstrated that local low-dose gamma irradiation could modify the TME and enhance immunotherapeutic benefits[19]. Barsoumian et al. demonstrated that high-dose irradiation of the primary lesion combined with low-dose radiation therapy (LDRT) of the metastatic lesion in mice with spontaneous lung tumors could increase the efficacy of ICB by mobilizing innate and adaptive immunity and enhancing the infiltration and function of effector immune cells in secondary tumors, thereby effectively controlling metastatic tumors[18]. In ovarian cancer, George Coukos and colleagues discovered that LDRT boosted peritoneal lymphocyte infiltration in mice with orthotopic intraperitoneal ovarian cancer,

reversed the loss of immune cells in the tumor microenvironment (TME), and enhanced the efficacy of combination immunotherapy. This study also translated the preclinical findings into a phase I clinical trial of LDRT in combination with low-dose cyclophosphamide and ICBs in tumor patients with a desertification tumor immune microenvironment, which supported the rationale for LDRT and combinatorial immunotherapy to effectively treat tumors with low lymphocyte cell infiltration[17]. Collectively, LDRT is an attractive strategy to reshape the TME to promote tumor immunotherapy in a variety of tumors. Since HCC is a solid tumor with numerous pathological barriers, it is challenging for lymphocytes to enter and penetrate HCC tumor tissue. In addition, rapidly growing tumor cells release immunosuppressive factors that result in HCC forming an immunosuppressive TME, which

greatly limits the efficacy of immunotherapy[55–58]. The development of rational combinatorial regimes that can boost efficacy by overcoming immune resistance is thus of utmost importance. The results of the IMbrave150 trial also highlighted the therapeutic potential of combining T + A with TME modulators[4]. Thus, we explored the efficacy and safety of LDRT in combination with T + A of HCC in the various preclinical models. In this study, the HCC mice treated with LR-DPVB exhibited therapeutic benefits without inducing substantial hepatotoxicity compared to those treated with DPVB alone in three mice HCC models. The scRNA-seq, flow cytometry and IHC analyses all showed obvious LR-DPVB-induced accumulation of CD8+ T cells, whereas CD4+ T cells were not significantly changed. Moreover, using the anti-CD8 antibody depleted of CD8+ T cells in animals with LR-DPVB weakened tumor regression, while the anti-CD4 antibody could not. These results proved the curative effect of LR-DPVB on HCC in preclinical models through CD8+ T cell-mediated antitumor immunity.

Previous studies continuously revealed the unique impact of LDRT on the tumor immune microenvironment. Klug, F. et al. showed that LDRT redifferentiated iNOS+ M1-type tumor-associated macrophages (TAMs), attracted cytotoxic T cells into tumors, and finally induced tumor cell death via iNOS[19]. For lung adenocarcinoma, it was reported that LDRT could boost the antitumor efficacy of ICBs by promoting the polarization of M1-type TAMs, increasing the infiltration of natural killer cells (NKs), and decreasing TGF-β levels in both preclinical and clinical investigations[18]. A study on ovarian cancer demonstrated that LDRT enhanced T cell infiltration and allowed IFN-γ-dependent responsiveness to combinatorial immunotherapy[17]. Although the immune-activation effects of LDRT were reported, the cellular mechanisms under LDRT in combination with T + A need to be further elucidated. Our scRNA-seq analysis and multiple experiments revealed that CD8+ T cells, but not CD4+ T cells, were indispensable to the therapeutic effect of LR-DPVB since their absence led to therapeutic failure. Notably, we further identified tumor-rejecting CD8+ T cells with both exhausted and cytolytic features following LR-DPVB in three HCC mice models. DEGs enrichment pathway analysis in CD8+ T cells also indicated that T cell differentiation and T cell receptor signaling activation were both significantly correlated with tumors following LR-DPVB. Pseudotime analysis showed that CD8_Tex was mainly differentiated from CD8_Tpex through middle-status transient effected CD8_Tef. Importantly, we found an intermediate transition cell population with specifically high expression of polyfunctional effector properties and cytolytic capacities in this evolutionary trajectory, which was consistent with previous studies showing that intratumoral stem-like CD8+ Tpex differentiated into temporary effector CD8+ T cells in response to vaccination or ICBs to promote tumor control[25]. The scRNA-seq analysis of the cell subsets of the T cells revealed that the most abundant population in the tumors treated with LR-DPVB was CD8_Tef, with canonical exhausted features, exhibiting effector functions and cytolytic capacities. Thus, the data indicated that the transient effector CD8+ T cells from the stem-like CD8+ Tpex played a critical role in the efficacy of the LR-DPVB combination. In addition, it should be noted that our study only focused on the sensitization effect of LDRT on the DPVB, which was considered a complete regime. There are still some limitations in clarifying the mechanism of anti-VEGFA treatment alone. Previous studies have shown that VEGFA directly inhibits T cell proliferation and cytotoxic activity by combining with VEGFR-2[59]. VEGFA/VEGFR also plays an important role in T cell exhaustion, especially in tumor-induced T cell exhaustion. VEGF-A increases PD-1 expression on CD8+ T cells and the coexpression of other immune checkpoints, such as CTLA-4, Tim-3, and Lag-3[60]. Anti-VEGFA treatment can enhance the antitumor effect of CD8+ T cells[61]. Thus, it is critical to further explore the mechanism of anti-VEGFA treatment in LR-DPVB in the future to clarify the efficacy of combination therapy.

It is a challenge to coordinate the concept of terminally differentiated T cells with the proliferation of intratumoral CD8+ T cells in response to PD-1 inhibition. Intratumoral T cell terminal exhaustion is a major barrier to eliciting an immunological response against malignancies[25,62]. There is a precursor subset of intratumoral exhausted CD8+ T cells (CD8+ Tpex) with less differentiation and coexpression of the inhibitory receptor PD-1 and the transcription factor TCF1 (encoded by *Tcf7*) that sustain the immune response in chronic infection, and tumor progression presents a link between T cell memory and exhaustion. These intratumoral TCF1+ PD-1+ CD8+ T cells display stem-like properties with augmentation, regeneration, and differentiation capacity and control the response to tumor immunotherapy[25]. Recent research aimed at enhancing the efficacy of ICB-based therapies has placed a greater emphasis on the control of the CD8+ Tpex stem-like cell pool. It is believed that the intratumoral niche is responsible for the maintenance and development of stem-like CD8+ Tpex[26], whereas tumor dLNs serve as the reservoir for ongoing antitumor immune responses[63]. In addition, a previous study showed that RT can significantly increase the enrichment of stem-like CD8+ Tpex cells through cGAS/STING signaling activation[64]. In our study, we found that LDRT promotes the intratumoral accumulation of stem-like CD8+ Tpex cells, which can further enhance the antitumor effects of DPVB. Using peripheral drainage blockade, we confirmed that the intratumoral stem-like CD8+ Tpex promoted by LDRT are derived from dLNs. Analysis of the DEGs and enrichment pathways identified by scRNA-seq and RNA-seq between the LDRT and CON groups indicated that the CXCL10/CXCR3 axis plays a critical role in LDRT-induced stem-like CD8+ Tpex. The in vivo and in vitro assays suggest that myeloid APCs are the primary source of upregulated CXCL10 following LDRT. This, in turn, promotes the recruitment of CXCR3+ CD8+ Tpex and enhances the therapeutic efficacy of DPVB. These findings are consistent with previous studies indicating that macrophage-derived CXCL10 can recruit stem-like TCF7+ CD8+ T cells, which predict effective tumor immunity and immune checkpoint blockade efficacy[44,65]. Flow cytometry and CXCR3 blockade further highlighted the importance of CXCR3 in the recruitment of stem-like CD8+ Tpex. Furthermore, recently published findings have shown that CXCR3 plays a dominant role in the recruitment and trafficking of CD8+ T cells, and the expression of the intratumoral CXCL10/CXCR3 axis has been considered to positively correlate with the antitumor response mediated by ICBs[47,65–67]. Collectively, these data indicated that LDRT induces CXCL10/CXCR3-mediated recruitment of stem-like CD8+ Tpex cells from dLNs. Although the stem-like CD8+ Tpex recruited by LDRT were mostly from the dLNs via the CXCL10/CXCR3 axis, we cannot completely rule out the possibility of other mechanisms that maintain stem-like CD8+ Tpex in the intratumor niche. A previous study showed that CXCR3 is a modulator of intratumoral CD8+ T cell infiltration[68]. Moreover, it has been reported that stem-like CD8+ Tpex reside in dense antigen-presenting cell (APC) niches within the tumor, which positively correlates with the response to ICBs[26]. Duraiswamy, J et al. showed that myeloid APC niches maintain antitumor T cells and license ICBs through CD28 costimulation[69]. The results of scRNA-seq analysis in our study showed that the expression of CD28 in CD8+ T cells of LDRT-treated tumors was significantly increased, suggesting that APCs may be involved in the recruitment and activation of CD8+ T cells mediated by LDRT. Last but not least, scRNA-seq analysis revealed a significant increase in the proportion of cDC1 infiltration subsequent to LDRT. Pre-clinical models have consistently demonstrated the essential role of cDC1 in T cell-mediated tumor regression and therapeutic responses to immune checkpoint blockade (ICB)[35–37]. Moreover, Jan P. Böttcher et al. recently reported that cDC1-CD8+ T cell clusters serve as niches for activation of stem-like CD8+ Tpex cells[70]. Therefore, further investigation into the interaction between APCs and CD8+ Tpex cells subsequent to LDRT holds promise.

Biomarkers of T + A combinatorial therapy remain largely unknown. The principal impediment to enhancing the T + A therapy response is the absence of adequate markers to identify potential treatment populations and predict treatment outcomes. In this study, stem-like CD8$^+$ Tpex cells were enhanced by LDRT, further sensitizing DPVB therapy, which hinted at the relationship between the infiltration of stem-like CD8$^+$ Tpex cells and the response of the T + A. Importantly, as a clinical proof of concept, we validated the relationship between stem-like CD8$^+$ Tpex infiltration and the efficacy of T + A treatment using PDTF platforms, indicating the potential clinical translation ability of stem-like CD8$^+$ Tpex. Multiplex immunohistochemistry (mIHC) showed that HCC patients with a T + A response presented higher infiltration of stem-like CD8$^+$ Tpex in tumor tissue, while the T + A nonresponse indicated the opposite, which suggested that the intratumoral stem-like CD8$^+$ Tpex infiltration may be a predictor to distinguish patients who are appropriate for T + A treatment. The patients with high infiltration of stem-like CD8$^+$ Tpex received T + A therapy, while those with low infiltration of stem-like CD8$^+$ Tpex could undergo additional LDRT based on T + A. Moreover, considering the emerging development of adoptive cell therapy, using the convenient adoptive stem-like CD8$^+$ Tpex to replace LDRT for sensitizing T + A therapy in patients with intratumoral low infiltration stem-like CD8$^+$ Tpex will be a promising strategy in the future. Our study also pointed out the significance of the Treg/Tef ratio in predicting the outcome of T + A-treated patients, which is consistent with a previous study showing that a reduced clinical benefit of T + A was associated with high Treg/Tef ratio[71]. While this previous study only reported that the therapeutic outcome of T + A might be affected by Treg/Tef ratio in the gene expression level, our study further verified the prognostic value of Treg/Tef lymphocytes infiltration ratio in T + A-treated HCC patients and even all HCC patients, which is a supplement and extension of previous research. In addition, the Treg/Tef ratio was only discovered in a limited proportion of HCC patients, and larger studies are required to determine the significance of this ratio in HCC patients. Despite a few limitations, this work demonstrates that the Treg/Tef ratio should be studied further as a possible biomarker of the antitumor effects of the T + A regime.

Collectively, our study proposes a therapeutic strategy by combining LDRT with T + A to improve the antitumor immunity of HCC. These findings shed fresh light on the process of sensitization by T + A treatment, highlighting the enhancement of CD8$^+$ Tpex stem cells. Finally, the stem-like CD8$^+$ Tpex and Treg/Tef might serve as biomarkers for the selection of patients with HCC before T + A combination therapy and be effective indicators of the outcome of this treatment.

## Methods

### Ethical statement
This research complies with all relevant ethical regulations. The animal research in this study were approved by the Institutional Animal Care and Use Committee of the South China Agricultural University (No. 2022d113). The maximal tumor size permitted by the committee is 2000 mm$^3$, which was not exceeded in the in vivo procedures. Clinical specimens were used for research purposes with prior patient consent and ethical approval from the institutional research ethics committees of Third Affiliated Hospital of Sun Yat-sen University (ethics approval number: II2023-068-01) and the Ethics Committee of Eastern Hepatobiliary Surgery Hospital, Naval Medical University (ethics approval number: EHBHKY2021-K-017).

### Mice and cell line
Two-week-old, six-week-old male and eight-week-old male C57BL/6J mice were purchased from the Model Animal Research Center of Nanjing University (China). All mice were raised under specific pathogen-free (SPF) conditions with restrict 12 hours day/night cycle at a temperature 18-22 °C and humidity 50-60%.

The Hepa1-6 cell line was obtained from ATCC and tested negative for mycoplasma (Cat# CL-0105). Cells were cultured with high-glucose Dulbecco's Modified Eagle's Medium (DMEM) (Gibco) supplemented with 10% fetal bovine serum (FBS) (PAN-Biotech) at 37 °C in 5% CO$_2$ condition.

### Liver orthotopic HCC model animal studies
Liver orthotopic Hepa1-6 HCC models were established at day −10[72]. Anesthesia was induced by intraperitoneal injection of 10% chloral hydrate (0.07 ml/10 g). The mouse was sterilized with iodophor and positioned supine, cutting through the skin, peritoneum, and muscle layers exposed the right liver lobe. 50 µL Hepa1-6 cells (1×10$^6$ cells) were injected into the liver lobe. Tumor-bearing mice were blindly randomized into treatment groups based on MRI at day −1. On day 0, the mice were start treated with PBS (CON), anti-mouse PD-L1 (BioX-Cell Cat# BE0101, 5 mg/kg per mouse) plus anti-mouse VEGFA (Biolegend Cat# 512810, 50 µg per mouse) (DPVB), low-dose radiation therapy (1 Gy locally delivered to the liver, LDRT), LDRT + DPVB (LR-DPVB). 4 days a cycle, 3 cycles totally. After 2 therapeutic cycles, the tumor tissues and liver tissues were harvested for immunology analysis and safety evaluation. MRI (M3 small animal magnetic resonance imaging apparatus) was performed weekly from the beginning of the treatment to observe tumor size. Complete response (CR: complete disappearance of tumor), partial response (PR: at least 30% decrease from baseline), stable disease (SD: between PR and PD), progressive disease (PD: at least 20% increase from baseline).

### Diethylnitrosamine and repeated carbon tetrachloride-induced hepatocarcinogen model (DEN + CCl$_4$ HCC model) animal studies
The male mice, aged two weeks, were administered DEN (25 mg/kg) and intraperitoneally injected with 20% CCl$_4$ in olive oil twice a week at a dosage of 1ul/g when they reached four weeks old. Subsequently, the mice were treated with anti-mouse PD-L1 (BioXCell Cat# BE0101, 5 mg/kg per mouse) combined with anti-mouse VEGFA (Biolegend Cat# 512810, 50 mg per mouse) (DPVB), and low-dose radiation therapy (LDRT) consisting of locally delivered liver irradiation at a dose of 1 Gy every four days for four weeks starting from when the mice were 14-weeks-old. Following treatment completion, tumor tissues and liver tissues were collected for evaluation of tumor lesions and immunological analysis.

### Hydrodynamic tail vein injection *Trp53*$^{KO}$/*MYC*$^{OE}$ somatic genome editing mouse HCC model (*Trp53*$^{KO}$/*MYC*$^{OE}$ HCC model) animal studies
Eight-week-old male C57BL/6 mice were intravenously injected with a plasmid mix diluted in PBS, with the total injection volume corresponding to 10% of their body weight, via the lateral tail vein within 6-8 seconds, as described[29]. The plasmid mixture comprises 30 µg of pXB-U6-based CRISPR-Cas9 vector encoding a mouse *Trp53* single-guide RNA (sgRNA) with the sequence CCTCGAGCTC CCTCTGAGCC, 10 µg of pSB-EF1A-based transposon vector encoding MYC, and 2 µg of Sleeping Beauty transposase (SB100X) vector. The C57BL/6 mice were intraperitoneally injected with anti-mouse PD-L1 (BioXCell Cat# BE0101, 5 mg/kg per mouse) combined with anti-mouse VEGFA (Biolegend Cat# 512810, 50 mg per mouse) (DPVB), and low-dose radiation therapy (LDRT) consisting of locally delivered liver irradiation at a dose of 1 Gy every four days for four weeks starting from starting from when the mice 3 weeks post hydrodynamic tail vein injection. Following the completion of treatment, tumor tissues and liver tissues were collected for assessment of tumor lesions and immunological analysis.

## LDRT therapy

Mice were well anesthetized to adequately expose the upper abdominal liver area before ionizing radiation. LDRT was then conducted locally with 1 Gy at a dose rate of 12 Gy/min (225 KV, 17.7 mA) using the X-ray irradiator (Rad Source, Cat# Rs2000).

For the LDRT of Hepa 1-6 cells, the cells were administrated 1 Gy a day, lasting for 2 days (total for 2 Gy). After 24 h of the last radiation, the cells were collected for the *Cxcl10* mRNA detection and the supernatant of cells was collected for the Cxcl10 ELISA assay.

## Safety evaluation

Serum samples and the livers of Hepa1-6 orthotopic tumor-bearing mice in each group were obtained after 2 therapeutic cycles. Serum samples of DEN + CCl$_4$ HCC mouse model and *Trp53$^{KO}$/MYC$^{OE}$* HCC model in each group were obtained after 4 weeks treatment. Liver function blood biochemical parameters alanine aminotransferase (ALT) and aspartate aminotransferase (AST) were analyzed by an automatic analyzer (Hitachi 3100, Japan). The livers were fixed for H&E staining to assess pathological changes. The body weight of the Hepa1-6 orthotopic tumor-bearing mice was measured every two days from the first day before treatment to the end of the second therapeutic cycle.

## TUNEL assay

The apoptosis of tumor cells was detected after treatment by the TUNEL cell apoptosis detection kit (Beyotime, Cat# C1086). Briefly, the sections were deparaffinized and hydrated (3 times, 5-10 minutes per time) before dropping 20 μg/ml of proteinase K desquamation and reacted at 37 °C for 30 minutes. The sections were washed three times with PBS. Then the TUNEL assay solution was dropped and incubated at 37 °C for 60 minutes in the dark. The sections were washed three times with PBS. The sections were sealed with antifluorescence quenching mounting solution containing 4',6-diamidino-2-phenylindole (DAPI) and then observed under a fluorescence microscope.

## Single-cell suspensions preparation

Tumor samples were dissociated in 1640 medium containing 10% fetal bovine serum (FBS) and 1% 100X penicillin-streptomycin solution (P/S) supplemented with 0.5 mg/ml of collagenase-IV (Sigma-Aldrich, Cat# C5138) and 0.1 mg/ml of DNase-I (Sigma-Aldrich, Cat# DN25) at 37 °C for 1 hour. Uniform single-cell suspensions were obtained after smashing digested tissues with a syringe plunger on a 40 μm filter. Then, differential speed centrifugation to isolate tumor-infiltrated immune cells (TIICs) from the single-cell suspensions by percoll (Pharmacia (GE), Cat# 17089109-1). The single cell suspensions from DEN+CCl$_4$ tumors were further sorted by CD45 Positive Selection Kit (STEMCELL, Cat# 18945). The protocol used for the preparation of single cells of dLNs was obtained by smashing tissues with a syringe plunger on a 40 μm filter. Human peripheral blood lymphocyte isolation medium (TBD, Cat# LTS1077) centrifugation was used to isolate peripheral blood mononuclear cells (PBMC) from blood samples.

## Single-cell RNA sequencing (scRNA-seq) of TIICs from Hepa1-6 HCC tumors

For the CON, LDRT, DPVB, LR-DPVB groups, 3 independent single-cell suspensions were evaluated for each group (12 samples in total). Individual single-cell suspensions of 12 samples were separated by flow cytometry sorting (BD FACS Aria II) from single-cell suspensions mixed with 12 samples according to GFP or tCD19 tags, followed by scRNA-seq processing. According to the protocol, cell suspension was loaded onto RhapsodyTM Cartridge (BD) using BD RhapsodyTM Cartridge Kit (BD, Cat# 633731) and BD RhapsodyTM Cartridge Kit (BD, Cat# 633733) to generate single-cell magnetic beads in microwells, that is, individual cells were suspended in sample buffer (BD). The captured cells were lysed, then the released RNA was barcoded by reverse transcription in individual microwells. cDNA was generated after 45 min of reverse transcription on a ThermoMixer®C (Eppendorf) at 1200 rpm and 37 °C, followed by amplification and quality assessment using an Agilent 4200. scRNA-seq library was constructed using BD RhapsodyTM WTA Amplification Kit (BD, Cat# 633801) according to the manufacturer's instructions, and finally sequenced using Illumina Novaseq6000 sequencer.

## Data analysis of scRNA-seq of TIICs from Hepa1-6 HCC tumors

The sequencing data (fastq file) was processed by BD Rhapsody analysis pipeline and the reference genome was GRCh38 (Ensembl). Single-cell data were mainly processed by the R package Seurat[73] (v.4.3.0) for cell filtering, dimensionality reduction clustering, etc. Performing cell filtration first, we removed cells: expressed fewer than 200 genes and had more than 25% mitochondrial genes. The NormalizeData and ScaleData functions were used to normalize the data. A total of 6,913 cells were utilized for scRNA-seq analysis. The top 2000 highly variable Genes (HVGs) were selected for principal component analysis (PCA), and then cell clustering and UMAP dimensionality reduction visualization were performed based on the top 20 principal components. We used the R package COSG[74] (v.0.9.0) to calculate the characteristic genes of each cell group to identify the cell identity. To identify specific T-cell subsets, we annotated cell identities using reference data in the R package ProjecTILs[75] for matching. To calculate differentially expressed genes (DEGs), we used the Wilcoxon Rank Sum test, the default method of FindMarker, for statistics, and retained DEGs with a *P* value of less than 0.05. For pseudotime analysis, we used the DDRTree method in the R package monocle[76] (v.2.18.0) for dimensionality reduction and pseudotime calculation of cells.

## Single-cell RNA sequencing (scRNA-seq) of CD45$^+$ TIICs from DEN + CCl$_4$ HCC tumor

For the CON, LDRT, DPVB, LR-DPVB groups, 3 independent single-cell suspensions were evaluated for each group (12 samples in total). Individual single-cell suspensions of 12 samples were separated by mouse CD45 Positive Selection Kit (STEMCELL, Cat# 18945). For each sample, the cell suspension was resuspended in Labeling Buffer with PBS containing 1% BSA. Add TruStain FcX™ Antibody (Biolegend, cat# 101319), incubate and add different hashtag antibody, like TotalSeq™-B0301 anti-mouse Hashtag 1 Antibody (Biolegend) to each part. After incubation, wash, resuspend and mixing, using a Single Cell 3' Library and Gel Bead kit (10X Genomics, cat# 1000092) and Chromium Single Cell B Chip kit (10X Genomics, cat# 1000074), the cell suspension was loaded onto a Chromium single cell controller (10X Genomics) to generate single-cell gel beads in the emulsion (GEMs) according to the manufacturer's protocol. Briefly, single cells were suspended in PBS containing 0.04% BSA. Approximately 48,000 cells were added to each channel and approximately 30,000 target cells were recovered. Captured cells were lysed and the released RNA was barcoded through reverse transcription in individual GEMs. Reverse transcription was performed on a C1000TM Touch Thermal Cycler (Bio Rad) at 53 °C for 45 min, followed by 85 °C for 5 min and a hold at 4 °C. Complementary DNA was generated and amplified, after which cDNA was separated for 3' gene expression library and for cell surface protein library using SPRIselect. According to the manufacturer's introduction, scRNA-seq libraries were constructed using a Single Cell 3' Library and Gel Bead kit and Chromium Single Cell B Chip kit. The libraries were sequenced using an Illumina Novaseq6000 sequencer with a paired-end 150-bp (PE150) reading strategy.

## Data analysis of scRNA-seq of CD45$^+$ TIICs from DEN + CCl$_4$ HCC tumor

The number of cells used for scRNA-seq of CD45$^+$ TIICs in DEN + CCl$_4$ HCC mice model was 23,150. Single-cell RNAseq data were processed using the Cell Ranger (v.7.0.0) with GRCm38 mouse reference and the

downstream analyses were using the filtered gene expression matrices by R software (v.4.0.4) with the Seurat package (v.4.0.5)[73]. For filtering, after removed the doublet or negative cells using the HTODemux function, genes expressed >3 and cells with >200 genes detected and <25% UMIs derived from the mitochondrial genome were selected for further analyses. For the purpose of identify and compare shared cell type that were present across these samples, we used Harmony[77] standard workflow for normalization, variable gene selection, integration, dimensionality reduction and clustering as all details can be found in the website tutorial (https://htmlpreview.github.io/?https://github.com/satijalab/seurat.wrappers/blob/master/docs/harmony.html). After cells were clustered, the FindAllMarkers function was used to detect cluster-specific expressed genes, base on which clusters were classified and annotated. Differential gene expression testing was performed using the FindMarkers function in Seurat with Wilcoxon Rank Sum test by default and the Benjamini–Hochberg method was used to estimate the false discovery rate (FDR). DEGs were filtered using a minimum log2 (fold change) of 0.25 and a maximum FDR value of 0.05. Enrichment analysis for the functions of the DEGs was conducted using clusterProfiler package[78]. The Monocle2 package (v2.18.0)[79] was used to analyze single-cell trajectories in order to discover the cell-state transitions.

## Flow cytometry analysis

Single-cell suspensions for flow cytometry analysis were prepared as described above. For extracellular staining, cells were stained for 30 min at 4 °C with the surface molecules fluorochrome-conjugated monoclonal antibodies in 100 μl PBS with 2% FBS, 2 mM EDTA. The cells were subsequently washed twice and fixed in Fix/Perm buffer (Tonbo Biosciences, Cat# TNB-1020-L050) for intracellular staining. Then, before adding intracellular markers' antibodies, cells were washed with the permeabilization buffer (Tonbo Biosciences, Cat# TNB-1213-L150). For intracellular cytokine secretion detection, cell suspensions were restimulated in vitro in the presence of cell activation cocktail (Biolegend, Cat# 423303) for 4 hours. Fluorescence data were acquired on a BD LSRFortessa cytometer and analyzed using FlowJo V.X. Antibodies and dye used for flow cytometric analysis are listed in Supplementary Table 8. FACS sequential gating strategies in Supplementary Fig. 13.

## Flow cytometry sorting

Single-cell suspensions for flow cytometry sorting were prepared as described above. The extracellular staining step was same as the flow cytometry analysis. The step of sorting was performing by CytoFLEX SRT cytometer. Antibodies and dye used for flow cytometric analysis are listed in Supplementary Table 8.

## Immunohistochemistry (IHC)

The tissues were fixed in 4% paraformaldehyde (JETWAY, Cat# JTW003-500), embedded in paraffin, and sectioned into 4 μm-thick slices. Endogenous peroxidase activity was blocked, followed by antigen retrieval. The slides were then stained with primary antibodies against CD4 (abcam, EPR19514, Cat# ab183685, 1:1000), CD8 (abcam, EPR21769, Cat# 217344, 1:2000) at 4 °C overnight. After incubation with a secondary antibody (Dako, Cat# K5007) for 1 hour at 37 °C, the signal was revealed by the DAB Kit (Dako, Cat# K5007). And the sections were counterstained with hematoxylin (Servicebio, Cat# G1004-500ML).

## In vivo depletion/blockade experiments

For the in vivo depletion/blockade experiments, specific antibodies as well as isotype controls were injected (i.p., 200 μg per mouse every 4 days) at the day before treatment started. Antibodies used for in vivo studies are listed as follows: Rat IgG2b isotype control-InVivo (Selleck Cat# A2116), anti-mouse CD8α-InVivo (Selleck Cat# A2102), anti-

mouse CD4-InVivo (Selleck Cat# A2101), InVivoMAb polyclonal Armenian hamster IgG (BioXCell, Cat# BE0091), InVivoMAb anti-mouse CXCR3 (CD183) (BioXCell, Cat# BE0249). For the lymphatic drainage blocking experiment, Fingolimod (FTY720) (Selleck Cat# S5002, 2.5 mg/kg) was delivered orally every 3 days. Each dose was administered 1 day (blockade/depletion antibodies) or 3 days (FTY720) before the appropriate treatment regime.

## Adoptive transfer cells

Single-cell suspensions and cells staining for flow cytometry sorting were prepared as described above. Then, the SLAMF6+ PD-1+ CD8+ and SLAMF6- PD-1+ CD8+ T cells were harvested from mice bearing Hepa1-6 orthotopic hepatoma for 20 days. $1 \times 10^5$ sorted SLAMF6+ PD-1+ CD8+ or SLAMF6- PD-1+ CD8+ T cells in 100 μL PBS were injected into the peritoneum of recipient mice with the DPVB every 4 days after 2 days of inoculation with Hepa1-6 hepatoma. The mice were treated with DPVB after 10 days inoculation of Hepa1-6 hepatoma. The part of recipient mice was sacrificed after 2 therapeutic cycles, and tumor samples were photographed and measured and then performed the TUNEL and flow cytometry assay. In another part of recipient mice was continued to be observed for changes in their tumor volume.

## Patient-derived tumor fragment (PDTF)

The resected HCC tissue for subsequent PDTF culture was immediately manually cut into the small tumor fragments (PDTFs) of 1 to 2 mm³ in size on ice. Next, individual PDTF were embedded in an artificial extracellular matrix. The precooled 96-well plate was coated with 40 μl matrix gel per well as the bottom layer, and the matrix gel was allowed to solidify at 37 °C for 20 to 30 min. PDTFs was rinsed multiple times on the cell filter in the 6-well plate. One PDTF was placed in one well on the precured substrate, and a second layer of 40 μl of matrix gel was added. The 96-well plates were placed in an incubator at 37 °C for 20 to 30 min. After solidification, 120 μl tumor medium was added to the matrix gel[34]. Anti-PD-L1 antibody (BioXcell, Cat# BE0285) and anti-VEGFA antibody (Roche, Cat# Avastin) 10 μg / ml at a final concentration of 5 μg / ml of anti-CD3 (Biolegend, Cat# 317326) in combination with 2 μg / ml anti-CD28 (Biolegend, Cat# 302943). PDTF cultures were reacted at 37 °C for 48 h after drug treatment. After 48 h of ex vivo incubation, the therapeutic effect of DPVB on the PDTFs was verified by TUNEL assay and H&E staining.

## RNA sequencing

The total RNA of tumor tissues was extracted using an RNA Quick Purification Kit (a segene, Cat# RN001-1) according to the manufacturer's instructions. Select the corresponding testing methods for quality inspection according to the requirements of samples and products. A certain amount of RNA samples is denatured at suitable temperature to open their secondary structure, and mRNA is enriched by oligo (dT) -attached magnetic beads. The reaction system is conFig.d. After reacting at the suitable temperature for a fixed period, RNAs are fragmented. Prepare the first-strand synthesis reaction system, and set up the reaction program, synthesize the first strand cDNA, prepare the second-strand synthesis reaction system, and set up the reaction program to synthesize the second strand cDNA. After the reaction system and program are conFig.d and set up, double-stranded cDNA fragments are subjected to end-repair, and then a single 'A' nucleotide is added to the 3' ends of the blunt fragments. The reaction system and program for adaptor ligation are subsequently conFig.d and set up to ligate adaptors with the cDNAs. The PCR reaction system and program are conFig.d and set up to amplify the product. The corresponding library quality control protocol will be selected depending upon product requirements. Single-stranded PCR products are produced via denaturation. The reaction system and program for circularization are subsequently conFig.d and set up. Single-stranded cyclized products are produced, while uncyclized linear DNA

molecules are digested. Single-stranded circle DNA molecules are replicated via rolling cycle amplification, and a DNA nanoball (DNB) which contain multiple copies of DNA is generated. Sufficient quality DNBs are then loaded into patterned nanoarrays using high-intensity DNA nanochip technique and sequenced through combinatorial Probe-Anchor Synthesis (cPAS). The subsequent analysis and data mining were performed on Dr. Tom Multi-omics Data mining system (https://biosys.bgi.com).

### Generation of bone marrow-dendritic cells (BMDCs) and bone marrow-derived macrophages (BMDMs)

The bone marrow was extracted from the femur and tibia of mice by flushing the bones with RPMI. Subsequently, the cells were filtered through a 70 μm filter, washed twice with PBS, and cultured at a density of $1.5 \times 10^6$ cells/ml. For generation of BMDCs, the mouse bone marrow cells were placed in 10 cm cell culture dishes and cultured in RPMI 1640 medium (Gibco) containing 20 ng/mL recombinant GM-CSF (Peprotech, Cat# 250-05), 10 ng/ml recombinant IL-4 (Peprotech, Cat# 214-14) and 10% FBS (PAN-Biotech). For generation of BMDMs, the mouse bone marrow cells were cultured in RPMI 1640 medium (Gibco) containing 100 ng/mL recombinant M-CSF (Peprotech, Cat# 315-02) and 10% FBS (PAN-Biotech). Fresh complete media with GM-CSF and IL-4 (or M-CSF) was added into culture on day 3. BMDCs and BMDMs were harvested for further experiments on day 7.

### Co-culture assays

The $3 \times 10^6$ Hepa1-6 with or without radiation were plated into 10 cm cell culture dishes. BMDMs or BMDCs were added and co-cultured with Hepa1-6 cells at the ratio of 1:1. The co-culture supernatant was collected 48 hr later.

### Multiplex immunohistochemistry (mIHC)

The mIHC staining was performed using tyramide signal amplification (TSA) Plus Fluorescence Kits (PerkinElmer, USA) combined with IHC (TSA-IHC). Various primary antibodies, including TCF1 (CST, C63D9, Cat# 2203 S, 1:200), PD-1 (CST, D4W2J, Cat# 86163 S, 1:200) and CD8 (abcam, C8/468 + C8/144B, Cat# ab199016, 1:200) were sequentially applied at 37 °C for 1 hour, followed by horseradish peroxidase-conjugated secondary antibody incubation and TSA. The slides were microwave heat-treated after each TSA operation. The nuclei were counterstained with DAPI. The images were acquired on the tissue FAXS SL spectra (TissueGnostics, Austria) and analyzed using tissue-FAXS SL viewer.

### Immunofluorescence (IF)

For immunofluorescence, tumor tissues were prepared as described for IHC and sections were incubated with primary antibodies against CD8 (proteintech, 1G2B10, Cat# 66868-1-1 g, 1:400) and GZMB (abcam, EPR22645-206, Cat# ab255598, 1:3000), CD4 (abcam, EPR6855, Cat# ab133616,1:500) and FOXP3 (abcam, mAbcam 22510, Cat# ab22510, 1:100) at 4 °C overnight. After washing with PBS, sections were incubated with Cy3 Goat-Anti-Rabbit IgG (H + L) Antibody (APE x BIO, Cat# K1209) or FITC Goat-Anti-Mouse IgG (H + L) Antibody (APE x BIO, Cat# K1201) at room temperature for 1 hour. The sections were sealed with antifluorescence quenching mounting solution containing DAPI and then observed under a fluorescence microscope. The IF staining was evaluated by ImageJ according to the double positive cells as a percentage of the total number of cells. Five fields were randomly selected for each slide. The ratio of Treg/Tef ratio was divided by the median of the ratio.

### Real-time PCR

The total RNA of tumor tissues and cells was extracted using an RNA Quick Purification Kit (a segene, Cat# RN001-1) according to the manufacturer's instructions. Then, reverse transcribe of the RNA to complementary DNA (cDNA) using HiScript III RT SuperMix for real time quantitative PCR (RT-qPCR). SYBR Green III (Vazyme, Cat# Q711-02/03) was used for the RT-qPCR of non-TaqMan primers, and Universal Master Mix II with UNG (Thermo Fisher, 4440038) was used for the RT-PCR of TaqMan primers. The primer sequences are listed as follow: *Gapdh*-Forward: CATCACTGCCACCCAGAAGACTG; *Gapdh*-Reverse: ATGCCAGTGAGCTTCCCGTTCAG. *Cxcl9*-Forward: CCTAGT-GATAAGGAATGCACGATG; *Cxcl9*-Reverse: CTAGGCAGGTTTGATCTC CGTTC. *Cxcl10*-Forward: ATCATCCCTGCGAGCCTATCCT; *Cxcl10*-Reverse: GACCTTTTTTGGCTAAACGCTTTC.

### Enzyme-linked immunosorbent assay (ELISA)

Cxcl10 from the supernatants of Hepa1-6 cells, BMDMs, BMDCs and co-culture ELISAs was performed according to the manufacturer's instructions using the mouse IP-10 SimpleStep ELISA kit (CXCL10) (Abcam, Cat# ab260067). Briefly, 50 μL of sample and 50 μL of the antibody cocktail were added to each well. Then, the cells were incubated for 1 hour at room temperature on a plate shaker set to 400 rpm. After incubation, each well was washed with 350 μL of 1X wash buffer three times. In the last wash, the plate was inverted and tapped gently against clean paper towels to remove excess liquid. Then, 100 μL of TMB Development Solution was added to each well and incubated for 10 minutes in the dark on a plate shaker set to 400 rpm. Finally, 100 μL of stop solution was added to each well, and the plate was shaken on a plate shaker for 1 minute to mix. The OD was recorded at 450 nm.

### Patient sample collection

HCC tumor tissues used for PDTFs were acquired from 6 HCC patients, aged between 34 and 61 years, who underwent open hepatectomy at the Third Affiliated Hospital of Sun Yat-sen University. HCC tumor biopsy specimens were collected from 9 patients with HCC, aged between 43 and 83 years, at the Eastern Hepatobiliary Surgery Hospital, Naval Medical University, for multiplex immunohistochemistry (mIHC) analysis. Fresh blood samples were collected from 9 HCC patients, aged between 45 and 64 years, after T + A treatment in the Third Affiliated Hospital of Sun Yat-sen University. Fresh clinical tumor samples and blood samples were acquired from 20 HCC patients, aged between 38 and 85 years, who underwent open hepatectomy at the Third Affiliated Hospital of Sun Yat-sen University. Samples for IF were collected from 120 patients who underwent laparoscopic or open hepatectomy at the Third Affiliated Hospital of Sun Yat-sen University from January 2011 to December 2017.

### Statistical analysis

GraphPad Prism V.7 and IBM SPSS Statistics V.20 were used to analyze the data. The statistical tests used for data analysis included Student's $t$ test (two-tailed), One-way repeated-measures ANOVA test, the $\chi^2$ test and the log-rank test. Univariate and multivariate statistical analyses were performed using the Cox regression model. The survival analysis was performed by the Kaplan-Meier analysis. $P < 0.05$ was considered to indicate statistical significance.

### Reporting summary

Further information on research design is available in the Nature Portfolio Reporting Summary linked to this article.

## Data availability

The raw data from four groups of scRNA-seq for mouse tumor infiltrating immune cells and RNA-seq for mouse tumor tissues in Hepa1-6 tumors generated in this study have been deposited in the Genome Sequence Archive (GSA) database under accession code CRA011374. The raw data from four groups of scRNA-seq for mouse CD45+ tumor infiltrating immune cells s in DEN + CCl4 tumors generated in this study have been deposited in the Genome Sequence Archive (GSA) database under the accession code CRA012644. The Kaplan-Meier Plotter

[http://kmplot.com/analysis] publicly available data used in this study are available in The Cancer Genome Atlas (TCGA) database (project TCGA-LIHC) under dbGaP Study Accession phs000178 [https://cancergenome.nih.gov/]. The remaining data are available within the Article, Supplementary Information or Source Data file. Source data are provided with this paper.

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

## Acknowledgements

This work was supported by the National Natural Science Foundation of China, 82103448 (to L.Y.), 81972286 (to Y.Y.), 82270688 (to H.L.), 82172585 (to W.L.). Guangdong Natural Science Foundation, 2023A1515010322 (to L.Y.). the Science and Technology Program of Guangdong Province (2019B020236003 to Y.Y.). the Guangdong Key Laboratory of Liver Disease Research (2020B1212060019 to Y.Y.). "Five of Five" program of Third Affiliated Hospital of Sun Yat-sen University (2023WW105 to H.L.). The authors would like to acknowledge eBioart for their invaluable guidance in the production of pattern diagrams.

## Author contributions

Y.Y., L.Y., S.C. and S.L. designed the project. S.L. and K.L., performed most of the experiments. H.L., K.W., H.Y. and Y.L. collected the clinical samples. K.W. and S.C. performed the clinical research. H.Y. and Z.L. performed the PDTF experiments. S.L and X.W. analyzed the RNA-seq data. M.S., Z.Y. and Y.H. performed immunohistochemistry assay. H.L. and W.L. analyzed the clinical data. H.L. directed the revision of the manuscript. S.L. and K.L. wrote the manuscript.

## Competing interests

The authors declare no competing interests.
