## [Peer Review File · Nature Communications]

Low-dose Radiotherapy Combined with Dual PD-L1 and VEGFA Blockade Elicits Antitumor Response in Hepatocellular Carcinoma Mediated by Activated Intratumoral CD8+ exhausted-like T cellsEditorial Note: Parts of this Peer Review File have been redacted as indicated to maintain patient confidentiality.

REVIEWER COMMENTS

Reviewer #1 (Remarks to the Author): with expertise in HCC, cancer (immuno)therapy

Combination treatment of anti-VEGF and anti-PD-L1 (atezolizumab) is currently the most effective first-line treatment for liver cancer (HCC). This study demonstrated in mouse HCC model that low-dose irradiation (LDRT/LR) improved the efficiency of dual VEGFA and PD-L1 blockade (DPVB). The study began from mouse HCC derived from a mouse HCC cell line, Hepa1-6. Authors performed scRNA seq to compare immune profiles of HCC tumors derived from mice treated with DPVB alone and LR+DPVB. They found that TCF1+PD1+HighCD8+ T cells (Tpex) were increased. [redacted] Furthermore, authors performed retrospective studies and found that patients with HCC infiltrated with higher number of TCF1+PD1+CD8+ T cells are more likely to respond.

Overall, the concept of the study is clear. This is a study with clinical relevance with simple experimental designs. Some experiments lack proper controls and the scRNA analysis is brief. Mechanistic link is not very extensive.

1) The whole study is dependent on one mouse HCC model. Hepa-1-6 was implanted into syngeneic mice. Hepa-1-6 is not an idealistic mouse HCC model as Hepa-1-6 tumors regress a few weeks after implantation due to certain level of immune rejection despite Hepa-1-6 is a syngeneic cell line. Standard models such as hepatocarcinogen (DEN-CCL4) and hydrodynamic tail vein injection (somatic genome editing)-mediated mouse HCC models should be employed to confirm this important observation. At this stage, it is unclear whether the combined effect of radiation+anti-PD-L1+anti-VEGF (LR+DPVB) is mouse model specific. Furthermore, Hepa-1-6 model is immunogenic and is quite responsive to immune checkpoint blockade, authors should explore other models which represent more immune desert (cold) models (few CD8+ T cell infiltrates)

2) The mechanisms by which radiation promotes stem-like CD8+ Tpex recruitment are not clear. Cxcl10/Cxcr3 might be responsible however what elicits Cxcl10 after radiation treatment. The answer is limited by the lack of experimental control with radiation alone.

3) Single cell analysis currently only includes (1) anti-PD-L1 and anti-VEGF treatment (DPVB) (2) anti-PD-L1 and antiVEGF plus radiation (LR+DPVB). Authors should include untreated and radiation alone in all analysis. Since scRNA seq data are from mice but not human, these experimental controls should be easy to access.

4) While the clinical trial data are highly appreciated, this is a single arm study. The patients' response might only be due to anti-PD-L1 and anti-VEGF (DPVB) but not the radiation.

5) What happened to other immune cells in the LDRT+DPVB as compared to single treatments and control? Antigen presenting cells should be considered as radiation might increase antigen presentation caused by dying cells.

6) Lack of consistencies in the data.

e.g. Figure 2 compared the single cell data, flow data, IHC data on CD⁺ T cells with control-treated and LR-DPVB treated mice. Figure 3 compared DPVB and LR-DPVB. Figure 4 switched back to control and LR-DPVB.

7) Number of T cells/ mass of the tumors should be used to present the T cell amount instead %.

8) Number of cells used for the scRNA analysis should be indicated. How many T cells were analyzed in each group?

Minor:

9) It is unclear how PDTF works. PDTF does not seem to be a standardized method to test the response of drugs.

10) Discussion is a bit out of scope. Authors should stay focused and are encouraged to make the discussion more concise.

11) Methodologies are very brief for some experiments.

12) Labeling is confusing. E.g. TCF1-PD1-HighCD8⁺ T cells (figure 3) and TCF1+PD1+CD8⁺ T cells are the same? The former is quite confusing.

Reviewer #2 (Remarks to the Author): with expertise in HCC, cancer (immuno)therapy

This manuscript reports the anti-tumor activity of dual PD-L1 and VEGFA blockade (DPVB) with low-dose radiotherapy (LDRT) in orthotopic models of murine HCC. The study demonstrated that LDRT enhanced DPVB by enlarging intratumoral stem-like CD8⁺ T_{pex},

recruited from the dLNs via the CXCL10/CXCR3 axis. The authors also suggest that stem-like CD8⁺ T_{PEX} may be biomarkers for selecting patients with HCC before T+A combination therapy and predicting treatment outcomes. While the paper is well-written, and the study provides new insights into the potential of combining immunotherapies and radiotherapy in clinical practice, I have several concerns.

Specific comments:

1. The authors used an orthotopic tumor model with murine Hepa1-6 HCC cell injection. The authors found the benefit of LR-DPVB with distinct immune profiles in this model. Additional models are needed to increase the clinical relevance of this study. The Hepa1-6 model is highly sensitive to immunotherapy and thus does not recapitulate most human HCC cases. Additionally, most HCC patients have underlying liver disease and poor liver function, and the ICB efficacy/resistance may depend on the type of liver damage. The key data reported in this manuscript must be confirmed in orthotopic or autochthonous models, ideally in ICB-resistant models with pre-existing liver injury, to show the potential to sensitize the tumor to ICBs and the correlation between the treatment efficacy and immune subset profiles and support the conclusions.

2. Moreover, I suggest a comparison with DPVB in addition to the comparison with LR-DPVB and the control group. This would allow for a more comprehensive analysis of the additive effect of combining low-dose radiation with DPVB therapy and elucidate the mechanisms of action.

3. Regarding the scRNAseq data analysis presented in Figs. 2, 4, and 5, it would be interesting to examine the changes in immune subsets other than CD8⁺T cells, such as B cell and macrophage clusters, which may significantly affect the tumor immune microenvironment (B cells also showed significant change in Supplementary Fig. 2D). Previous studies have reported that LDRT can reshape the immune microenvironment. For example, it would be important to determine whether these changes indicate a reduction in immunosuppressive subsets, increased T-cell activation, or both. A previous study by Herrera et al. found that LDRT treatment reprogrammed the tumor microenvironment of tumors with scarce immune infiltration. And LDRT combination with immunotherapy induced simultaneous mobilization of innate and adaptive immunity, including CD4⁺

effector T cells, CD8⁺ T cells, and myeloid compartments, such as macrophages, DCs, and monocytes (PMID: 34479871).

4. scRNA sequencing analysis revealed that CD8_{Tex} was the enriched T cell subset after LR-DPVB treatment compared to DPVB-treated tumors. The authors stated that CD8_{Tex} from the LR-DPVB-treated tumors expressed higher levels of effector function and cytolytic capacity-related molecules, such as *Ifng*, *Tnf*, and *Gzmb*. However, Supplementary Fig. 3D shows no statistically significant difference in the expression level of these molecules in CD8_{Tex} between LR-DPVB and DPVB treatment groups. Instead, the statistically significant difference in the expression levels of *Ifng* and *Gzmb* was in the CD8_T subset, as shown in Fig. 3D. Therefore, the conclusion that "the enhancement of the tumor-rejecting effect after LR-DPVB treatment was largely dependent on the magnitude of intratumoral CD8⁺ Tex, which featured potent effector function and cytolytic capacity" needs additional evidence.

5. In Fig. 4, the authors demonstrate that SLAMF6⁺PD-1⁺CD8 T cells include more GzmB-positive cells than SLAMF6⁻PD-1⁺CD8 T cells. It would be important to examine the cytokine production capacities of these subsets, specifically depending on the expression of *Tim3*, as previous reports have shown that this can vary.

6. It would be interesting to present the durability of responses for the survival and tumor volume curves shown in Fig. 1. Additionally, from the perspective of clinical relevance, it would be helpful to describe the subsequent toxicity evaluation (e.g., weight loss, liver enzymes) after day 7.

7. According to the hypothesis tested in this study, stem-like CD8⁺ T_{pex} are recruited following stimulation by LDRT. In the animal study presented in Fig. 1A, LDRT was administered from day 0. However, in the experiment shown in Fig. 4D, where cells were used to mimic the enhancement of CD8⁺ T_{pex} by LDRT, the cells were administered from day -8. What is the explanation for the difference in the timing of treatment between these two experiments?

8. The results indicate that *Cxcl9* and *Cxcl10* were significantly upregulated in LDRT-treated tumors. What is the mechanism?

[redacted]

Minor comments:

1. In Fig. 4K, the authors demonstrated a significant increase in apoptotic tumor cells in the PDTFs derived from stem-like CD8 + Tpex high infiltration HCC tissues after T+A treatment. What criteria were used to define "high" infiltration in this context?
2. The figure caption for Fig 2C is missing, which makes it difficult to differentiate the different colors representing the types of T cells.

Reviewer #3 (Remarks to the Author): with expertise in cancer immunology, omics

The authors have done a fabulous work. But there are some significant concerns that need to be addressed

Major comments

- (1) The order in which the figures and figure panels were introduced is messed up. For example, the authors directly jumped into Fig. 3d without introducing Fig. 3a-c
- (2) “By further analyzing of these scRNA-seq data, we also identified CD8 160 T cells populations featured with a higher exhaustion and effector function were 161 inclined to accumulate after LR-DPVB in contrast to DPVB (Supplementary Fig. 3C, 162 Fig. 3D).” Fig. 3d does not seem to support the conclusion of this sentence. In fact, Fig. 3d is poorly labeled and I do not understand what I am seeing here. Also there are not really any differences between the two groups
- (3) The authors need to remove most, if not all, of their pseudotime analyses. “Pseudotime curve analysis revealed ... from LR-DPVB-treated tumors showed significant 206 enrichment of CD8_Tef (Fig. 3G right, 3H)” Pseudotime analysis tools are known to be misleading (citation: A comparison of single-cell trajectory inference methods | Nature Biotechnology). And the authors’ results (Fig. 3gh) here indeed show that there is no extra benefit of putting their scRNA-seq data under the context of pseudotime – they can draw the same conclusion using much simpler analyses and without doing the so-called pseudotime analyses

[redacted]

- (5) In this section: “7. Stem-like CD8+ T_{pex} cells and Treg/Tef ratio among T+A-treated HCC 327 patients and all HCC patients”, the authors showed that Treg/Tef ratio is predictive/prognostic in T+A treated patients or even baseline HCC patients (Fig. 7e). Then these results suggest that their previous statement on the discovery of Treg/Tef ratio for predicting LR-DPVB treatment response is non-specific. “As our scRNA-seq analysis previously showed that Treg/CD8_Tef may be a 344 potential curative effect index after LR-DPVB treatment (Fig. 3J-K)”. So I wouldn’t call Treg/Teff a curative effect index for LR-DPVB. It is just a general prognostic/predictive marker. Despite all the experimental and analysis results, the authors seem to have failed to really reveal anything that is specific to the synergistic effect of radiation treatment and T+A, which is the core of the whole study.
- (6) Immunotherapy+(low dose) radiation treatment is already a very well explored idea, in many different cancer types. What is new about the authors’ work?

Minor comments

- (1) “making it became the first successful first-line immunotherapy for HCC around the world 4-6”. Minor grammar problem. Can change to “making it the first ...” English editing by a native speaker is needed throughout the manuscript. Grammar seems off in too many places.
- (2) “Most Chinese HCC patients have a history of cirrhosis, which is the 56 predominant risk factor for RILD; therefore, the radiation tolerated dose of the liver is 57 significantly lower than that which is reported in foreign studies 16.” The intended audience for this manuscript should not be limited to any nation. The way this sentence is constructed misleadingly implies this manuscript is intended for Chinese readers only.
- (3) “Mouse reactive antibodies to realize dual PD-L1 and VEGFA blockade (DPVB) in the following experiments.” This sentence is broken and I don’t understand its meaning

(4) “Volcano plot and heatmap analyses showed significantly higher expression of CD8 T 156 cells following LR-DPVB relative to DPVB of exhausted markers” I understand the meaning of the sentence here, but the grammar is broken

(5) “we recognized nine different T-cell clusters that displayed 173 a comparable expression pattern of essential markers defining the lineage of T cells 174 across tumor types” I do not understand what the authors mean by “comparable” here

(6) “Since previous studies showed that TCF-1 and SLAMF6 were highly 230 coexpressed in stem-like CD8+ Tpex, while few terminal exhausted CD8+ T expressed 231 SLAMF6 30, we chose SLAMF6 as a cell-surface marker to distinguish the CD8+ 232 Tpex and terminal exhausted CD8+ T to isolate live stem-like CD8+ Tpex for further 233 investigation (Fig. 4C-D)” Sorry. I missed the logic here. Why you cannot use TCF-1 for this purpose?

POINT-TO-POINT RESPONSE TO REVIEWERS' COMMENTS

We express our gratitude to all reviewers for the insightful comments and recommendations, which have significantly enhanced the quality of this study. We believe we have addressed reviewers' comments to the best of our ability resulting in a significantly strengthened manuscript.

In our revised manuscript, we conducted experiments in additional HCC models including DEN+CCl₄ HCC model and *Trp53^{KO}/MYC^{OE}* HCC model to validate the effect and immunological profiling of LR-DPVB, presenting four experimental groups for comprehensive phenotypic validation. In mechanistic research, we further identified the primary cellular origin responsible for the upregulation of CXCL10 after LDRT. Additionally, we investigated the alteration of the immune subsets other than T cell after LR-DPVB treatment by performing scRNA-seq analysis and flow cytometry assays. **[redacted]** We believe that the new data improve the credibility of our work, and supports the conclusions drawn from our study, providing compelling preclinical evidence for the use of low-dose radiotherapy combined with atezolizumab and bevacizumab in the treatment of unresectable hepatocellular carcinoma patients.

In the following point-by-point response, the reviewers' comments are presented in blue, while our responses are displayed in black.

Reviewer #1 (Remarks to the Author): with expertise in HCC, cancer (immuno) therapy

Combination treatment of anti-VEGF and anti-PD-L1 (atezolizumab) is currently the most effective first-line treatment for liver cancer (HCC). This study demonstrated in mouse HCC model that low-dose irradiation (LDRT/LR) improved the efficiency of dual VEGFA and PD-L1 blockade (DPVB). The study began from mouse HCC derived from a mouse HCC cell line, Hepa1-6. Authors performed scRNA seq to compare immune profiles of HCC tumors derived from mice treated with DPVB alone and LR+DPVB. They found that TCF1+PD1+HighCD8+ T cells (Tpex) were increased. **[redacted]** Furthermore, authors performed retrospective studies and found that patients with HCC infiltrated with higher number of TCF1+PD1+CD8+ T cells are more likely to respond. Overall, the concept of the study is clear. This is a study with clinical relevance with simple

experimental designs. Some experiments lack proper controls and the scRNA analysis is brief. Mechanistic link is not very extensive.

1) The whole study is dependent on one mouse HCC model. Hepa-1-6 was implanted into syngeneic mice. Hepa-1-6 is not an idealistic mouse HCC model as Hepa-1-6 tumors regress a few weeks after implantation due to certain level of immune rejection despite Hepa-1-6 is a syngeneic cell line. Standard models such as hepatocarcinogen (DEN+CCI4) and hydrodynamic tail vein injection (somatic genome editing)-mediated mouse HCC models should be employed to confirm this important observation. At this stage, it is unclear whether the combined effect of radiation+anti-PD-L1+anti-VEGF (LR+DPVB) is mouse model specific. Furthermore, Hepa-1-6 model is immunogenic and is quite responsive to immune checkpoint blockade, authors should explore other models which represent more immune desert (cold) models (few CD8⁺ T cell infiltrates).

Response: We express our gratitude for the valuable comments provided by the reviewer. Following the suggestions, we conducted further verification for the combined therapeutic effect of LR-DPVB in two different mouse models: diethylnitrosamine and repeated carbon tetrachloride induced hepatocarcinogen model (DEN+CCL₄ HCC model) (**Fig. 1E**) and hydrodynamic tail vein injection (HDTV_i) *Trp53^{KO}/MYC^{OE}* somatic genome editing mediated mouse HCC models (*Trp53^{KO}/MYC^{OE}* HCC model) (**Fig. 1K**).

Specifically, DEN + CCL₄ mouse model was established by a single injection of DEN (25mg/kg) followed by repeated administration of CCL₄ (20%, 1μl/g, twice a week), thereby recapitulating genotoxic injury and advanced fibrosis-associated hepatocellular carcinoma (HCC) in humans¹⁻³. Furthermore, immunohistochemistry staining (IHC) revealed a scarcity of CD8⁺ T cell infiltrates in the liver tumors of the CON group from the DEN+CCL₄ HCC model (**Fig. 1I**), indicating that this model is closer to an immune desert environment resembling "cold" human HCC with limited responsiveness to immunotherapy. In the DEN+CCL₄ HCC model, mice in LR-DPVB showed a robust tumor control with a lower liver weight and a less tumor nodules comparing with DPVB group (**Fig. 1F-J**).

To validate this observation of LR-DPVB efficacy and established the clinical relevance between efficacy and key driver genes of human HCC, we took advantage of HDTV_i to generate a *Trp53^{KO}/MYC^{OE}* HCC model in which oncogenic *MYC* can be genomically integrated, and that *Trp53* is deficient to recapitulate the features of HCC, as previously described⁴⁻⁶. We observed more

therapeutic effects and survival benefits from LR-DPVB compared to DPVB in *Trp53^{KO}/MYC^{OE}* HCC model which is known to be an immune desert (“cold”) model with few CD8⁺ T cell infiltrates and ICB resistance reported in previous studies ⁴ (**Fig. 1L-P**).

Additionally, no significant changes were observed in mouse circulating liver enzymes alanine transaminase (ALT) and aspartate aminotransferase (AST) of the mice with different treatment in the same model, indicating minimal toxicity to the liver under long-term treatment with LR-DPVB (**Supplementary Fig. 1G-J**).

Overall, these results confirmed that LR-DPVB led to a marked therapeutic response with no obvious toxicity in various HCC models. **The aforementioned results and figures have been incorporated into the revised manuscript.**

References:

1. Liu, C., Yang, Y., Chen, C., Li, L., Li, J., Wang, X., Chu, Q., Qiu, L., Ba, Q., Li, X. & Wang, H. Environmental eustress modulates β -ARs/CCL2 axis to induce anti-tumor immunity and sensitize immunotherapy against liver cancer in mice. *Nat Commun* **12**, 5725 (2021).
2. Wang, B., Fu, J., Yu, T., Xu, A., Qin, W., Yang, Z., Chen, Y. & Wang, H. Contradictory effects of mitochondria- and non-mitochondria-targeted antioxidants on hepatocarcinogenesis by altering DNA repair in mice. *Hepatology* **67**, 623-635 (2018).
3. Fu, Y., Mackowiak, B., Feng, D., Lu, H., Guan, Y., Lehner, T., Pan, H., Wang, X. W., He, Y. & Gao, B. MicroRNA-223 attenuates hepatocarcinogenesis by blocking hypoxia-driven angiogenesis and immunosuppression. *Gut* (2023).
4. Yuen, V. W.-H., Chiu, D. K.-C., Law, C.-T., Cheu, J. W.-S., Chan, C. Y.-K., Wong, B. P.-Y., Goh, C.-C., Zhang, M. S., Xue, H. D.-G., Tse, A. P.-W., Zhang, Y., Lau, H. Y.-H., Lee, D., Au-Yeung, R. K. H., Wong, C.-M. & Wong, C. C.-L. Using mouse liver cancer models based on somatic genome editing to predict immune checkpoint inhibitor responses. *J Hepatol* **78**, 376-389 (2023).
5. Lee, D., Xu, I. M.-J., Chiu, D. K.-C., Leibold, J., Tse, A. P.-W., Bao, M. H.-R., Yuen, V. W.-H., Chan, C. Y.-K., Lai, R. K.-H., Chin, D. W.-C., Chan, D. F.-F., Cheung, T.-T., Chok, S.-H., Wong, C.-M., Lowe, S. W., Ng, I. O.-L. & Wong, C. C.-L. Induction of Oxidative Stress Through Inhibition of Thioredoxin Reductase 1 Is an Effective Therapeutic Approach for Hepatocellular Carcinoma. *Hepatology* **69**, 1768-1786 (2019).
6. Chiu, D. K.-C., Yuen, V. W.-H., Cheu, J. W.-S., Wei, L. L., Ting, V., Fehlings, M., Sumatoh, H., Nardin, A., Newell, E. W., Ng, I. O.-L., Yau, T. C.-C., Wong, C.-M. & Wong, C. C.-L. Hepatocellular Carcinoma Cells Up-regulate PVRL1, Stabilizing PVR and Inhibiting the Cytotoxic T-Cell Response via TIGIT to Mediate Tumor Resistance to PD1 Inhibitors in Mice. *Gastroenterology* **159**, 609-623 (2020).

2) The mechanisms by which radiation promotes stem-like CD8⁺ T_H1 recruitment are not clear. Cxcl10/Cxcr3 might be responsible however what elicits Cxcl10 after radiation treatment. The answer is limited by the lack of experimental control with radiation alone.

Response: We do appreciate the reviewer's comments, and this is a crucial point. Actually, we did

make a comparison between the LDRT group and the CON group. By performing KEGG enrichment pathway analysis on RNA-seq data of Hepa1-6 tumors from LDRT and CON group, we found that genes differentially highly expressed in LDRT-treated tumors were significantly associated with *cytokine and cytokine receptor interaction pathways* (**Fig. 5F**). Meanwhile, the *Cxcl9* and *Cxcl10* family genes was significantly enhanced after LDRT treatment among the *cytokine and cytokine receptor interaction pathway* (**Fig. 5G**). The qPCR experiments showed the upregulation of *Cxcl10* after LDRT in both Hepa1-6 model and *Trp53^{KO}/MYC^{OE}* model (**Supplementary Fig. 10F, 9J**). Moreover, CXCR3, which is the corresponding receptor of Cxcl9 and Cxcl10, was significantly upregulated in CD8⁺ T cells after LDRT treatment (**Fig. 5H, Supplementary Fig. 9L**). Notably, flow cytometry results showed that LDRT only increased the stem-like CD8⁺ T_{pex} in CXCR3⁺ CD8⁺ T cells but did not significantly change the percentage of stem-like CD8⁺ T_{pex} in CXCR3⁻ CD8⁺ T cells (**Fig. 5H, Supplementary Fig. 9L**). Furthermore, anti-CXCR3 treatment substantially weaken LDRT-induced accumulation of stem-like CD8⁺ T_{pex} cells in Hepa1-6 model and *Trp53^{KO}/MYC^{OE}* model (**Supplementary Fig. 10J, 9O-P**). Based on the above findings, we concluded that the stem-like CD8⁺ T_{pex} enhanced by LDRT was mainly recruited through the CXCL10/CXCR3 axis.

According to the recommendations of the reviewers, we further investigated which subset of cells was primarily responsible for the upregulation of CXCL10 following LDRT. As previous study has demonstrated, CXCL10 is primarily produced by antigen presenting cells (APCs), including dendritic cells and macrophages, as well as tumor cells ¹. In our experiments RT-qPCR and ELISA assays showed the expression level of CXCL10 in Hepa1-6 cells and supernatant of the cells was not significantly upregulated after LDRT treatment (**Supplementary Fig. 10G**). While, in the Hepa1-6 model and *Trp53^{KO}/MYC^{OE}* model, *in vivo* experiments revealed a significant increase in *CXCL10* expression in tumor-associated macrophages (TAMs, F4/80⁺ CD11b⁺ CD45⁺ L/D⁻) and dendritic cells (DCs, CD11c⁺ CD45⁺ L/D⁻) sorted from tumor-infiltrated immune cells (TIICs) in the LDRT group compared to the CON group (**Supplementary Fig. 10H, 9K**). Furthermore, *in vitro* co-culture experiments demonstrated a significant increase in the concentration of CXCL10 in the supernatant when bone marrow-dendritic cells (BMDCs) and bone marrow-derived macrophages (BMDMs) were co-cultured with LDRT-treated tumor cells compared to co-cultured with tumor cells that did not undergo LDRT treatment (**Supplementary Fig. 10I**). These results suggested that

myeloid APCs were the primary source of upregulated CXCL10 following LDRT, which in turn promotes recruitment of CXCR3⁺ CD8⁺ T_{pex}. **The corresponding results and figures have been incorporated into the fifth part of the Results of revised manuscript.**

References:

1. Reschke, R. & Gajewski, T. F. CXCL9 and CXCL10 bring the heat to tumors. *Sci Immunol* **7**, eabq6509 (2022).

3) Single cell analysis currently only includes (1) anti-PD-L1 and anti-VEGF treatment (DPVB) (2) anti-PD-L1 and antiVEGF plus radiation (LR+DPVB). Authors should include untreated and radiation alone in all analysis. Since scRNA seq data are from mice but not human, these experimental controls should be easy to access.

Response: We thank the reviewer's suggestions. To these concerns, we revised as follows.

Indeed, we have conducted a comprehensive analysis of the scRNA-seq data of Hepa1-6 tumors encompassing all four treatment groups in our previous study (**Supplementary Fig. 2D**). However, to elucidate the fundamental mechanisms underlying LR-DPVB combination therapy, we exclusively presented the corresponding comparisons (LR-DPVB vs DPVB) in our previously submitted manuscript. According to the reviewer's suggestion, **we have organized and presented the scRNA-seq analysis of all four groups of Hepa1-6 tumors in revised manuscript (Fig. 3, Supplementary Fig. 4-5, 7)**. Furthermore, to better present the correlation of immune profiling and therapeutic effect of LR-DPVB, we performed the scRNA-seq analysis of DEN-CCl₄ tumors treated with four groups in revised manuscript. **The scRNA-seq analysis and corresponding flow cytometry assays of four groups were also shown in Fig. 3, Supplementary Fig. 4-5, 7.**

[redacted]

Wang, K., Xiang, Y.-J., Yu, H.-M., Cheng, Y.-Q., Liu, Z.-H., Zhong, J.-Y., Feng, S., Ni, Q.-Z., Zhu, H.-F., Pan, W.-W., Li, J.-J., Liang, C., Zhou, H.-K., Meng, Y., Lau, W. Y. & Cheng, S.-Q. Intensity-modulated radiotherapy combined with systemic atezolizumab and bevacizumab in treatment of hepatocellular carcinoma with extrahepatic portal vein tumor thrombus: A preliminary multicenter single-arm prospective study. *Front Immunol* **14**, 1107542 (2023).

2. Herrera, F. G., Ronet, C., Ochoa de Olza, M., Barras, D., Crespo, I., Andreatta, M., Corria-Osorio, J., Spill, A., Benedetti, F., Genolet, R., Orcurto, A., Imbimbo, M., Ghisoni, E., Navarro Rodrigo, B., Berthold, D. R., Sarivalasis, A., Zaman, K., Duran, R., Dromain, C., Prior, J., Schaefer, N., Bourhis, J., Dimopoulou, G., Tsourti, Z., Messemaker, M., Smith, T., Warren, S. E., Foukas, P., Rusakiewicz, S., Pittet, M. J., Zimmermann, S., Sempoux, C., Dafni, U., Harari, A., Kandalaf, L. E., Carmona, S. J., Dangaj Laniti, D., Irving, M. & Coukos, G. Low Dose Radiotherapy

4) What happened to other immune cells in the LDRT+DPVB as compared to single treatments and control? Antigen presenting cells should be considered as radiation might increase antigen presentation caused by dying cells.

Response: We thank the reviewer for the comment. According to the reviewer's suggestion, further analysis and experiments were performed.

As illustrated of scRNA-seq analysis in **Supplementary Fig. 2D** and **Fig. 2C**, except for the CD8 T cell population, macrophages exhibited the highest enrichment among tumor infiltrating immune cells (TIICs) in Hepa1-6 HCC tumors treated with LR-DPVB compared to the CON group. In addition, we also analyzed the alterations of immune cell populations between LR-DPVB and DPVB in Hepa1-6 HCC tumors by scRNA-seq. Notably, compared to DPVB alone, the proportion of macrophages was significantly elevated in TIICs of Hepa1-6 HCC tumors following treatment with LR-DPVB (**Figure R1. 1A**). These findings suggest that macrophages may also play an important role in the therapeutic efficacy of LR-DPVB.

Figure R1. 1: (A) Fold-change of TIICs populations from LR-DPVB and DPVB groups in Hepa1-6 HCC tumor. (B) Fold-change of antigen presented cell populations (APCs) from LR-DPVB and DPVB groups in Hepa1-6 HCC tumor.

The pivotal role of antigen-presenting cells (APCs) in the immune response to tumors following radiotherapy has been widely acknowledged¹⁻², thus prompting our further investigation into alterations in APC populations subsequent to LR-DPVB treatment. As depicted in **Figure R1. 1B**, LR-DPVB significantly upregulated the proportion of macrophages. Flow cytometry analysis confirmed a significant increase in macrophage infiltration following LR-DPVB treatment compared to DPVB alone (**Figure R1. 1C**). Additionally, scRNA-seq data analysis revealed that

LR-DPVB treatment resulted in an upregulation of the inflammation marker *Tnfrsf1a* within the macrophage compartment of Hepa1-6 tumors (**Figure R1. 1D**). Macrophages are commonly classified as pro-inflammatory M1-like and anti-inflammatory M2-like. By assessing the gene expression levels of *Nos2* (CD86) and *Mrc1* (CD206), we briefly identified distinct populations of M1-type and M2-type macrophages of DEN+CCl₄ tumors (**Figure R1. 1E-F**). ScRNA-seq analysis revealed that LR-DPVB exhibited an upregulation of genes associated with enhanced antigen presentation ability, such as *H2k1* and *H2d1* (**Figure R1. 1G**). We utilized flow cytometry to assess the phenotype alteration of macrophages. The LR-DPVB group exhibited a significant upregulation of the M1-like macrophage phenotype in Hepa1-6 tumors, while there was a marked decrease in the accumulation of M2-like macrophages, indicating recruitment and M1-repolarization of macrophages with LR-DPVB-induced immunoactivity (**Figure R1. 1H**).

Figure R1. 1: (C) Representative and quantification of flow cytometry plots of % F4/80⁺ CD11b⁺ in live CD45⁺ cells of Hepa1-6 HCC tumor TIICs at the end of the therapeutic cycle. (D) Violin plots representing the expression of *Tnfrsf1a* in macrophage subsets of Hepa1-6 HCC tumor with indicated treatment. (E) tSNE map of the indicated marker in macrophages of DEN+CCl₄ tumor. (F) tSNE map of the M1-type and M2-type macrophages of DEN+CCl₄ tumor. (G) Violin plots representing the expression of *H2d1* and *H2k1* in M1-type tumor-associated macrophage (TAMs) of DEN+CCl₄ tumor with indicated treatment. (H) Expression of CD86 and CD206 of F4/80⁺ CD11b⁺ cells in the Hepa1-6 HCC tumor with indicated treatment. Data shown as means ± SD of three independent experiments. * $P < 0.05$, ** $P < 0.01$, *** $P < 0.001$ and ns. for no significance.

Meanwhile, we found different status of dendritic cells (DCs) of scRNA-seq data from

DEN+CCI₄ tumor, which could be further annotated as Type 1 conventional dendritic cells (cDC1), cDC2, mature DC^{3,4} (**Figure R1. 1I-J**). Notably, the infiltration of cDC1 and mature DCs was significantly enhanced in tumors treated with LR-DPVB and LDRT, compared to the DPVB-treated tumors and CON group tumor, respectively (**Figure R1. 1K**). Volcano plot revealed that LR-DPVB exhibited an upregulation of genes associated with major histocompatibility complex-I (MHC-I) presentation, such as *Psm5* and *H2d1*, both in DCs and mature DC (**Figure R1. 1L-M**). Flow cytometry analysis confirmed the upregulation of antigen-presenting markers CD86 and major histocompatibility complex-I (MHC-I) molecules in tumor-infiltrating DCs of LR-DPVB treated group compared to the DPVB-treated group (**Figure R1. 1N**), indicating enhanced maturation and antigen-presenting function. Preclinical mouse models have consistently demonstrated the essential role of cDC1 in T cell-mediated tumor regression and therapeutic responses to immune checkpoint blockade (ICB)⁵⁻⁷. Moreover, Jan P. Böttcher etc. recently reported that cDC1-CD8⁺ T cell clusters serve as niches for activation of stem-like CD8⁺ T_{pe} cells⁸. Therefore, further investigation is warranted to elucidate the interaction between cDC1 cells and CD8 T cells following LDRT. Collectively, these findings provide compelling evidence supporting the significant enhancement of pro-inflammatory innate immunocyte infiltration into tumors by combining LDRT with DPVB treatment, promoting activation of the tumor's innate immune microenvironment. **Considering the integrity of the logic, the above data was not included in the revised manuscript.**

Figure R1. 1: (I) tSNE maps of scRNAseq data from DCs of DEN+CCI₄ tumor (n=3 per group). (J) DCs expressing indicated genes across major cell populations and their corresponding average expression (size of dot indicates the percentage of cells in each population; expression intensity is indicated by color). (K) Fold-change of DCs populations from LR-DPVB and DPVB groups (left panel) and LDRT and CON groups (right panel) in DEN+CCI₄ tumor. (L-M) Violin plots representing the expression of *Psm5* and *H2d1* in DC (L) and mature DC (M) of DEN+CCI₄ tumor with indicated treatment. (N) Expression of CD86 and MHC-I of CD11c⁺ cells in the Hepa1-6 HCC tumor with indicated treatment. Data shown as means ± SD of three independent experiments. * $P < 0.05$, ** $P < 0.01$, *** $P < 0.001$ and ns. for no significance.

References:

- Herrera, F. G., Ronet, C., Ochoa de Olza, M., Barras, D., Crespo, I., Andreatta, M., Corria-Osorio, J., Spill, A., Benedetti, F., Genolet, R., Orcurto, A., Imbimbo, M., Ghisoni, E., Navarro Rodrigo, B., Berthold, D. R., Sarivalasis, A., Zaman, K., Duran, R., Dromain, C., Prior, J., Schaefer, N., Bourhis, J., Dimopoulou, G., Tsourti, Z., Messemaker, M., Smith, T., Warren, S. E., Foukas, P., Rusakiewicz, S., Pittet, M. J., Zimmermann, S., Sempoux, C., Dafni, U., Harari, A., Kandalaf, L. E., Carmona, S. J., Dangaj Laniti, D., Irving, M. & Coukos, G. Low Dose Radiotherapy Reverses Tumor Immune Desertification and Resistance to Immunotherapy. *Cancer Discov* (2021).
- Nishiga, Y., Drainas, A. P., Baron, M., Bhattacharya, D., Barkal, A. A., Ahrari, Y., Mancusi, R., Ross, J. B., Takahashi, N., Thomas, A., Diehn, M., Weissman, I. L., Graves, E. E. & Sage, J. Radiotherapy in combination with CD47 blockade elicits a macrophage-mediated abscopal effect. *Nat Cancer* **3**, 1351-1366 (2022).
- Bosteels, V., Maréchal, S., De Nolf, C., Rennen, S., Maelfait, J., Tavernier, S. J., Veters, J., Van De Velde, E., Fayazpour, F., Deswarte, K., Lamoot, A., Van Duyse, J., Martens, L., Bosteels, C., Roelandt, R., Emmaneel, A., Van Gassen, S., Boon, L., Van Isterdael, G., Guillas, I., Vandamme, N., Höglinger, D., De Geest, B. G., Le Goff, W.,

- Saeys, Y., Ravichandran, K. S., Lambrecht, B. N. & Janssens, S. LXR signaling controls homeostatic dendritic cell maturation. *Sci Immunol* **8**, eadd3955 (2023).
4. Alam, J., Yazdanpanah, G., Ratnapriya, R., Borchering, N., de Paiva, C. S., Li, D. & Pflugfelder, S. C. Single-cell transcriptional profiling of murine conjunctival immune cells reveals distinct populations expressing homeostatic and regulatory genes. *Mucosal Immunol* **15**, 620-628 (2022).
5. Böttcher, J. P., Bonavita, E., Chakravarty, P., Blees, H., Cabeza-Cabrerizo, M., Sammicheli, S., Rogers, N. C., Sahai, E., Zelenay, S. & Reis e Sousa, C. NK Cells Stimulate Recruitment of cDC1 into the Tumor Microenvironment Promoting Cancer Immune Control. *Cell* **172** (2018).
6. Broz, M. L., Binnewies, M., Boldajipour, B., Nelson, A. E., Pollack, J. L., Erle, D. J., Barczak, A., Rosenblum, M. D., Daud, A., Barber, D. L., Amigorena, S., Van't Veer, L. J., Sperling, A. I., Wolf, D. M. & Krummel, M. F. Dissecting the Tumor Myeloid Compartment Reveals Rare Activating Antigen-Presenting Cells Critical for T Cell Immunity. *Cancer Cell* **26**, 938 (2014).
7. Hildner, K., Edelson, B. T., Purtha, W. E., Diamond, M., Matsushita, H., Kohyama, M., Calderon, B., Schraml, B. U., Unanue, E. R., Diamond, M. S., Schreiber, R. D., Murphy, T. L. & Murphy, K. M. Batf3 deficiency reveals a critical role for CD8alpha+ dendritic cells in cytotoxic T cell immunity. *Science* **322**, 1097-1100 (2008).
8. Meiser, P., Knolle, M. A., Hirschberger, A., de Almeida, G. P., Bayerl, F., Lacher, S., Pedde, A.-M., Flommersfeld, S., Hönninger, J., Stark, L., Stögbauer, F., Anton, M., Wirth, M., Wohlleber, D., Steiger, K., Buchholz, V. R., Wollenberg, B., Zielinski, C. E., Braren, R., Rueckert, D., Knolle, P. A., Kaissis, G. & Böttcher, J. P. A distinct stimulatory cDC1 subpopulation amplifies CD8+ T cell responses in tumors for protective anti-cancer immunity. *Cancer Cell* **41** (2023).

6) Lack of consistencies in the data.

e.g. Figure 2 compared the single cell data, flow data, IHC data on CD8⁺ T cells with control-treated and LR-DPVB treated mice. Figure 3 compared DPVB and LR-DPVB. Figure 4 switched back to control and LR-DPVB.

Response: We do appreciate the reviewer's comments. To these concerns, we response as follows. In fact, we conducted experiments on four distinct groups to investigate the effects and cellular mechanisms of LR-DPVB. However, in order to elucidate the mechanism underlying LR-DPVB treatment, we have emphasized the corresponding groups in specific figures within our previously submitted manuscript.

In details, we initially aimed to identify the primary cell type responsible for the effects of LR-DPVB by comparing changes in cellular states between LR-DPVB and untreated (CON) groups. Our findings indicate that CD8⁺ T cells play a pivotal role in mediating the effects of LR-DPVB (**Fig. 2**). Then, we compared the differential expression of genes (DEGs) and CD8 subsets between LR-DPVB and DPVB to elucidate the mechanism by which LDRT sensitizes DPVB treatment, revealing that enhanced effector and cytotoxic function in CD8⁺ T cells mediated the primary

antitumor response in LR-DPVB (**Fig. 3**). Finally, to investigate the cellular changes following LDRT, we further compared CD8 subsets between LDRT and control groups. Our analysis and experiments revealed that the progenitor exhausted CD8⁺ T cells with enhanced DPVB-mediated antitumor effects was present in LDRT-treated subjects (**Fig. 4**).

Considering the data coherence, we have incorporated both experimental results and scRNA-seq analysis of four distinct groups in the revised manuscript.

7) Number of T cells/ mass of the tumors should be used to present the T cell amount instead %.

Response: We thank the reviewer's advice and this point has been well taken. **The number of T cells/ mass of the tumors have been incorporated in the revised manuscript (Fig. 2E).**

8) Number of cells used for the scRNA analysis should be indicated. How many T cells were analyzed in each group?

Response: A total of 6,913 cells were utilized for scRNA-seq analysis. The number of T cells in the Hepa1-6 HCC tumors analyzed by scRNA-seq was 433, 252, 549 and 547 for the CON, LDRT, DPVB, and LR-DPVB groups respectively. Despite the limited cell numbers used in scRNA-seq analysis, we validated our findings through repeatedly flow cytometry experiments guided by data clues provided by scRNA-seq.

Additionally, we further analyzed CD45⁺ tumor-infiltrating immune cells (TIICs) of four treatment groups (CON, LDRT, DPVB, LR-DPVB) in DEN+CCl₄ HCC mice model by scRNA-seq in revised manuscript (**Fig. 2I**). To enhance the accuracy of information capture of TIICs, CD45⁺ TIICs were sorted by CD45 Positive Selection Kit for subsequent scRNA-seq and two sequencing libraries were constructed. The number of immune cells used for scRNA-seq analysis of CD45⁺ TIICs in DEN+CCl₄ HCC mice tumors was 23,150 and the number of T cells analyzed by scRNA-seq in CON, LDRT, DPVB and LR-DPVB groups was 297, 380, 1048 and 635 respectively. **We have added the above contents in the Methods of revised manuscript.**

Minor:

9) It is unclear how PDTF works. PDTF does not seem to be a standardized method to test the response of drugs.

Response: We apologize for the lack of clarity regarding this point in our initial manuscript submission. The patient-derived tumor fragment (PDTF) is a valuable platform for investigating the early immunological response of human tumor tissue to *ex vivo* drug treatment, as it preserves both the tumor microenvironment and architecture ¹. Furthermore, changes in the early immunological profile of PDTFs are strongly correlated with clinical outcomes in corresponding patients. This novel approach has been utilized to assess the early immunological response of various treatments, including immune checkpoint blockade, therapeutic nanoparticles, and anti-inflammatory drugs ¹⁻³. In our study, we employed PDTFs to investigate the correlation between the accumulation of stem-like CD8⁺ T_{PEX} in tumors and the early efficacy of T+A therapy in HCC patients. **This information has been incorporated into the revised manuscript.**

References:

1. Voabil, P., de Bruijn, M., Roelofsen, L. M., Hendriks, S. H., Brokamp, S., van den Braber, M., Broeks, A., Sanders, J., Herzig, P., Zippelius, A., Blank, C. U., Hartemink, K. J., Monkhorst, K., Haanen, J., Schumacher, T. N. & Thommen, D. S. An *ex vivo* tumor fragment platform to dissect response to PD-1 blockade in cancer. *Nat Med* **27**, 1250-1261 (2021).
2. Sun, Y., Yang, J., Li, Y., Luo, J., Sun, J., Li, D., Wang, Y., Wang, K., Yang, L., Wu, L. & Sun, X. Single low-dose INC280-loaded theranostic nanoparticles achieve multirouted delivery for MET-targeted primary and liver metastatic NSCLC. *Mol Cancer* **21**, 212 (2022).
3. Pelly, V. S., Moeini, A., Roelofsen, L. M., Bonavita, E., Bell, C. R., Hutton, C., Blanco-Gomez, A., Banyard, A., Bromley, C. P., Flanagan, E., Chiang, S.-C., Jørgensen, C., Schumacher, T. N., Thommen, D. S. & Zelenay, S. Anti-Inflammatory Drugs Remodel the Tumor Immune Environment to Enhance Immune Checkpoint Blockade Efficacy. *Cancer Discovery* **11**, 2602-2619 (2021).

10) Discussion is a bit out of scope. Authors should stay focused and are encouraged to make the discussion more concise.

Response: We thank the reviewer for the suggestion, and the reviewer's point is well taken. **We have refined the discussion to concentrate on our findings.**

11) Methodologies are very brief for some experiments.

Response: Revised manuscript has been enriched with more detailed methodologies, in response to your valuable reminder.

12) Labeling is confusing. E.g. TCF1-PD1-HighCD8+ T cells (figure 3) and TCF1+PD1+CD8+ T

cells are the same? The former is quite confusing.

Response: We sincerely apologize for the oversight on our part. The TCF1⁺ PD-1^{high} CD8⁺ T cells depicted in **Fig. 3** are identical to those labeled as TCF1⁺ PD1⁺ CD8⁺ T cells. **Appropriate revisions have been made in the revised manuscript (Fig. 3).**

Reviewer #2 (Remarks to the Author): with expertise in HCC, cancer (immuno)therapy

This manuscript reports the anti-tumor activity of dual PD-L1 and VEGFA blockade (DPVB) with low-dose radiotherapy (LDRT) in orthotopic models of murine HCC. The study demonstrated that LDRT enhanced DPVB by enlarging intratumoral stem-like CD8⁺ T_{pex}, recruited from the dLNs via the CXCL10/CXCR3 axis. The authors also suggest that stem-like CD8⁺ T_{pex} may be biomarkers for selecting patients with HCC before T+A combination therapy and predicting treatment outcomes. While the paper is well-written, and the study provides new insights into the potential of combining immunotherapies and radiotherapy in clinical practice, I have several concerns.

Specific comments:

1. The authors used an orthotopic tumor model with murine Hepa1-6 HCC cell injection. The authors found the benefit of LR-DPVB with distinct immune profiles in this model. Additional models are needed to increase the clinical relevance of this study. The Hepa1-6 model is highly sensitive to immunotherapy and thus does not recapitulate most human HCC cases. Additionally, most HCC patients have underlying liver disease and poor liver function, and the ICB efficacy/resistance may depend on the type of liver damage. The key data reported in this manuscript must be confirmed in orthotopic or autochthonous models, ideally in ICB-resistant models with pre-existing liver injury, to show the potential to sensitize the tumor to ICBs and the correlation between the treatment efficacy and immune subset profiles and support the conclusions.

Response: We thank the reviewer's comments. This concern was also raised by reviewer #1, indicating the necessity for additional validation across different HCC models.

we conducted further verification of the therapeutic effect of LR-DPVB in two autochthonous mouse HCC models: diethylnitrosamine and repeated carbon tetrachloride induced hepatocarcinogen model (DEN+CCl₄ HCC model) (**Fig. 1E**), hydrodynamic tail vein injection *Trp53^{KO}/MYC^{OE}* somatic genome editing mediated mouse HCC model (*Trp53^{KO}/MYC^{OE}* HCC

model) (**Fig. 1K**). The DEN + CCl₄ HCC model was established by a single injection of DEN (25mg/kg) followed by repeated administration of CCl₄ (20%, 1μl/g, twice a week), thereby recapitulating genotoxic injury and advanced fibrosis-associated hepatocellular carcinoma (HCC) in humans. Furthermore, immunohistochemistry staining (IHC) revealed a scarcity of CD8⁺ T cell infiltrates in the liver tumors of the CON group from the DEN+CCl₄ HCC model (**Fig. 1I**), indicating that this model represents an immune desert environment resembling "cold" human HCC with limited responsiveness to immunotherapy. We confirmed that treatment with LR-DPVB resulted in less advanced liver lesions in the DEN+CCl₄ HCC model, as evidenced by improvements in gross appearance, liver weight, tumor nodule numbers, and H&E staining (**Fig. 1F-J**). To validate this observation of LR-DPVB efficacy and established the clinical relevance between efficacy and key driver genes of human HCC, we took advantage of hydrodynamic tail-vein injections (HDTV_i) to generate a *Trp53*^{KO}/*MYC*^{OE} HCC model in which oncogenic *MYC* can be genomically integrated, and that *Trp53* is deficient to recapitulate the features of HCC, as previously described¹⁻³. We observed more therapeutic benefits from LR-DPVB compared to DPVB in *Trp53*^{KO}/*MYC*^{OE} HCC model which is known to be an immune dessert "cold" model with few CD8⁺ T cell infiltrates and ICB resistance reported in previous studies¹ (**Fig. 1L-P**). Furthermore, there were no significant differences observed in mouse circulating liver enzymes alanine transaminase (ALT) and aspartate aminotransferase (AST) between the LR-DPVB group and other groups in the same model, which indicates minimal increase in toxicity to the liver function under long-term treatment with LR-DPVB (**Supplementary Fig. 1G-J**).

To further explore the correlation between the efficacy of LR-DPVB treatment and immune subset profiles in the mice HCC models with pre-existing liver injury, we further sorted CD45⁺ tumor-infiltrating lymphocytes (TILs) from four groups (CON, LDRT, DPVB, LR-DPVB) of the DEN+CCl₄ HCC model for scRNA-seq analysis and performed repeatedly flow cytometry assays both in DEN+CCl₄ HCC model and *Trp53*^{KO}/*MYC*^{OE} HCC model (**Fig. 2H-M**). By comparing the proportion of immune cells, we found that CD8 T cells was also the cell population with the most obvious change after LR-DPVB treatment in DEN+CCl₄ model (**Fig. 2H-I**). Consistently, flow cytometry, IHC analysis and T cells depletion experiment indicated that the CD8⁺ T cells did exert a significant impact on tumor progression (**Fig. 1I-J, O-P, 2J-M, Supplementary Fig. 3B-E**). We further analyzed the alterations in T cell subsets of DEN+CCl₄ tumors treated with LR-DPVB. The

scRNA-seq analysis and flow cytometry analysis confirmed a significant enrichment of expands the transitory effected CD8⁺ T cells (CD8_Tef) featured a dramatic enrichment of effector function and cytolytic capacity of LR-DPVB group in DEN+CCl₄ HCC model and *Trp53^{KO}/MYC^{OE}* HCC model (**Supplementary Fig. 5, 6E-K, 7B**). Flow cytometry and scRNA-seq analysis revealed a significant enrichment of the progenitor exhausted CD8⁺ T cells in tumor after LDRT treatment (**Fig. 4, Supplementary Fig. 9**). The use of peripheral lymphoid drainage blockade in the DEN+CCl₄ HCC model and CXCR3 depletion in the *Trp53^{KO}/MYC^{OE}* HCC model confirmed that migration of CD8_Tpex from the draining lymph nodes (dLNs) to the tumor occurs through the CXCL10/CXCR3 axis after LDRT treatment (**Supplementary Fig. 9**).

Overall, our results emphasize the synergistic treatment efficacy of LR-DPVB across various HCC mouse models and confirmed the correlation between the treatment efficacy and immune subset profiles. **The aforementioned results have been incorporated into the revised manuscript.**

References:

1. Yuen, V. W.-H., Chiu, D. K.-C., Law, C.-T., Cheu, J. W.-S., Chan, C. Y.-K., Wong, B. P.-Y., Goh, C.-C., Zhang, M. S., Xue, H. D.-G., Tse, A. P.-W., Zhang, Y., Lau, H. Y.-H., Lee, D., Au-Yeung, R. K. H., Wong, C.-M. & Wong, C. C.-L. Using mouse liver cancer models based on somatic genome editing to predict immune checkpoint inhibitor responses. *J Hepatol* **78**, 376-389 (2023).
2. Lee, D., Xu, I. M.-J., Chiu, D. K.-C., Leibold, J., Tse, A. P.-W., Bao, M. H.-R., Yuen, V. W.-H., Chan, C. Y.-K., Lai, R. K.-H., Chin, D. W.-C., Chan, D. F.-F., Cheung, T.-T., Chok, S.-H., Wong, C.-M., Lowe, S. W., Ng, I. O.-L. & Wong, C. C.-L. Induction of Oxidative Stress Through Inhibition of Thioredoxin Reductase 1 Is an Effective Therapeutic Approach for Hepatocellular Carcinoma. *Hepatology* **69**, 1768-1786 (2019).
3. Chiu, D. K.-C., Yuen, V. W.-H., Cheu, J. W.-S., Wei, L. L., Ting, V., Fehlings, M., Sumatoh, H., Nardin, A., Newell, E. W., Ng, I. O.-L., Yau, T. C.-C., Wong, C.-M. & Wong, C. C.-L. Hepatocellular Carcinoma Cells Up-regulate PVRL1, Stabilizing PVR and Inhibiting the Cytotoxic T-Cell Response via TIGIT to Mediate Tumor Resistance to PD1 Inhibitors in Mice. *Gastroenterology* **159**, 609-623 (2020).

2. Moreover, I suggest a comparison with DPVB in addition to the comparison with LR-DPVB and the control group. This would allow for a more comprehensive analysis of the additive effect of combining low-dose radiation with DPVB therapy and elucidate the mechanisms of action.

Response: We thank the reviewer for the valuable comment, which highlights an important point. In the previously submitted manuscript, we identified that CD8⁺ T cells is the most crucial cell group in combination therapy based on the comparison between LR-DPVB and CON group. Further analysis of tumor-infiltrating T cell subpopulations revealed that exhausted CD8⁺ T exhibiting effector function and cytotoxicity are pivotal for enhancing the tumor rejection capacity of LR-

DPVB.

As recommended by the reviewer, we conducted a comparison of changes in cell subsets among tumor-infiltrating immune cells (TIICs) between LR-DPVB and DPVB groups. Our analysis revealed that macrophages exhibited the most significant alterations (**Figure R2. 1A**). Flow cytometry also showed that LR-DPVB promoted the recruitment of macrophages into the tumor comparing with DPVB alone (**Figure R2. 1B**). Additionally, scRNA-seq analysis demonstrated upregulation of *Tnfrsf1a*, an inflammation marker, in macrophages following LR-DPVB treatment of Hepa1-6 tumor (**Figure R2. 1C**). Macrophages are commonly classified as pro-inflammatory M1-like and anti-inflammatory M2-like. By assessing the gene expression levels of *Nos2* (CD86) and *Mrc1* (CD206), we briefly identified distinct populations of M1-type and M2-type macrophages of DEN+CCl₄ tumors (**Figure R2. 1D-E**). Single-cell RNA sequencing analysis revealed that LR-DPVB exhibited an upregulation of genes associated with enhanced antigen presentation ability, such as *H2k1* and *H2d1* (**Figure R2. 1F**). Furthermore, the flow cytometry showed a significant increase in M1-like macrophage phenotype in Hepa1-6 tumors from the LR-DPVB group, while there was a notable reduction in M2-like macrophage accumulation. These results suggested that there was recruitment and repolarization of macrophages with decreased immunosuppressive capacity after LR-DPVB (**Figure R2. 1G**).

Figure R2. 1: (A) Fold-change of TIICs populations from LR-DPVB and DPVB groups in Hepa1-6 HCC tumor. (B) Representative and quantification of flow cytometry plots of % F4/80⁺ CD11b⁺ in live CD45⁺ cells of Hepa1-6 HCC tumor TIICs at the end of the therapeutic cycle. (C) Violin plots representing the expression of *Tnfrsf1a* in macrophage subsets of Hepa1-6 HCC tumor with indicated treatment. (D) tSNE map of the indicated marker in macrophages of DEN+CCl₄ tumor. (E) tSNE map of the M1-type and M2-type macrophages of DEN+CCl₄ tumor. (F) Violin plots representing the expression of *H2d1* and *H2k1* in M1-type tumor-associated macrophage (TAMs) of DEN+CCl₄ tumor with indicated treatment. (G) Expression of CD86 and CD206 of F4/80⁺ CD11b⁺ cells in the Hepa1-6 HCC tumor with indicated treatment. Data shown as means ± SD of three independent experiments. * $P < 0.05$, ** $P < 0.01$, *** $P < 0.001$ and ns. for no significance.

Collectively, our findings reveal that LDRT promoted the intra-tumoral recruitment and M1-phenotype polarization of macrophages in DPVB treatment, which contributed to the inflammatory remodeling of tumor immune microenvironment. **Considering the logical coherence, the above data was not included in the revised manuscript.**

3. Regarding the scRNAseq data analysis presented in Figs. 2, 4, and 5, it would be interesting to examine the changes in immune subsets other than CD8T cells, such as B cell and macrophage clusters, which may significantly affect the tumor immune microenvironment (B cells also showed significant change in Supplementary Fig. 2D). Previous studies have reported that LDRT can reshape the immune microenvironment. For example, it would be important to determine whether these changes indicate a reduction in immunosuppressive subsets, increased T-cell activation, or both. A previous study by Herrera et al. found that LDRT treatment reprogrammed the tumor microenvironment of tumors with scarce immune infiltration. And LDRT combination with immunotherapy induced simultaneous mobilization of innate and adaptive immunity, including CD4⁺ effector T cells, CD8⁺ T cells, and myeloid compartments, such as macrophages, DCs, and monocytes (PMID: 34479871).

Response: We thank the reviewer for the comments. According to the suggestion of the reviewer, we conducted further investigations into the alterations in subsets of tumor infiltrating immune cells (TIICs) beyond CD8⁺ T cells. Through comparisons between LR-DPVB and DPVB, as well as LR-DPVB and CON, we have discovered that macrophages also play a crucial role in the effects of LR-DPVB, in addition to CD8⁺ T cells (**Figure 2C, Figure R2. 1A**).

As shown in **Figure R2. 1A-G** and **Figure R2. 2A**, our study revealed that LDRT significantly recruited macrophages, promoted M1-type inflammatory changes and antigen presentation ability of macrophages. Meanwhile, through analysis of different status of dendritic cells (DCs) from

DEN+CCl₄ tumor, we found the LDRT-treated tumors exhibited a predominant infiltration of cDC1 and mature DC (**Figure R2. 2B-D**). Volcano plot revealed that LDRT exhibited an upregulation of genes involved in major histocompatibility complex-I (MHC-I) presentation, such as *Psm5* and *H2d1*, both in DCs and mature DC (**Figure R2. 2E-F**). Flow cytometry analysis confirmed the LDRT-mediated upregulation of antigen-presenting markers CD86 and MHC-I molecules on tumor-infiltrating DCs (**Figure R2. 2G**), indicating enhanced maturation and antigen-presenting function of DCs. Compared with DPVB treatment alone, LDRT combined with DPVB synergistically enhanced above functions of macrophages and DCs (**Figure R2. 1-2**). **Considering the integrity of the logic, the data refer to the macrophages and DCs alteration after the LR-DPVB was not included in the revised manuscript.**

Additionally, during the investigation of the underlying mechanism behind LDRT-induced recruitment of stem-like CD8⁺ Tpex, we discovered that myeloid antigen-presenting cells (APCs), including macrophages and DCs, predominantly secreted CXCL10 to facilitate the recruitment of CXCR3⁺ CD8⁺ Tpex to the tumor site following LDRT (see the **specific comments #8, Supplementary Fig. 9J-K, 10H-I**). The CD8⁺ Tpex further sensitized DPVB therapy. The collective findings of this study suggest that the concurrent activation of innate and adaptive immunity through the combination of LDRT and DPVB effectively impedes tumor progression.

Interestingly, we observed a significant increase in the proportion of cDC1 infiltration following LDRT (**Figure R2. 2D**). Pre-clinical models have consistently demonstrated the essential role of cDC1 in T cell-mediated tumor regression and therapeutic responses to immune checkpoint blockade (ICB) ¹⁻³. Moreover, *Jan P. Böttcher* Et al. recently reported that cDC1-CD8⁺ T cell clusters serve as niches for activation of stem-like CD8⁺ Tpex cells ⁴. Therefore, further investigation into the interaction between cDC1 cells and CD8⁺ Tpex cells subsequent to LDRT holds significant promise.

Figure R2. 2: (A) Fold-change of TIICs populations from LDRT and CON groups in Hepa1-6 HCC tumor. (B) tSNE maps of scRNAseq data from DCs of DEN+CCI₄ tumor (n=3 per group). (C) DCs expressing indicated genes across major cell populations and their corresponding average expression (size of dot indicates the percentage of cells in each population; expression intensity is indicated by color). (D) Fold-change of DCs populations from LR-DPVB and DPVB groups (left panel) and LDRT and CON groups (right panel) in DEN+CCI₄ tumor. (E-F) Violin plots representing the expression of *Psm5* and *H2d1* in DC (E) and mature DC (F) of DEN+CCI₄ tumor with indicated treatment. (G) Expression of CD86 and MHC-I of CD11c⁺ cells in the Hepa1-6 HCC tumor with indicated treatment. Data shown as means \pm SD of three independent experiments. * $P < 0.05$, ** $P < 0.01$, *** $P < 0.001$ and ns. for no significance.

Notably, our study primarily focused on elucidating changes in myeloid APCs and T cells after LDRT treatment. Although a significant decrease in the proportion of B cells was observed in the LR-DPVB group, the specific role and underlying mechanism behind this phenomenon remain unclear. Hence, additional research regarding B cells is warranted in future studies.

References:

- Böttcher, J. P., Bonavita, E., Chakravarty, P., Brees, H., Cabeza-Cabrerizo, M., Sammicheli, S., Rogers, N. C., Sahai, E., Zelenay, S. & Reis e Sousa, C. NK Cells Stimulate Recruitment of cDC1 into the Tumor Microenvironment Promoting Cancer Immune Control. *Cell* **172** (2018).
- Broz, M. L., Binnewies, M., Boldajipour, B., Nelson, A. E., Pollack, J. L., Erle, D. J., Barczak, A., Rosenblum,

M. D., Daud, A., Barber, D. L., Amigorena, S., Van't Veer, L. J., Sperling, A. I., Wolf, D. M. & Krummel, M. F. Dissecting the Tumor Myeloid Compartment Reveals Rare Activating Antigen-Presenting Cells Critical for T Cell Immunity. *Cancer Cell* **26**, 938 (2014).

3. Hildner, K., Edelson, B. T., Purtha, W. E., Diamond, M., Matsushita, H., Kohyama, M., Calderon, B., Schraml, B. U., Unanue, E. R., Diamond, M. S., Schreiber, R. D., Murphy, T. L. & Murphy, K. M. Batf3 deficiency reveals a critical role for CD8alpha+ dendritic cells in cytotoxic T cell immunity. *Science* **322**, 1097-1100 (2008).

4. Meiser, P., Knolle, M. A., Hirschberger, A., de Almeida, G. P., Bayerl, F., Lacher, S., Pedde, A.-M., Flommersfeld, S., Hönninger, J., Stark, L., Stögbauer, F., Anton, M., Wirth, M., Wohlleber, D., Steiger, K., Buchholz, V. R., Wollenberg, B., Zielinski, C. E., Braren, R., Rueckert, D., Knolle, P. A., Kaissis, G. & Böttcher, J. P. A distinct stimulatory cDC1 subpopulation amplifies CD8+ T cell responses in tumors for protective anti-cancer immunity. *Cancer Cell* **41** (2023).

4. scRNA sequencing analysis revealed that CD8_Tex was the enriched T cell subset after LR-DPVB treatment compared to DPVB-treated tumors. The authors stated that CD8_Tex from the LR-DPVB-treated tumors expressed higher levels of effector function and cytolytic capacity-related molecules, such as *Ifng*, *Tnf*, and *Gzmb*. However, Supplementary Fig. 3D shows no statistically significant difference in the expression level of these molecules in CD8_Tex between LR-DPVB and DPVB treatment groups. Instead, the statistically significant difference in the expression levels of *Ifng* and *Gzmb* was in the CD8_T subset, as shown in Fig. 3D. Therefore, the conclusion that "the enhancement of the tumor-rejecting effect after LR-DPVB treatment was largely dependent on the magnitude of intratumoral CD8+ Tex, which featured potent effector function and cytolytic capacity" needs additional evidence.

Response: We appreciate the reviewer's comment and apologize for the writing error regarding the *P* value in previous **Supplementary Fig. 3D**. The correct corresponding *P* values of effector function and cytolytic capacity-related molecules (*Tnf*, *Gzmb*, and *Ifng*) in CD8_Tex between LR-DPVB and DPVB were 0.028, 0.38, and 0.79 respectively. **We have corrected above *P* values in revised manuscript (Fig. 3D).**

As for the reviewer's concerns of the non-statistically significant *P* value of the expression level of the effector function and cytolytic capacity-related molecules in CD8_Tex between LR-DPVB and DPVB treatment groups, we respond as follows. scRNA-seq offers distinct advantages in elucidating potential insights at the individual cell level. Despite the higher resolution of cellular differences provided by scRNA-seq compared to bulk RNA-seq, current consensus suggests that high-throughput scRNA-seq data still have suboptimal information capture rates and suffer from

extreme sparsity and variability due to frequent drop-out events during sequencing¹⁻³. During the scientific research practice, scRNA-seq findings often necessitate subsequent validation across RNA-protein levels using a range of experimental approaches, including qPCR, flow cytometry, immunofluorescence, ect. This is why current studies often incorporate a discovery group, where scRNA-seq identifies differences, and a validation group, where other sequencing or experimental methods confirm these differences^{4,5}. Therefore, in our study, we employed scRNA-seq analysis as a clue and conducted repeatedly flow cytometry experiments to further validate the differential expression of cytotoxic-related molecules in CD8⁺ Tex cells across different groups. The flow cytometry assays revealed that LR-DPVB treatment significantly increased the effector function and cytolytic capacity-related molecules of TCF1⁻ PD-1⁺ CD8⁺ T cells compared to DPVB (**Fig. 3I, Supplementary Fig. 7A**). These findings suggest that the enhanced tumor-rejecting effect observed after LR-DPVB treatment is largely attributed to the potent intra-tumoral exhausted CD8⁺ T cell population with heightened effector function and cytolytic capacity.

References:

1. He, Y., Chen, X., Tu, N. H. & Luo, J. Deep Multi-Constraint Soft Clustering Analysis for Single-Cell RNA-Seq Data via Zero-Inflated Autoencoder Embedding. *IEEE/ACM Trans Comput Biol Bioinform* **20**, 2254-2265 (2023).
2. Li, H., Xiao, X., Wu, X., Ye, L. & Ji, G. scLINE: A multi-network integration framework based on network embedding for representation of single-cell RNA-seq data. *J Biomed Inform* **122**, 103899 (2021).
3. Macosko, E. Z., Basu, A., Satija, R., Nemesh, J., Shekhar, K., Goldman, M., Tirosh, I., Bialas, A. R., Kamitaki, N., Martersteck, E. M., Trombetta, J. J., Weitz, D. A., Sanes, J. R., Shalek, A. K., Regev, A. & McCarroll, S. A. Highly Parallel Genome-wide Expression Profiling of Individual Cells Using Nanoliter Droplets. *Cell* **161**, 1202-1214 (2015).
4. Chen, S., Huang, C., Liao, G., Sun, H., Xie, Y., Liao, C., Wang, J., He, M., Hu, H., Dai, Z., Ren, X., Zeng, X., Lin, Z., Zhang, G.-P., Xie, W., Shen, S., Li, S., Peng, S., Kuang, D.-M., Zhao, Q., Duda, D. G. & Kuang, M. Distinct single-cell immune ecosystems distinguish true and de novo HBV-related hepatocellular carcinoma recurrences. *Gut* (2023).
5. Wang, X., Zha, H., Wu, W., Yuan, T., Xie, S., Jin, Z., Long, H., Yang, F., Wang, Z., Zhang, A., Gao, J., Jiang, Y., Wang, L., Hu, C., Wan, Y. Y., Li, Q.-J., Symonds, A. L. J., Jia, Q. & Zhu, B. CD200⁺ cytotoxic T lymphocytes in the tumor microenvironment are crucial for efficacious anti-PD-1/PD-L1 therapy. *Sci Transl Med* **15**, eabn5029 (2023).

5. In Fig. 4, the authors demonstrate that SLAMF6⁺PD-1⁺CD8 T cells include more GzmB-positive cells than SLAMF6⁻PD-1⁺CD8 T cells. It would be important to examine the cytokine production capacities of these subsets, specifically depending on the expression of Tim3, as previous reports have shown that this can vary.

Response: We thank the reviewer for the comment. In fact, our initial manuscript demonstrated that adoptive transfer of SLAMF6⁺ PD-1⁺ CD8⁺ T cells, as opposed to SLAMF6⁻ PD-1⁺ CD8⁺ T cells, significantly enhanced DPVB-mediated tumor regression by augmenting the cytotoxic capacity of CD8⁺ T cells (**Fig. 4C-H**). The aim of our study was to demonstrate that SLAMF6⁺ PD-1⁺ CD8⁺ T cells, which are progenitor exhausted CD8⁺ T cells, can enhance the antitumor effect of DPVB more effectively by generating GZMB⁺ CD8⁺ T cells compared to SLAMF6⁻ PD-1⁺ CD8⁺ exhausted T cells. Our primary objective did not involve comparing the cytokine production capacities exhibited by these two subsets.

Furthermore, in accordance with the reviewer's suggestion, we conducted additional assessments of cytokine production capacities for both progenitor exhausted CD8⁺ T cells and terminal exhausted CD8⁺ T cells (distinguished by SLAMF6 and TIM3 expression¹) following DPVB treatment. These results showed the SLAMF6⁺ TIM3⁻ CD8⁺ T cells can enhance the antitumor effect of DPVB by producing more GZMB⁺ CD8⁺ T cells compared to the SLAMF6⁻ TIM3⁺ CD8⁺ T cells (**Figure R2. 3**).

Taken together, these results indicated that the progenitor exhausted CD8⁺ T cells, but not terminal exhausted CD8⁺ T cells, could significantly promote DPVB-mediated tumor regression via enhancing the cytotoxic capacity of CD8⁺ T cells.

Figure R2. 3: (A-B) Illustration of the *in vivo* adoptive transfer experiment, as described above in methods. (C) Representative image of Hepa1-6 liver orthotopic tumor at the end of the therapeutic cycles (circled by yellow lines). (D) Representative and quantification of flow cytometry plots of % Gzmb⁺ in CD8⁺ T cells of the indicated groups.

Scale bars: 100 μ m. (E) Tumor growth curves of Hepa1-6 bearing mice of the indicated days. Data shown as means \pm SD of three independent experiments. * $P < 0.05$, ** $P < 0.01$, *** $P < 0.001$ and ns. for no significance.

References:

1. Miller, B. C., Sen, D. R., Al Aboosy, R., Bi, K., Virkud, Y. V., LaFleur, M. W., Yates, K. B., Lako, A., Felt, K., Naik, G. S., Manos, M., Gjini, E., Kuchroo, J. R., Ishizuka, J. J., Collier, J. L., Griffin, G. K., Maleri, S., Comstock, D. E., Weiss, S. A., Brown, F. D., Panda, A., Zimmer, M. D., Manguso, R. T., Hodi, F. S., Rodig, S. J., Sharpe, A. H. & Haining, W. N. Subsets of exhausted CD8 T cells differentially mediate tumor control and respond to checkpoint blockade. *Nat Immunol* **20**, 326-336 (2019).

6. It would be interesting to present the durability of responses for the survival and tumor volume curves shown in Fig. 1. Additionally, from the perspective of clinical relevance, it would be helpful to describe the subsequent toxicity evaluation (e.g., weight loss, liver enzymes) after day 7.

Response: We appreciate the reviewer's suggestion. In our previous research, the Kaplan-Meier survival curves of four groups in the Hepa1-6 HCC model were observed for 50 days, demonstrating sustained treatment response (**Fig. 1D**). The significant difference in survival between the DPVB and LR-DPVB groups has been well established (overall survival rate at 50 days: DPVB: 5/14 (35.7%); LR-DPVB: 12/14 (85.7%); $P < 0.05$). As suggested by the reviewer, we further extended our observation to assess the durability of responses in the four groups of Hepa1-6 HCC model up to 60 days (n=10). The results demonstrated that LR-DPVB exhibited a significantly superior survival benefit compared to DPVB (overall survival rate at 60 days: DPVB: 3/10 (30%); LR-DPVB: 8/10 (80%); $P < 0.001$; **Figure R2. 4A**). Considering that both control and LDRT groups had less than a 30% survival rate three weeks after treatment initiation, we further monitored tumor volume within a period of 21 days to evaluate longer durability of efficacy responses with LR-DPVB treatment. The results showed a significant difference between the LR-DPVB and DPVB group, indicating that LR-DPVB effectively reduced tumor progression (**Figure R2. 4B**).

As recommended by the reviewer, we assessed subsequent toxicity from a clinical relevance perspective including body weight changes and liver enzyme levels after three therapeutic cycles (**Figure R2. 4C-D**). During three cycles, there were no abnormalities observed in body weight or liver function test parameters such as alanine transaminase (ALT) and aspartate aminotransferase (AST), all values remained within normal variation ranges consistent with observations made over two therapeutic cycles.

In addition, using DEN+CCl₄ HCC model and *Trp53^{KO}/MYC^{OE}* HCC model, we investigated both

efficacy responses and subsequent toxicity evaluation after four weeks of therapeutic cycles (**Fig. 1E-P, Supplementary Fig. 1G-J**). These findings unequivocally demonstrate the sustained response and safety profile associated with LR-DPVB therapy. **The efficacy responses and subsequent toxicity evaluation of DEN+CCl₄ HCC model and *Trp53*^{KO}/*MYC*^{OE} HCC model were added in revised manuscript (Fig. 1F-J, L-P, Supplementary Fig. 1G-J).**

Figure R2. 4: (A) Kaplan–Meier curve of Hepa1-6 HCC model mice treated with 4 different regimes (n=10 per group). *P* values were determined by log-rank test. (B) Tumor growth curves of Hepa1-6 HCC model evaluated by MRI. (C) Body weight changes of Hepa1-6 HCC model mice during the three therapeutic cycles. (D) Serum levels of alanine transaminase (ALT), aspartate aminotransferase (AST) in Hepa1-6 HCC model mice were measured at the end of the third therapeutic cycles. Data shown as means \pm SD of three independent experiments. * $P < 0.05$, ** $P < 0.01$, *** $P < 0.001$ and ns. for no significance.

7. According to the hypothesis tested in this study, stem-like CD8⁺ T_{pex} are recruited following stimulation by LDRT. In the animal study presented in Fig. 1A, LDRT was administered from day 0. However, in the experiment shown in Fig. 4D, where cells were used to mimic the enhancement of CD8⁺ T_{pex} by LDRT, the cells were administered from day -8 . What is the explanation for the difference in the timing of treatment between these two experiments?

Response: We thank the reviewer's comments. To these concerns, we respond as follows. In our

animal study on LDRT combined with DPVB, the initiation time of LDRT was determined based on treatment cycle initiation after tumor formation as reported in previous radiotherapy studies ^{1,2}. To ensure effective infiltration of adoptive cells into the tumor and achieve a synergistic therapeutic effect with DPVB, cell transplantation was performed in mice two days after tumor cell implantation, following the *in vivo* adoptive cell transfer protocol previously reported by Miller, B. C ³.

References:

1. Herrera, F. G., Ronet, C., Ochoa de Olza, M., Barras, D., Crespo, I., Andreatta, M., Corria-Osorio, J., Spill, A., Benedetti, F., Genolet, R., Orcurto, A., Imbimbo, M., Ghisoni, E., Navarro Rodrigo, B., Berthold, D. R., Sarivalasis, A., Zaman, K., Duran, R., Dromain, C., Prior, J., Schaefer, N., Bourhis, J., Dimopoulou, G., Tsourti, Z., Messemaker, M., Smith, T., Warren, S. E., Foukas, P., Rusakiewicz, S., Pittet, M. J., Zimmermann, S., Sempoux, C., Dafni, U., Harari, A., Kandalaft, L. E., Carmona, S. J., Dangaj Laniti, D., Irving, M. & Coukos, G. Low Dose Radiotherapy Reverses Tumor Immune Desertification and Resistance to Immunotherapy. *Cancer Discov* (2021).
2. Sheng, H., Huang, Y., Xiao, Y., Zhu, Z., Shen, M., Zhou, P., Guo, Z., Wang, J., Wang, H., Dai, W., Zhang, W., Sun, J. & Cao, C. ATR inhibitor AZD6738 enhances the antitumor activity of radiotherapy and immune checkpoint inhibitors by potentiating the tumor immune microenvironment in hepatocellular carcinoma. *J Immunother Cancer* **8** (2020).
3. Miller, B. C., Sen, D. R., Al Aboosy, R., Bi, K., Virkud, Y. V., LaFleur, M. W., Yates, K. B., Lako, A., Felt, K., Naik, G. S., Manos, M., Gjini, E., Kuchroo, J. R., Ishizuka, J. J., Collier, J. L., Griffin, G. K., Maleri, S., Comstock, D. E., Weiss, S. A., Brown, F. D., Panda, A., Zimmer, M. D., Manguso, R. T., Hodi, F. S., Rodig, S. J., Sharpe, A. H. & Haining, W. N. Subsets of exhausted CD8 T cells differentially mediate tumor control and respond to checkpoint blockade. *Nat Immunol* **20**, 326-336 (2019).

8. The results indicate that *Cxcl9* and *Cxcl10* were significantly upregulated in LDRT-treated tumors.

What is the mechanism?

Response: We appreciate the reviewer's comment and regret for our negligence to elucidate the mechanism of LDRT-mediated intra-tumoral *Cxcl10* upregulation in our previously submitted manuscript. In our research, KEGG enrichment pathway analysis of RNA-seq data from LDRT-treated tumors and CON group tumors revealed a significant enrichment in *cytokine and cytokine receptor interaction pathway* of LDRT-treated tumor (**Fig. 5F**). Specifically, the *Cxcl9* and *Cxcl10* family genes within this pathway were significantly upregulated after LDRT treatment (**Fig. 5G**). RT-qPCR experiments confirmed the increased expression of *Cxcl10* following LDRT in both Hepa1-6 tumor and *Trp53^{KO}/MYC^{OE}* tumor (**Supplementary Fig. 9J, 10F**).

To figure out the specific mechanism underlying LDRT-mediated intra-tumoral upregulation of *Cxcl10*, we further investigated the subset of cells primarily responsible for this effect. The previous study has demonstrated that *Cxcl10* is primarily produced by antigen presenting cells (APCs),

including dendritic cells and macrophages, as well as tumor cells ¹. However, the RT-qPCR and ELISA assays did not show a significant upregulation of Cxcl10 expression in LDRT-treated Hepa1-6 cells or their supernatant (**Supplementary Fig. 10G**). While, in the *in vivo* experiments, we observed a significant increase in *Cxcl10* expression in TAMs (F4/80⁺ CD11b⁺ CD45⁺ L/D⁻) and DCs (CD11c⁺ CD45⁺ L/D⁻) cells sorted from tumor-infiltrated immune cells in the LDRT group compared to the CON group for both Hepa1-6 tumors and *Trp53^{KO}/MYC^{OE}* tumors (**Supplementary Fig. 9K, 10H**). The *in vitro* co-culture assays showed that the Cxcl10 were significantly increased in bone marrow-derived dendritic cells (BMDCs) and bone marrow-derived macrophages (BMDMs) co-cultured with the LDRT-treated tumor cells, compared to co-cultured with tumor cells that without LDRT treatment (**Supplementary Fig. 10I**). Collectively, these results suggest that myeloid APCs are the primary source of Cxcl10 upregulation following LDRT. **Corresponding results were revised in the fifth part of the Results in resubmitted manuscript (Supplementary Fig. 10).**

References:

1. Reschke, R. & Gajewski, T. F. CXCL9 and CXCL10 bring the heat to tumors. *Sci Immunol* 7, eabq6509 (2022).

[redacted]

Minor comments:

1. In Fig. 4K, the authors demonstrated a significant increase in apoptotic tumor cells in the PDTFs derived from stem-like CD8⁺ T_{pex} high infiltration HCC tissues after T+A treatment. What criteria were used to define "high" infiltration in this context?

Response: We thank the reviewer for the comment. The infiltration degree of intratumoral stem-like CD8⁺ T_{pex} was determined by dividing it with the median value of the ratio between TCF1⁺ PD1⁺ CD8⁺ T cells and total CD8⁺ T cells, as evaluated by flow cytometry in 6 HCC tissues. Greater than the median value indicates more CD8 infiltration (high infiltration), less than the median value indicates less CD8 infiltration (low infiltration). **Appropriate modifications have been made in the fourth part of Results in the revised manuscript.**

2. The figure caption for Fig 2C is missing, which makes it difficult to differentiate the different

colors representing the types of T cells.

Response: We apologized for our negligence. This point has been well taken. **The figure caption for Fig. 2C has been added.**

Reviewer #3 (Remarks to the Author): with expertise in cancer immunology, omics

The authors have done a fabulous work. But there are some significant concerns that need to be addressed.

Major comments

(1) The order in which the figures and figure panels were introduced is messed up. For example, the authors directly jumped into Fig. 3d without introducing Fig. 3a-c

Response: We thank the reminder from the reviewer and this point has been well taken. **The order of the figures and figure panels has been adjusted to align with the sequence described in the results section.**

(2) “By further analyzing of these scRNA-seq data, we also identified CD8 T cells populations featured with a higher exhaustion and effector function were inclined to accumulate after LR-DPVB in contrast to DPVB (Supplementary Fig. 3C, Fig. 3D).” Fig. 3d does not seem to support the conclusion of this sentence. In fact, Fig. 3d is poorly labeled and I do not understand what I am seeing here. Also there are not really any differences between the two groups.

Response: We are sorry for the less intuitive presentation of this part of results in the initially submitted manuscript. Considering the violin plots may not be intuitive enough to show the expression difference of the markers, we used the bubble diagram to illustrate the expression of these markers. This analysis showed the effector function-associated markers were higher expressed of CD8 T cells in LR-DPVB compared with those in DPVB (**Supplementary Fig. 4C**). **Appropriate modifications have been made to the revised manuscript.**

(3) The authors need to remove most, if not all, of their pseudotime analyses. “Pseudotime curve analysis revealed ... from LR-DPVB-treated tumors showed significant enrichment of CD8_Tef (Fig. 3G right, 3H)” Pseudotime analysis tools are known to be misleading (citation: A comparison of single-cell trajectory inference methods | Nature Biotechnology). And the authors’

results (Fig. 3gh) here indeed show that there is no extra benefit of putting their scRNA-seq data under the context of pseudotime – they can draw the same conclusion using much simpler analyses and without doing the so-called pseudotime analyses.

Response: We express our gratitude to the reviewers for their valuable suggestions. The following response is hereby provided in reply. The primary objective of pseudotime curve analysis in our study is twofold: firstly, it aims to demonstrate the differentiation relationship between CD8_Tpex and CD8_Tex; secondly, it reveals the presence of a transient effector CD8⁺ T cell population (CD8_Tef) during the differentiation process from CD8_Tpex into CD8_Tex, which plays a crucial role in LR-DPVB. While we were able to establish that CD8_Tex originates from CD8_Tpex through UMAP plots of Project_Tils analysis, we did not specifically identify the subset of transient effector CD8⁺ T cells within the reference dataset (**Fig.3A**). Meanwhile, this information is vital for further investigating the correlation between Treg/T cell subsets ratio and LR-DPVB efficacy (**Fig.3J-K**). If the plausibility of the pseudotime analysis is deemed insufficient in this study, we may consider excluding it from the manuscript.

[redacted]

(5) In this section: “7. Stem-like CD8⁺ Tpex cells and Treg/Tef ratio among T+A-treated HCC patients and all HCC patients”, the authors showed that Treg/Tef ratio is predictive/prognostic in T+A treated patients or even baseline HCC patients (Fig. 7e). Then these results suggest that their previous statement on the discovery of Treg/Tef ratio for predicting LR-DPVB treatment response is non-specific. “As our scRNA-seq analysis previously showed that Treg/CD8_Tef may be a potential curative effect index after LR-DPVB treatment (Fig. 3J-K)”. So I wouldn’t call Treg/Teff a curative effect index for LR-DPVB. It is just a general prognostic/predictive marker. Despite all the experimental and analysis results, the authors seem to have failed to really reveal anything that is specific to the synergistic effect of radiation treatment and T+A, which is the core of the whole study.

Response: We express our gratitude to the reviewers for the valuable comments. According to the comments, we offer the following response. Based on the results obtained from scRNA-seq analysis (which revealed a significant down-regulation of *Treg/Tef* ratio in LR-DPVB with therapeutic

benefit in both Hepa1-6 model and DEN-CCl₄ model (**Fig. 3J-K, Supplementary Fig. 6L**), we conducted an analysis on clinical samples and observed a significant positive correlation between Treg/Tef ratio in PBMCs of HCC patients treated with T+ A and the extent of tumor progression post-treatment. This suggests that a lower Treg/Tef ratio may be indicative of a more favorable outcome in patients undergoing T+A therapy, which is consistent with previous studies demonstrating reduced clinical benefits associated with higher Treg/Tef ratios ¹. Therefore, the Treg/Tef ratio exhibits specificity in predicting an enhanced efficacy of T+A therapy. Furthermore, we validated this correlation between Treg/Tef ratio and prognosis among overall HCC patients, where surprisingly, we found a significant association between Treg/Tef ratio and patient prognosis.

The correlation between a low Treg/Tef ratio and the therapeutic efficacy of LR-DPVB combination therapy constitutes one of the fundamental aspects of our research. The predictive value of the Treg/Tef ratio for prognosticating patients with HCC represents an extension and advancement of this study, as it is a statistically derived conclusion obtained from a large patient sample. If the reviewers deem that the inclusion of Treg/Tef ratio analysis in overall HCC prognosis impacts the elucidation of the fundamental content within the entire research, we may consider its exclusion.

References:

1. Zhu, A. X., Abbas, A. R., de Galarreta, M. R., Guan, Y., Lu, S., Koeppen, H., Zhang, W., Hsu, C.-H., He, A. R., Ryoo, B.-Y., Yau, T., Kaseb, A. O., Burgoyne, A. M., Dayyani, F., Spahn, J., Verret, W., Finn, R. S., Toh, H. C., Lujambio, A. & Wang, Y. Molecular correlates of clinical response and resistance to atezolizumab in combination with bevacizumab in advanced hepatocellular carcinoma. *Nat Med* (2022).

(6) Immunotherapy+(low dose) radiation treatment is already a very well explored idea, in many different cancer types. What is new about the authors' work?

Response: We express our gratitude for the reviewer's insightful comment regarding the innovativeness of our research. To these concerns, we respond as follows.

Although the combination of immunotherapy and radiotherapy has been extensively investigated in various malignancies, its application in hepatocellular carcinoma (HCC) is relatively restricted due to the presence of poor liver function and cirrhosis background among HCC patients ¹⁻³. Previous studies have reported the efficacy of LDRT combined with immunotherapy in a model of ovarian cancer with intraperitoneal metastasis, but the focus of the study was to explore the vital

role of LDRT, the combinatorial immunotherapy strategy was an auxiliary role to address the immune targets upregulated by LDRT (e.g., α PD1 and α CTLA4 blocking Ab to activate T cells, agonistic α CD40 Ab to activate APCs, and low-dose cyclophosphamide to attenuated Treg cells) ⁴. Our study aims to investigate the synergistic effect and mechanism of LDRT in combination with atezolizumab (anti-PD-L1) and bevacizumab (anti-VEGFA) (T+A), with a primary focus on elucidating their potential clinical applications.

T+A is the only first-line immunotherapy regimen for HCC currently, but the magnitude of its benefit remains limited ⁵. On one hand, the intricate immune microenvironment of HCC contributes to poor immune cell infiltration and non-responsiveness to T+A therapy. On the other hand, there are no effective markers available to predict treatment response, which significantly limits therapeutic outcomes in patients with HCC. Therefore, optimizing the T+A treatment regimen is urgently needed. Radiotherapy has been proposed to facilitate the conversion of “cold” tumors into “hot” ones and activate the tumor immune microenvironment ^{4,6}. Moreover, recent studies have demonstrated that low-dose 1Gy radiation with well-tolerated for patients can transiently inflame the tumor immune microenvironment ⁴. Therefore, we aim to further investigate whether LDRT enhances T+A therapy sensitivity and how it mechanistically augments the antitumor effect of T+A therapy.

In our study, we investigated the synergistic therapeutic efficacy and mechanism of LDRT combined with T+A in multiple preclinical HCC models from a clinical application perspective. We also identified stem-like CD8⁺ T_{pex} as a predictor for T+A treatment efficacy and established a new foundation for clinical selection, which was validated in samples from T+A-treated HCC patients. In terms of mechanism, we elucidated from molecular, cellular and overall levels that LDRT promotes the migration and recruitment of stem-like CD8⁺ T_{pex}, the progenitor of CD8⁺ T_{ex}, from tumor-draining lymph nodes to tumor lesions through the CXCR3 axis. The alteration of these immune cells finally enhanced the efficacy of T+A. Collectively, our findings offer insights into the underlying mechanisms that drive the synergistic effect of LDRT combined with T+A therapy, with significant clinical translational implications for both treatment and response prediction of HCC patients.

References:

1. Nabavizadeh, N., Waller, J. G., Fain, R., Chen, Y., Degnin, C. R., Elliott, D. A., Mullins, B. T., Patel, I. A., Dyer,

- B. A., Fakhoury, K., Naugler, W. E., Farsad, K., Tanyi, J. A., Fuss, M., Thomas, C. R. & Hung, A. Y. Safety and Efficacy of Accelerated Hypofractionation and Stereotactic Body Radiation Therapy for Hepatocellular Carcinoma Patients With Varying Degrees of Hepatic Impairment. *Int J Radiat Oncol Biol Phys* **100**, 577-585 (2018).
2. Bae, S. H., Park, H. C., Yoon, W. S., Yoon, S. M., Jung, I.-H., Lee, I. J., Kim, J. W., Seong, J., Kim, T. H., Nam, T.-K., Choi, Y., Lee, S. Y., Jang, H. S., Lee, D. S. & Kim, J. H. Treatment Outcome after Fractionated Conformal Radiotherapy for Hepatocellular Carcinoma in Patients with Child-Pugh Classification B in Korea (KROG 16-05). *Cancer Res Treat* **51**, 1589-1599 (2019).
3. Liang, S.-X., Zhu, X.-D., Xu, Z.-Y., Zhu, J., Zhao, J.-D., Lu, H.-J., Yang, Y.-L., Chen, L., Wang, A.-Y., Fu, X.-L. & Jiang, G.-L. Radiation-induced liver disease in three-dimensional conformal radiation therapy for primary liver carcinoma: the risk factors and hepatic radiation tolerance. *Int J Radiat Oncol Biol Phys* **65**, 426-434 (2006).
4. Herrera, F. G., Ronet, C., Ochoa de Olza, M., Barras, D., Crespo, I., Andreatta, M., Corria-Osorio, J., Spill, A., Benedetti, F., Genolet, R., Orcurto, A., Imbimbo, M., Ghisoni, E., Navarro Rodrigo, B., Berthold, D. R., Sarivalasis, A., Zaman, K., Duran, R., Dromain, C., Prior, J., Schaefer, N., Bourhis, J., Dimopoulou, G., Tsourti, Z., Messemaker, M., Smith, T., Warren, S. E., Foukas, P., Rusakiewicz, S., Pittet, M. J., Zimmermann, S., Sempoux, C., Dafni, U., Harari, A., Kandalaft, L. E., Carmona, S. J., Dangaj Laniti, D., Irving, M. & Coukos, G. Low Dose Radiotherapy Reverses Tumor Immune Desertification and Resistance to Immunotherapy. *Cancer Discov* (2021).
5. Finn, R. S., Qin, S., Ikeda, M., Galle, P. R., Ducreux, M., Kim, T.-Y., Kudo, M., Breder, V., Merle, P., Kaseb, A. O., Li, D., Verret, W., Xu, D.-Z., Hernandez, S., Liu, J., Huang, C., Mulla, S., Wang, Y., Lim, H. Y., Zhu, A. X. & Cheng, A.-L. Atezolizumab plus Bevacizumab in Unresectable Hepatocellular Carcinoma. *The New England Journal of Medicine* **382**, 1894-1905 (2020).
6. McLaughlin, M., Patin, E. C., Pedersen, M., Wilkins, A., Dillon, M. T., Melcher, A. A. & Harrington, K. J. Inflammatory microenvironment remodelling by tumour cells after radiotherapy. *Nat Rev Cancer* **20**, 203-217 (2020).

Minor comments

(1) “making it became the first successful first-line immunotherapy for HCC around the world”.

Minor grammar problem. Can change to “making it the first ...” English editing by a native speaker is needed throughout the manuscript. Grammar seems off in too many places.

Response: We thank the reviewer for the suggestion, and the reviewer's point is well taken. **The entire manuscript has been meticulously revised and subsequently proofread by professional native speakers prior to resubmission.**

(2) “Most Chinese HCC patients have a history of cirrhosis, which is the predominant risk factor for RILD; therefore, the radiation tolerated dose of the liver is significantly lower than that which is reported in foreign studies.” The intended audience for this manuscript should not be limited to any nation. The way this sentence is constructed misleadingly implies this manuscript is intended for Chinese readers only.

Response: We are sorry for the misleadingly statements. **The content in the Introduction has been**

modified.

(3) “Mouse reactive antibodies to realize dual PD-L1 and VEGFA blockade (DPVB) in the following experiments.” This sentence is broken and I don’t understand its meaning

Response: We thank the reviewer for pointing out this mistake. The sentence means that the mouse-reactive antibodies, anti-PD-L1 and anti-VEGFA, were utilized in the experiments involving dual PD-L1 and VEGFA blockade (DPVB). **The error in the first part of the Results has been rectified.**

(4) “Volcano plot and heatmap analyses showed significantly higher expression of CD8 T cells following LR-DPVB relative to DPVB of exhausted markers” I understand the meaning of the sentence here, but the grammar is broken.

Response: We appreciate the reviewer for pointing out the mistake. **Appropriate modifications have been made in the third part of the Results in revised manuscript.**

(5) “we recognized nine different T-cell clusters that displayed a comparable expression pattern of essential markers defining the lineage of T cells across tumor types” I do not understand what the authors mean by “comparable” here

Response: We sincerely apologize for any confusion caused by the improper use of words. In fact, we want to express “We identified nine distinct T-cell clusters based on their relative expression levels of specific markers, which define the lineage of T cells”. **This sentence has been modified in the third part of the Results in revised manuscript.**

(6) “Since previous studies showed that TCF-1 and SLAMF6 were highly coexpressed in stem-like CD8+ Tpex, while few terminal exhausted CD8+ T expressed SLAMF6, we chose SLAMF6 as a cell-surface marker to distinguish the CD8+ Tpex and terminal exhausted CD8+ T to isolate live stem-like CD8+ Tpex for further investigation (Fig. 4CD)” Sorry. I missed the logic here. Why you cannot use TCF-1 for this purpose?

Response: The comments provided by the reviewer are highly appreciated, and we would like to present our response as follows. Since TCF-1 functions as an intranuclear transcription factor, it is essential to fix and permeabilize cells prior to flow cytometry-based intranuclear protein staining.

However, this fixation and permeabilization process leads to cell death, making TCF-1 unsuitable as a marker for sorting single living cells. Therefore, we employed SLAMF6, a membrane surface marker that exhibits high co-expression with TCF-1 in stem-like CD8⁺ TpeX cells, for the purpose of isolating viable single cells ¹.

References:

1. Miller, B. C., Sen, D. R., Al AboSy, R., Bi, K., Virkud, Y. V., LaFleur, M. W., Yates, K. B., Lako, A., Felt, K., Naik, G. S., Manos, M., Gjini, E., Kuchroo, J. R., Ishizuka, J. J., Collier, J. L., Griffin, G. K., Maleri, S., Comstock, D. E., Weiss, S. A., Brown, F. D., Panda, A., Zimmer, M. D., Manguso, R. T., Hodi, F. S., Rodig, S. J., Sharpe, A. H. & Haining, W. N. Subsets of exhausted CD8 T cells differentially mediate tumor control and respond to checkpoint blockade. *Nat Immunol* **20**, 326-336 (2019).

REVIEWERS' COMMENTS

Reviewer #1 (Remarks to the Author):

Authors have employed more animal models to confirm their hypothesis. Some of the points were addressed. I have no further comments.

Reviewer #2 (Remarks to the Author):

The authors have added substantial new data to confirm the conclusions and answer initial questions using two appropriate animal models. I think the conclusions are well supported by the evidence. I only have some minor suggestions for improvement:

1. The authors responded that, "Considering the integrity of the logic, the data refer to the macrophages and DCs alteration after the LR-DPVB was not included in the revised manuscript." I think these are important data, given the published literature on this topic. I also suggest that these data should be included in the supplement and mentioned in the discussion section.

Reviewer #3 (Remarks to the Author):

Thank you so much for addressing my comments!! My remaining comments:

(1) “we validated this correlation between Treg/Tef ratio and prognosis among overall HCC patients, where surprisingly, we found a significant association between Treg/Tef ratio and patient prognosis.” The authors seem to have missed my point. Treg/Tef is associated with LR-DPVB but also associated with HCC overall survival. But the “overall HCC patients” do not have any treatment (LR-DPVB). Therefore, Treg/Tef may just be predicting whether the HCC patients on LR-DPVB treatment will inherently do well. But the additional changes in prognosis of the HCC patients due to the treatment is not being predicted by LR-DPVB.

(2) “Volcano plot and heatmap analyses showed that the CD8_T cell in LR-DPVB 195 group exhibited significantly elevated levels of exhausted markers”. “CD8_T” -> “CD8⁺ T”. “exhausted markers” -> “exhaustion markers”?

POINT-TO-POINT RESPONSE TO REVIEWERS' COMMENTS

I would like to express my gratitude to the reviewers for their valuable feedback, which has significantly enhanced the scholarly value of this work.

In the following point-by-point responses, the reviewer's comments are presented in blue, while our responses are displayed in black.

Reviewer #1 (Remarks to the Author):

Authors have employed more animal models to confirm their hypothesis. Some of the points were addressed. I have no further comments.

Response: We would like to thank the reviewer for valuable suggestions and comments during the previous revision of the manuscript.

Reviewer #2 (Remarks to the Author):

The authors have added substantial new data to confirm the conclusions and answer initial questions using two appropriate animal models. I think the conclusions are well supported by the evidence. I only have some minor suggestions for improvement:

1. The authors responded that, "Considering the integrity of the logic, the data refer to the macrophages and DCs alteration after the LR-DPVB was not included in the revised manuscript." I think these are important data, given the published literature on this topic. I also suggest that these data should be included in the supplement and mentioned in the discussion section.

Response: We thank the reviewer's suggestions. As recommended by the reviewer, the data refer to the macrophages and DCs alteration after the LR-DPVB have been include in the **Supplementary Fig. 9** and we discussed these findings in the **Discussion section of revised manuscript**.

Reviewer #3 (Remarks to the Author):

Thank you so much for addressing my comments!! My remaining comments:

(1)“we validated this correlation between Treg/Tef ratio and prognosis among overall HCC patients, where surprisingly, we found a significant association between Treg/Tef ratio and patient prognosis.” The authors seem to have missed my point. Treg/Tef is associated with LR-DPVB but also associated with HCC overall survival. But the “overall HCC patients” do not have any treatment (LR-DPVB). Therefore, Treg/Tef may just be predicting whether the HCC patients on LR-DPVB treatment will inherently do well. But the additional changes in prognosis of the HCC patients due to the treatment is not being predicted by LR- DPVB.

Response: We express our gratitude to the reviewer for the valuable comments. In animal experiments, our study demonstrated that LR-DPVB exerts a superior combined anti-tumor effect in animal experiments, accompanied by changes of *Treg/Tef* ratio. Using clinical samples, we further verified the correlation between Treg/Tef ratio in peripheral blood of HCC patients after T+A treatment and tumor progression. These findings suggested that a decreased Treg/Tef ratio may reflect an augmentation of the anti-tumor immune response to some extent. Due to the high heterogeneity of HCC and potential confounding effects from other anti-tumor therapies, there were significant variations in the Treg/Tef ratio among different patients with HCC. This prompted us to conduct a comprehensive investigation into the prognostic significance of Treg/Tef ratio in an expanded cohort of HCC patients. Through rigorous statistical analysis, we have preliminarily demonstrated a significant association between this ratio and the prognosis of HCC patients, indicating it may serve as a potential pathological marker for assessing the systemic anti-tumor immune response in HCC.

(2)“Volcano plot and heatmap analyses showed that the CD8_T cell in LR-DPVB group exhibited significantly elevated levels of exhausted markers”. “CD8_T” -> “CD8⁺ T”. “exhausted markers” -> “exhaustion markers”?

Response: We thank the reviewer's advice and this point has been well taken. **Appropriate modifications have been implemented in the third section of the Results in the revised manuscript.**